# LONG-TAILED TEST-TIME ADAPTATION FOR VISION-LANGUAGE MODELS

**Xucong Wang**[1,3], **Zhe Zhao**[1,3], **Zekun Wang**[1], **Xiaofeng Cao**[4], **Xu Wang**[1,2], **Di Wu**[1],
**Pengkun Wang**[1,2,*] & **Yang Wang**[1,2,*]
[1] University of Science and Technology of China (USTC)
[2] Suzhou Institute for Advanced Research, USTC [3] City University of Hong Kong
[4] School of Computer Science and Technology, Tongji University
{xuco,zz4543,wangzekun,wdcxy}@mail.ustc.edu.cn,
xiaofengcao@tongji.edu.cn, {wx309,pengkun,angyan}@ustc.edu.cn

## ABSTRACT

Test-Time Adaptation (TTA) aims to further adapt models to unlabeled test sets arriving in a sequential datastream, thereby progressively strengthening the model's generalization ability. While existing TTA methods for Vision-Language Models (VLMs) are primarily designed and evaluated on (nearly) balanced dataset configurations, real-world test sets may exhibit a long-tailed distribution where major classes dominate the decision boundaries of minor classes, presenting unique challenges. As the first attempt to solve this problem, this paper proposes Long-tailed Test-Time Adaptation (dubbed as L-TTA), which consists of three co-designed mechanisms: Synergistic Prototypes (SyPs), Rebalancing Shortcuts (RSs), and Balanced Entropy Minimization (BEM). SyPs introduce two fine-grained prototypes to enrich tail classes with extra inter-class knowledge; RSs employ learnable shortcuts to achieve learnable adaptation, regularized by class re-allocation loss to enforce distinct feature clustering; BEM restrains excessive entropy minimization of confident classes with extra penalty term, with theoretical propositions to justify its rebalancing capabilities. Extensive experiments over 15 datasets under various long-tailed settings highlight the superior performance of L-TTA in both accuracy and class balancing. Code: https://github.com/xuc865/LTTA.

## 1 INTRODUCTION

Pretrained Vision-Language Models (VLMs) like CLIP (Radford et al., 2021) and ALIGN (Jia et al., 2021), show remarkable zero-shot and generalization ability, giving credit to their strong joint-modality modeling capabilities (Li et al., 2022d; Gandelsman et al., 2023; 2024) that learn from web-scaled image-text datasets. Aware of this, various studies propose Parameter-Efficient Fine-Tuning (PEFT) (Jia et al., 2022; Bahng et al., 2022; Zhao et al., 2023b; Han et al., 2024; Tian et al., 2024; Sheng et al., 2025) over VLMs to efficiently transfer them to specific downstream tasks; as two leading approaches in PEFT, adapter tuning (Zhang et al., 2022; Gao et al., 2024) and prompt learning (Zhou et al., 2022b; Wang et al., 2022; Lu et al., 2022; Khattak et al., 2023; Wang et al., 2023a; Yao et al., 2024) freeze the model and only finetunes several learnable modules plugged in layers, enjoying higher efficiency and wider flexibility than the full finetuning.

However, the dramatic distribution gap (Lu et al., 2022) between the labeled training sets and the unlabeled test sets always hinders the generalization of VLMs over the latter ones. To circumvent this issue, recent studies introduce a novel scheme named Test-Time Adaptation (TTA) (Wang et al., 2020; Niu et al., 2022; Zhang et al., 2024c; Shu et al., 2022; Zhang et al., 2025c;a) for VLMs, which enables the model to adjust itself to test sets following an unsupervised **"Entropy-Minimization (EM)"** manner (Wang et al., 2020; Shu et al., 2022; Feng et al., 2023; Yoon et al., 2024; Zhang et al., 2024c;b; Gao et al., 2024; Zanella & Ben Ayed, 2024) during inference. For example, TPT (Shu et al., 2022) and DiffTPT (Feng et al., 2023) select the most confident augmented views and minimize the entropy of prediction probabilities over these views; Following methods

---
*Corresponding Author.

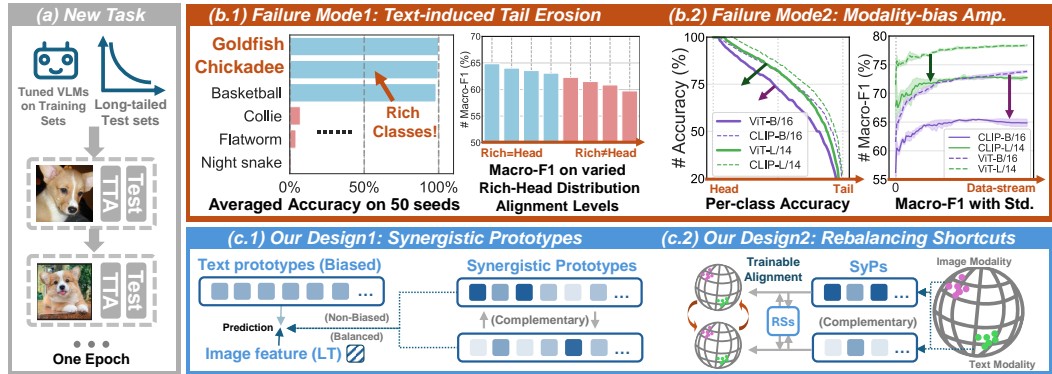

Figure 1: **(a):** The TTA task under long-tailed settings with VLMs. **(b):** Specific failure modes for LT-TTA with VLMs. **(c):** Two targeted designs: Synergistic Prototypes and Rebalancing Shortcuts.

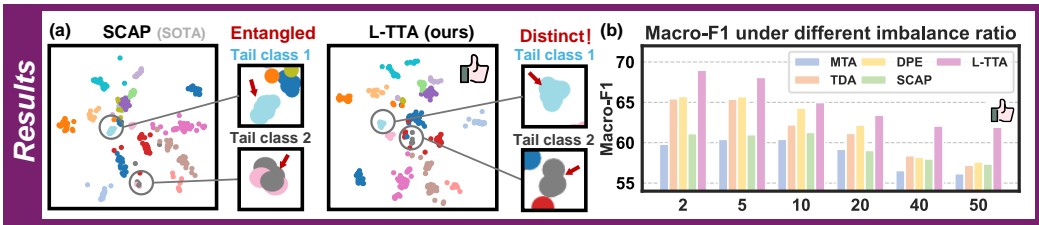

Figure 2: **Results:** Existing SOTAs show severe degradation in tail-class representations (**Left**) and vulnerability to imbalance ratios (**Right**), while our L-TTA shows considerable robustness.

introduce extra knowledge or regularization over prediction probabilities (i.e, C-TPT (Yoon et al., 2024), O-TPT (Sharifdeen et al., 2025)), among supportive views (i.e, SwapPrompt (Ma et al., 2023), PromptAlign (Abdul Samadh et al., 2023)) or historical features (i.e, DMN-ZS (Zhang et al., 2024d), HistTPT (Zhang et al., 2024c), TDA (Karmanov et al., 2024), DPE (Zhang et al., 2024a), BPRE (Qiao et al., 2025)). Another focus is to dynamically schedule different prompts to perceive varied semantics, thus making them specialized and avoiding the error accumulation over a single prompt (Zhang et al., 2024b; Xiao et al., 2025; Wang et al., 2025).

Despite their superior performance on (nearly) balanced datasets, these methods suffer from significant degradation on Long-Tailed (LT) test sets, as evidenced in Figure 2. A primary and fundamental reason is that *TTA is a one-epoch process where training and testing are intertwined, causing later predictions to be influenced by all preceding steps* (except for methods that perform TTA individually for each sample, like TPT (Shu et al., 2022)). This characteristic also precludes the use of traditional LT strategies like up-sampling or compute-intensive regularization. We further identify two specific failure modes unique to the VLM-based LT-TTA setting: ❶ **Text-induced Tail Erosion:** The textual modality exacerbates long-tailed challenges because text embeddings themselves carry biases from pre-training. As demonstrated in Figure 1 (b.1), certain classes consistently yield higher accuracy than others, regardless of their status as head or tail classes. We refer to these as *rich classes*. When *rich classes* coincide with *head classes*, tail erosion is further intensified. ❷ **Modality-bias Amplification:** Applying unimodal LT-TTA methods (Niu et al., 2023; Zhao et al., 2023a) to VLMs progressively amplifies the inherent mismatch between visual and textual representations. Figure 1(b.2) shows that adapting the unimodal SAR method on a VLM backbone results in a significant performance drop and instability compared to its use on a pure visual backbone. This underscores the critical need for bi-modal adaptation that refine the multi-modal manifold of VLMs.

To circumvent these problems, we propose Long-tailed Test-Time Adaptation (L-TTA). First, L-TTA equips two prototypes to accumulate multi-modal semantics beyond text embeddings, meanwhile apply a more granular update strategy to augment tail class representations. (▶ Mitigating **Asp. I, II**); Then, to dynamically balance the accumulated knowledge of head/tail classes, L-TTA introduces learnable Rebalancing Shortcuts (RSs) that are directly applied to the prototypes, optimized with a class re-allocation loss to boost the discernable feature clustering. (▶ Mitigating **Asp. II**) Furthermore, we propose Balanced EM (BEM) to counteract the head-class bias inherent in EM, providing a tailored optimization objective for LT-TTA. BEM weighs the class priors with

the prediction confidence, thereby favoring finer adaptation of uncertain and tail classes; Also, we introduce two propositions to guarantee its theoretical capabilities. Extensive experiments demonstrate that L-TTA outperforms existing methods in various benchmarks under long-tailed settings (see T-SNE and macro-F1 comparison reported in Figure 2), showing remarkable robustness to the noise settings along with high computational efficiency. Our contribution can be summarized as:

❶ We first study the Test-Time Adaptation (TTA) under long-tailed scenarios, and highlight the drawbacks of existing approaches under such circumstances.

❷ We introduce L-TTA, the first TTA for long-tailed settings. L-TTA introduces synergistic Prototypes (SyPs), learnable Rebalancing Shortcuts (RSs), and Balanced Entropy Minimization (BEM) to deal with degraded tail distributions and decision boundaries.

❸ Extensive experiments show that L-TTA surpasses existing methods in both accuracy and class balancing capabilities. Ablation studies show that L-TTA is efficient and robust.

## 2 RELATED WORKS

### 2.1 VLMS & TEST-TIME ADAPTATION FOR VLMS

We have observed an exponential advancement of Vision-Language Models (VLMs) (Radford et al., 2021; Wang et al., 2023b; Hurst et al., 2024), which learn and modulate diversified information as human do. As one of the earliest VLMs (Radford et al., 2021; Jia et al., 2021; Li et al., 2022a; Liu et al., 2023; Hurst et al., 2024), CLIP introduces image-text contrastive pretraining to learn the modality-shared space with the web-scaled unlabeled data, which endows it with strong zero-shot generalization ability and wide compatibility (Wang et al., 2023b; Rao et al., 2025).

Building upon this, emerging studies propose Parameter-Efficient FineTuning (PEFT) mechanisms to efficiently adapt CLIP (Zhang et al., 2022; 2023; Liu et al., 2024). However, they mainly adopt the labeled sets for finetuning and struggle to generalize to domain-shifted test-sets. To deal with this, Test-time Adaptation (TTA) refines VLMs on test-data with an unsupervised Entropy Minimization (EM) scheme; For example, TPT selects augmented views yielding highest confidence and applies EM to optimize the text prompts; Following advancements concentrate on 1) calibrating the uncertainty in predictions (Yoon et al., 2024; Sharifdeen et al., 2025); 2) applying extra regularizations between views (Ma et al., 2023; Abdul Samadh et al., 2023); 3) caching historical predictions (Zhang et al., 2024d; Karmanov et al., 2024; Qiao et al., 2025); 4) Exploring visual adaptations or reinforcement learning (Osowiechi et al., 2024; Hakim et al., 2025; Zhao et al., 2023b).

Notably, an emerging line of TTA also focuses on non-i.i.d. test-data (Boudiaf et al., 2022; Gong et al., 2022; Niu et al., 2023; Wang et al., 2024; Zhao et al., 2023a). For example, LAME (Boudiaf et al., 2022) adjusts model outputs via Laplacian regularized maximum-likelihood estimation to avoid catastrophic degradation. DA-TTA (Wang et al., 2024) aligns source and test distributions using a dedicated loss and incorporates domain shift detection for continual adaptation. SAR (Niu et al., 2023) is a sharpness-aware and reliable EM that leverages group/layer normalization to stabilize TTA. DELTA (Zhao et al., 2023a) uses batch-renormalization and online re-weighting to reduce class bias. Our L-TTA differs from these methods with its unique focus on Long-tailed TTA of VLMs, where the cross-modality misalignment and text-prior bias pose unique challenges.

### 2.2 LONG-TAILED LEARNING

Long-Tailed (LT) Learning Li et al. (2024b); Zhang et al. (2025b); Jin et al. (2023); Li et al. (2024a; 2022b) is built upon the reality that real-world datasets always exhibit a pervasive long-tailed distribution, in which learning head classes would result in degraded decisions of tail classes. Classical LT methods include Augmentation (Chawla et al., 2002; Kuo et al., 2020), resampling (Wallace et al., 2011; Chawla et al., 2002), reweighting (Menon et al., 2020; Cui et al., 2019; Cao et al., 2019; Ren et al., 2020), scaling (Li et al., 2023), pre-defining unbiased targets (Li et al., 2022c) or contrastive learning (Zhu et al., 2022; Du et al., 2024). Recently, several studies focus on LT problem over VLMs, for example, LTGC (Zhao et al., 2024) proposes a generative framework leveraging recursive reasoning and filtering operations to enrich tail classes; LPT (Dong et al., 2022) applies the Visual Prompt Tuning (VPT) to long-tailed recognition by dynamically enriching semantic groups between tail classes. Candle (Shi et al., 2024) introduces virtual prototypes and a refined logit-adjustment loss

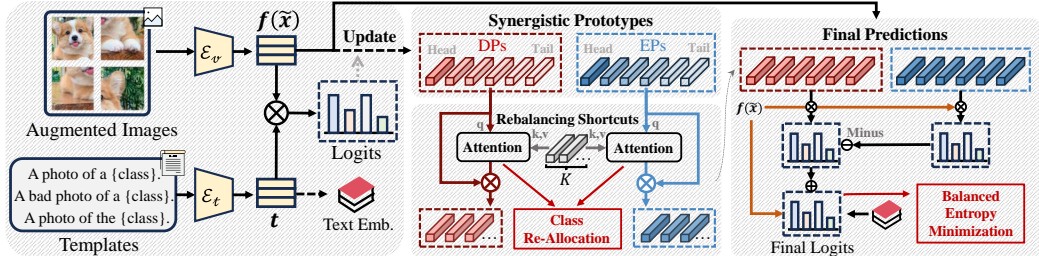

Figure 3: The overview of L-TTA. For an image, we obtain its augmented views and primary logits. In ***Synergistic Prototypes (SyPs)***, we update DPs and EPs with confident and uncertain visual embeddings evaluated by these primary logits, then synthetically combine them to enrich tail representations. In ***Rebalancing Shortcuts***, the SyPs are combined with learnable shortcuts and adapted with class re-allocation loss. The final logits are optimized with ***Balanced Entropy Minimization***.

to solve the long-tailed few-shot adaptation of VLMs. In contrast, we first concentrate on the LT problem in test-time adaptation of VLMs.

## 3 METHODOLOGY

### 3.1 PRELIMINARIES

We begin by presenting the general settings of TTA on VLMs: given a pretrained CLIP $F$ comprising the visual encoder $V = \{V_1, V_2, \cdots, V_L\}$ and text encoder $T = \{T_1, T_2, \cdots, T_L\}$, we aim to further adjust the prompt (could be hand-crafted like "a photo of a [CLASS]." or pre-optimized (Zhou et al., 2022a;b)) $\boldsymbol{m} \in \mathbb{R}^{D_t}$ to fit the test data. Here $D_t$ is the text hidden dimension. Concretely, we're firstly given the class texts of all samples, denoted as $\boldsymbol{y}$. We denote $\mathcal{C}$ as classes, $\mathcal{C}_i$ as a class, $|\mathcal{C}_i|$ as its class cardinality (size), $C$ as number of classes. For an image $\boldsymbol{x} \in \mathcal{X}_{test}$, we generate its visual embeddings with $V$, and textual embeddings of all classes with $T$, where each layer $l$ of $\mathcal{E}_v/\mathcal{E}_t$ is:

$$
\begin{aligned}
[\boldsymbol{c}_l, \boldsymbol{i}_l] &= \mathcal{E}_{v,l}([\boldsymbol{c}_{l-1}, \boldsymbol{i}_{l-1}]); \ \boldsymbol{i}_0 = \mathrm{E}_v(\boldsymbol{x}); \\
[\boldsymbol{e}_l, \boldsymbol{k}_l, \boldsymbol{t}_l] &= \mathcal{E}_{t,l}([\boldsymbol{e}_{l-1}, \boldsymbol{k}_{l-1}, \boldsymbol{t}_{l-1}]); \ \boldsymbol{k}_0 = \mathrm{E}_t([\boldsymbol{m}, \boldsymbol{y}])
\end{aligned}
\tag{1}
$$

Where $\boldsymbol{i}_l$, $\boldsymbol{k}_l$ are intermediate visual and textual features. $\boldsymbol{c}_l$, $\boldsymbol{e}_l$ are cls/eot-token; After obtaining $\boldsymbol{c}_L$ and $\boldsymbol{e}_L$ from the last layer, we map them into a joint space with separate projections $P_v$ and $P_t$:

$$
\boldsymbol{f} = P_v(\boldsymbol{c}_L); \quad \boldsymbol{t} = P_t(\boldsymbol{e}_L);
\tag{2}
$$

Here $\boldsymbol{f} \in \mathbb{R}^D$, $\boldsymbol{t} \in \mathbb{R}^{C \times D}$. $D$ is the dimension of the joint space. We generate output logits $\boldsymbol{z}$ by calculating the cosine similarity, i.e, $\boldsymbol{z} = cos(\boldsymbol{f}, \boldsymbol{t})$. The final predictions are:

$$
\mathbb{P}(y_c|\tilde{\boldsymbol{x}}) = \frac{\exp(z_c/\tau)}{\sum_{c'} \exp(z_{c'}/\tau)}; \ \hat{y} = \mathtt{argmax}_c \mathbb{P}(y_c|\tilde{\boldsymbol{x}})
\tag{3}
$$

Here $\tau$ is the temperature to control the density of predictions, $\mathtt{cos}$ is the cosine similarity. ***In the following, we also denote $\sigma$ as the softmax function, $\boldsymbol{f}(\boldsymbol{x})$ as the visual embeddings of $\boldsymbol{x}$.***

### 3.2 L-TTA: LONG-TAILED TEST-TIME ADAPTATION

We then introduce the framework of L-TTA (shown in Figure 3), consisting of Synergistic Prototypes, Rebalancing Shortcuts, and Balanced Entropy Minimization.

▶ **Synergistic Prototypes.** Leveraging prototypes to maintain historical knowledge has emerged as a trend in recent TTA methods. However, this scheme faces the following tricky problem under long-tailed scenarios: *Tail-class prototypes are more likely to be uninitialized at the beginning phase, while persistently storing inadequate semantics as the datastream progresses.* To solve this problem, we introduce our first innovation named Synergistic Prototypes (SyPs), which comprises Deterministic Prototypes (DPs) and Exclusionary Prototypes (EPs). DPs maintain the similar functionality as well-studied prototypes in (Karmanov et al., 2024; Zhang et al., 2024a) to register class-deterministic features; In contrast, EPs store the most improbable features of each class **generated from all possible samples**, meaning that **EP of each class can be always updated along the datastream**, thereby alleviating the above problem. We elaborate on our innovation in the following.

Concretely, for image $\boldsymbol{x}$ at the step $s$, we randomly crop $Q$ views $\tilde{\boldsymbol{x}} = \{\tilde{\boldsymbol{x}}_i\}_{i=1}^Q$ from it, generating their visual embeddings $\boldsymbol{f}(\tilde{\boldsymbol{x}})$ and predictions $\mathbb{P}(\boldsymbol{y}|\tilde{\boldsymbol{x}}) = \sigma(\boldsymbol{f}(\tilde{\boldsymbol{x}})\boldsymbol{t})$. Recall $\boldsymbol{t}$ is the pre-stored textual embeddings. We gradually update our **Deterministic Prototypes (DPs)** $\boldsymbol{v} = \{\boldsymbol{v}_c\}_{c=1}^C$ of the maximal class $c^*$ in $\mathbb{P}(\boldsymbol{y}|\tilde{\boldsymbol{x}})$, i.e, $\boldsymbol{v}_{c^*}$, with embeddings yielding lower prediction entropy than the threshold $\theta$ via an Exponential Moving Average (EMA) manner:

$$\boldsymbol{v}_{c^*} \leftarrow \frac{(N_{c^*,s}^{\text{DP}} - 1)\boldsymbol{v}_{c^*} + \tilde{\boldsymbol{v}}_{c^*}}{||(N_{c^*,s}^{\text{DP}} - 1)\boldsymbol{v}_{c^*} + \tilde{\boldsymbol{v}}_{c^*}||}, \tilde{\boldsymbol{v}}_{c^*} = \texttt{avg}_{i \in \mathcal{T}} \boldsymbol{f}(\tilde{\boldsymbol{x}}_i), \quad s.t. \ \mathcal{T} = \{i | \mathbb{H}(\mathbb{P}(\boldsymbol{y}|\tilde{\boldsymbol{x}}_i)) < \theta\} \quad (4)$$

where $\mathbb{H}(\cdot)$ calculates the entropy, $|| \cdot ||$ calculates the Euclidean norm. $N_{c,s}^{\text{DP}}$ is the update counter for class $c$ in DPs until step $s$ and increases by 1 at each step. If $\mathcal{T} \neq \varnothing$, we update $\theta$ with the minimal entropy in $\mathcal{T}$ following the above EMA manner. Although not all views are eligible to represent classes and some of them are even ambiguous or subject to multiple classes, their prediction distributions could reveal fine semantic correlations between classes; By dynamically maintaining the visual embeddings $\boldsymbol{f}(\tilde{\boldsymbol{x}})$ **for all the classes** into extra prototypes, we could indirectly exclude the features least likely to occur in every class accordingly. This idea leads to our design of **Exclusionary Prototypes (EPs)** $\boldsymbol{u} = \{\boldsymbol{u}_c\}_{c=1}^C$: Specifically, for logit $\mathbb{P}(y_c|\tilde{\boldsymbol{x}}_i)$ of each view $\tilde{\boldsymbol{x}}_i$, we update EPs of all classes based on $\mathbb{P}(y_c|\tilde{\boldsymbol{x}}_i)$ as the following:

$$\boldsymbol{u}_c \leftarrow \frac{(N_{c,s}^{\text{EP}} - \phi_c)\boldsymbol{u}_c + \tilde{\boldsymbol{u}}_c}{||(N_{c,s}^{\text{EP}} - \phi_c)\boldsymbol{u}_c + \tilde{\boldsymbol{u}}_c||}, \ \tilde{\boldsymbol{u}}_c = \boldsymbol{f}(\tilde{\boldsymbol{x}}_i), \ \phi_c = \frac{\max_{c'} \mathbb{P}(y_{c'}|\tilde{\boldsymbol{x}}_i) - \mathbb{P}(y_c|\tilde{\boldsymbol{x}}_i)}{\max_{c'} \mathbb{P}(y_{c'}|\tilde{\boldsymbol{x}}_i)}; \quad \forall c \in \mathcal{C} \quad (5)$$

Recall $C$ is the number of classes, $N_{c,s}^{\text{EP}}$ is the update counter for class $c$ in EPs until step $s$ and increases by $\phi_c$ at each step. Notably, our EPs largely differ from TDA (Karmanov et al., 2024)'s "negative cache", which selects visual features whose prediction entropy is within specific thresholds and combine them into the negative cache of *the predicted class*. EP employs the prediction of every view to guide EP updates of *all classes*, consequently capturing more refined inter-class associations and enriching tail class representations. Furthermore, this mechanism also grants EPs considerable robustness against OOD semantics, by endowing them with less $\phi_c$ in EMA updating.

▶ **Rebalancing Shortcuts.** Besides statically storing historical knowledge, learnable adaptation enables the model to dynamically adjust the class balance and rectify its predictions. Considering that prompts in existing TTA methods bring extra gradient flow across the text encoder, we keep the prompts frozen (Zhang et al., 2024a), then introduce learnable **Rebalancing Shortcuts (RSs)** over our SyPs to achieve dynamic adaptation. Furthermore, considering the peculiarity of our long-tailed TTA settings, we design RSs as the impetus to boost the active feature clustering and interactions of different classes in our SyPs. Formally, RS is implemented by a cross-attention with shared hyper-class vectors; assume there are $K$ hyper-class vectors $\boldsymbol{q} = \{\boldsymbol{q}_j\}_{j=1}^K$; we treat DPs $\boldsymbol{v}$, EPs $\boldsymbol{u}$ as queries, and merge the RSs with the following formula:

$$\boldsymbol{v}_c \leftarrow \texttt{Attn}([\boldsymbol{v}_c, \boldsymbol{t}_c], \boldsymbol{q}_j)\boldsymbol{q}_j + \boldsymbol{v}_c; \quad \boldsymbol{u}_c \leftarrow \texttt{Attn}([\boldsymbol{u}_c, \boldsymbol{t}_c], \boldsymbol{q}_j)\boldsymbol{q}_j + \boldsymbol{u}_c; \quad (6)$$

Where $\texttt{Attn}$ calculates the attention score. We aim to ensure that these hyper-class vectors achieve ideal clustering atop the prototypes and promote knowledge transfer between head and tail classes. Gaining inspiration from the Load Balancing Loss (LBL) of mixture-of-experts in LLMs (Qiu et al., 2025), we propose to treat each hyper-class as an expert, then incorporate a Class Re-Allocation (CRA) loss to delegate each expert with evenly distributed class semantics (notably, before Eq. 6):

$$\mathcal{L}_{\text{CRA}} = \sum_j (\texttt{avg}_c(c_{c,j}(\boldsymbol{v})) \cdot \texttt{avg}_c\texttt{Attn}([\boldsymbol{v}_c, \boldsymbol{t}_c], \boldsymbol{q}_j) + \texttt{avg}_c(c_{c,j}(\boldsymbol{u})) \cdot \texttt{avg}_c\texttt{Attn}([\boldsymbol{u}_c, \boldsymbol{t}_c], \boldsymbol{q}_j)) \quad (7)$$

Where $c_{c,j}(\boldsymbol{v}) = \mathbb{1}(j = \texttt{Argmax}_{j'}(\texttt{Attn}([\boldsymbol{v}_c, \boldsymbol{t}_c], \boldsymbol{q}_{j'})))$ is the pseudo label generated by binarizing attention scores in each row; $\mathbb{1}(x)$ equals 1 when $x$ is true, else 0. Here, the average of attention scores over all class prototypes for expert $j$, i.e, $\texttt{avg}_c\texttt{Attn}(\cdot, \boldsymbol{q}_j)$ is treated as the expert activations; $\texttt{avg}_c(c_{c,j}(\cdot))$ stands for the counts of expert $j$ being the top-1 entry in $\texttt{Attn}(\cdot, \boldsymbol{q}_j)$. By minimizing the dot product of above two terms for $\boldsymbol{u}$ and $\boldsymbol{v}$, we encourage all prototypes to distribute their attention weights uniformly among hyper-class vectors, resulting in discernable feature clustering and reducing dominance of head-class prototypes. The synergy between SyPs and RSs operates as follows: RSs rely on the comprehensive and stable prototypes from SyPs to compute reallocations, SyPs leverage RSs to go beyond static caching and enable proactive refinement.

The next part will summarize how summarizes how SyPs and RSs enhance predictions; additionally, we propose a Balanced Entropy Minimization (BEM), a variant of EM that will be jointly employed with CRA to optimize shortcuts.

▶ **Final Predictions & Balanced Entropy Minimization.** We first introduce the final prediction of L-TTA at each step for both training and inference, which comprehensively utilizes the modified synergistic prototypes for enhanced and robust predictions:

$$\mathbb{P}_{\texttt{LTTA}}(y_c|\boldsymbol{x}) = \sigma(\boldsymbol{f}(\boldsymbol{x})\boldsymbol{t}_c + \mathcal{A}(\boldsymbol{f}(\boldsymbol{x})\boldsymbol{v}_c) - \mathcal{A}(\boldsymbol{f}(\boldsymbol{x})\boldsymbol{u}_c)) \quad (8)$$

Where $\mathcal{A}(x) = \lambda_1 \exp(-\lambda_2(1-x))$ is an affinity function for scaling (Gao et al., 2024). $\lambda_1, \lambda_2$ are hyper-parameters. We then introduce our Balanced Entropy Minimization (BEM) to optimize this prediction at each step. Entropy Minimization (EM) (Wang et al., 2020; Shu et al., 2022) is pervasive in TTA methods; In contrast, in the context of long-tailed TTA tasks, EM would specifically degrade the decision boundary of tail classes. We introduce Proposition 1 to formalize this claim:

**Proposition 1** *For the long-tailed TTA tasks, denote $\boldsymbol{z} = \{z_1, z_2.., z_C\}$ as the output logits, in which we assume $|\mathcal{C}_1| > |\mathcal{C}_2| > .. > |\mathcal{C}_C|$. $|\mathcal{C}_i|$ is the class cardinality of i. We split $\mathcal{C}$ into $\mathcal{C}_{\texttt{head}}$ and $\mathcal{C}_{\texttt{tail}}$ with certain measurements. The following holds true: $\mathbb{E}_{i \sim \mathcal{C}_{\texttt{head}}} \nabla_{z_i} \mathbb{H} < 0 < \mathbb{E}_{i \sim \mathcal{C}_{\texttt{tail}}} \nabla_{z_i} \mathbb{H}$.*

We defer the proof to **Appx. A.** This proposition indicates that predictions of head-class become more confident than those of tail-classes after EM optimization, as they are most prone to constitute the maximal term. Unfortunately, solutions to this problem are non-trivial, owing to the distinct characteristics of EM relative to classic Cross Entropy loss: EM is unsupervised and only amplifies the model's intrinsic bias; Moreover, its gradient with respect to the logits does not exhibit a clear linear relationship with the logit distribution; i.e, if the distribution becomes sharper, the positive optimization obtained by the confident class may also increase. This implies that when combining class priors into the logits for rebalancing like what logit adjustment (Menon et al., 2020) or balanced softmax (Ren et al., 2020) did, we may further exacerbate the model's bias toward the head classes and damage the decision boundaries. Therefore, a unique variant of EM specifically to long-tailed settings is of necessity. This paper then introduces **Balanced Entropy Minimization (BEM)**, designed with an intuitive rationale: since adjusting high-confidence classes may further amplify the bias, we can focus on calibrating uncertain classes, actively guiding their optimization with the class priors. In implementation, BEM incorporates an additional penalty term into the EM loss:

$$\mathcal{L}_{\texttt{BEM}} = \mathbb{H}'(\tilde{\mathbb{P}}) = -\sigma(\boldsymbol{z}') \log(\sigma(\boldsymbol{z}')), \quad \boldsymbol{z}' = \boldsymbol{z} - (1 - \tilde{\mathbb{P}})^\beta \log(\frac{\boldsymbol{\pi}}{\sum_i \pi_i}) \quad (9)$$

where $\beta$ is for controlling the penalty degree, $\boldsymbol{\pi} \in \mathbb{R}^C$ is the class prior and set to the cardinality of all classes (notably, the class prior is continually updated based on the current predicted pseudo-labels) $\{|\mathcal{C}_i|\}_{i=1}^C$ in default. Our key innovation here is to add an extra penalty term $(1 - \tilde{\mathbb{P}})^\beta$ to the log-amplified prior, which drastically reduces the contribution of confident classes and favors classes that are both rare and uncertain. Consequently, our BEM maintains the learning of head samples, while achieving finer adaptation of tail samples in capturing more discernable semantics, thereby mitigating the deviation of the decision boundary. Further comparisons of BEM and classic LT methods are shown in **Appx. G.** In summary, we introduce the following Proposition 2 to demonstrate the theoretical capability of our proposed BEM:

**Proposition 2** *For the long-tailed TTA tasks, denote $\boldsymbol{z} = \{z_1, z_2.., z_C\}$ as the output logits, in which we assume $|\mathcal{C}_1| > |\mathcal{C}_2|.. > |\mathcal{C}_C|$. $|\mathcal{C}_i|$ is the class cardinality of i. We split $\mathcal{C}$ into $\mathcal{C}_{\texttt{head}}$ and $\mathcal{C}_{\texttt{tail}}$ with certain measurements. The following holds true (Recall $\mathbb{H}'$ is from Eq. 9):*

$$|\mathbb{E}_{i \sim \mathcal{C}_{\texttt{head}}} \nabla_{z_i} \mathbb{H} - \mathbb{E}_{i \sim \mathcal{C}_{\texttt{tail}}} \nabla_{z_i} \mathbb{H}| > |\mathbb{E}_{i \sim \mathcal{C}_{\texttt{head}}} \nabla_{z_i} \mathbb{H}' - \mathbb{E}_{i \sim \mathcal{C}_{\texttt{tail}}} \nabla_{z_i} \mathbb{H}'| \quad (10)$$

The proof is in **Appx. A.** This proposition reveals that applying BEM shortens the optimization gap between head and tail classes. Thus, our BEM is both intuitively and theoretically interpretable. The final objective of L-TTA consists $\mathcal{L}_{\texttt{BEM}}$ and $\mathcal{L}_{\texttt{CRA}}$, connected by a hyper-parameter $\eta$:

$$\mathcal{L}_{\texttt{LTTA}} = \mathcal{L}_{\texttt{BEM}}(\mathbb{P}_{\texttt{LTTA}}) + \eta \mathcal{L}_{\texttt{CRA}} \quad (11)$$

## 4 EXPERIMENTS

▶ **Datasets.** We adopt four benchmarks but manipulate them to follow the long-tailed settings: ❶ OOD Benchmark (OODB), which assesses the robustness of models towards unseen data by conducting TTA over four OOD variants of ImageNet (Deng et al., 2009): ImageNet-A (Hendrycks

Table 1: **Results on Long-tailed OOD Benchmark**. We conduct 5 runs for each experiment with `imb` varying from 10 to 50. The accuracy of head / tail classes are in **Appx. C**. The "OOD Average" is the average results of four OOD datasets excluding ImageNet. The best results are in **Bold**.

| Methods \| imb = 10 | ImageNet-A | | ImageNet-R | | ImageNet-S | | ImageNet-V2 | | ImageNet | | OOD Average | |
|---|---|---|---|---|---|---|---|---|---|---|---|---|
| | Acc. | Mac. | Acc. | Mac. | Acc. | Mac. | Acc. | Mac. | Acc. | Mac. | Acc. | Mac. |
| CLIP [ICML'21] | 46.84 | 43.71 | 72.27 | 70.40 | 42.17 | 40.74 | 59.63 | 51.62 | 63.75 | 58.94 | 58.04 | 50.84 |
| TPT [NeurIPS'22] | 52.79 | 48.08 | 77.33 | 74.28 | 45.22 | 43.60 | 61.34 | 55.70 | 65.53 | 60.36 | 59.17 | 55.42 |
| C-TPT [ICLR'24] | 50.70 | 46.36 | 75.82 | 72.92 | 44.14 | 42.62 | 60.44 | 53.00 | 64.92 | 60.13 | 57.78 | 53.73 |
| MTA [CVPR'24] | 57.15 | 51.98 | 77.04 | 74.88 | 48.59 | **46.50** | 63.61 | 62.69 | 67.79 | 62.57 | 61.60 | 59.01 |
| TDA [CVPR'24] | 60.00 | 54.30 | 81.23 | 78.00 | 47.55 | 45.77 | 66.21 | 60.53 | 69.10 | 63.89 | 63.75 | 59.65 |
| ZERO [NeurIPS'24] | 56.72 | 51.18 | 77.06 | 72.94 | 47.98 | 46.09 | 63.46 | 58.52 | 65.92 | 60.11 | 61.31 | 57.18 |
| RLCF [ICLR'24] | 57.74 | 52.09 | 79.47 | 75.53 | 46.00 | 44.42 | 64.39 | 58.23 | 69.19 | 63.90 | 61.90 | 57.57 |
| DPE [NeurIPS'24] | 60.31 | 54.41 | 80.58 | 77.42 | 49.32 | 45.45 | 67.78 | 63.32 | 70.16 | 64.19 | 64.50 | 60.15 |
| WATT [NeurIPS'24] | 60.33 | 53.06 | 80.29 | 76.96 | 48.25 | 46.83 | 66.75 | 60.36 | 70.04 | 64.48 | 63.91 | 59.30 |
| CLIPArTT [WACV'25] | 58.58 | 53.00 | 78.22 | 73.50 | 46.42 | 44.66 | 65.30 | 58.95 | 68.17 | 62.01 | 62.13 | 57.53 |
| O-TPT [CVPR'25] | 50.03 | 45.61 | 75.45 | 72.61 | 44.06 | 43.79 | 62.53 | 54.92 | 65.28 | 60.76 | 58.02 | 54.23 |
| SCAP [CVPR'25] | 60.54 | 52.26 | 80.57 | 75.48 | 48.97 | 45.38 | 67.41 | 56.55 | 70.64 | 64.80 | 64.37 | 57.42 |
| **L-TTA** (Ours) | **61.78** | **55.97** | **82.86** | **78.56** | **50.25** | 45.99 | **68.99** | **64.19** | **71.30** | **65.83** | **65.97** | **61.18** |
| Methods \| imb = 20 | ImageNet-A | | ImageNet-R | | ImageNet-S | | ImageNet-V2 | | ImageNet | | OOD Average | |
| | Acc. | Mac. | Acc. | Mac. | Acc. | Mac. | Acc. | Mac. | Acc. | Mac. | Acc. | Mac. |
| CLIP [ICML'21] | 46.89 | 41.68 | 72.57 | 69.30 | 42.25 | 40.45 | 58.36 | 50.60 | 63.36 | 57.33 | 55.02 | 50.51 |
| TPT [NeurIPS'22] | 53.30 | 47.11 | 77.76 | 72.82 | 46.04 | 43.79 | 60.78 | 54.81 | 65.22 | 60.70 | 59.47 | 54.63 |
| C-TPT [ICLR'24] | 50.65 | 45.14 | 76.41 | 71.27 | 46.01 | 43.03 | 59.21 | 53.76 | 64.61 | 60.28 | 58.07 | 53.30 |
| MTA [CVPR'24] | 57.15 | 51.98 | 77.03 | 74.87 | 48.60 | **46.64** | 63.93 | 62.68 | 66.53 | 61.60 | 61.68 | 59.04 |
| TDA [CVPR'24] | 60.40 | 53.38 | 81.60 | 75.94 | 45.65 | 43.46 | 64.70 | 56.17 | 68.92 | 63.36 | 63.09 | 57.24 |
| ZERO [NeurIPS'24] | 53.60 | 48.80 | 77.02 | 70.10 | 46.87 | 43.32 | 62.18 | 56.30 | 65.24 | 59.98 | 59.92 | 54.63 |
| RLCF [ICLR'24] | 56.62 | 50.52 | 78.51 | 72.22 | 45.57 | 42.93 | 64.76 | 56.63 | 69.70 | 62.16 | 61.37 | 55.58 |
| DPE [NeurIPS'24] | 60.05 | 53.87 | 81.18 | 75.87 | 50.10 | 45.02 | 65.50 | 60.53 | 67.41 | 63.53 | 64.21 | 58.82 |
| WATT [NeurIPS'24] | 60.08 | 52.58 | 80.38 | 75.14 | 47.15 | 44.60 | 66.61 | 60.02 | 69.39 | 63.83 | 63.56 | 58.09 |
| CLIPArTT [WACV'25] | 57.87 | 52.89 | 79.94 | 72.35 | 45.36 | 42.75 | 65.20 | 58.19 | 68.01 | 61.74 | 62.09 | 56.55 |
| O-TPT [CVPR'25] | 50.22 | 44.92 | 76.02 | 70.87 | 42.84 | 42.93 | 61.83 | 53.47 | 63.88 | 59.90 | 57.73 | 53.05 |
| SCAP [CVPR'25] | 60.36 | 52.20 | 79.52 | 75.44 | 48.45 | 43.12 | 65.84 | 55.97 | 66.83 | 62.97 | 63.54 | 56.68 |
| **L-TTA** (Ours) | **61.23** | **54.79** | **82.31** | **76.48** | **50.44** | 46.24 | **67.29** | **64.55** | **70.35** | **64.10** | **64.92** | **60.52** |
| Methods \| imb = 50 | ImageNet-A | | ImageNet-R | | ImageNet-S | | ImageNet-V2 | | ImageNet | | OOD Average | |
| | Acc. | Mac. | Acc. | Mac. | Acc. | Mac. | Acc. | Mac. | Acc. | Mac. | Acc. | Mac. |
| CLIP [ICML'21] | 44.74 | 36.26 | 71.52 | 62.60 | 39.36 | 36.08 | 58.03 | 50.58 | 63.18 | 53.70 | 53.41 | 46.38 |
| TPT [NeurIPS'22] | 51.13 | 40.84 | 76.77 | 66.30 | 42.99 | 41.28 | 61.27 | 54.95 | 65.29 | 56.57 | 58.04 | 50.84 |
| C-TPT [ICLR'24] | 51.66 | 41.01 | 75.58 | 65.07 | 44.01 | 41.32 | 61.07 | 56.86 | 65.42 | 55.00 | 58.08 | 51.07 |
| MTA [CVPR'24] | 57.15 | 51.98 | 75.04 | 74.88 | 46.59 | 42.77 | 63.61 | **62.69** | 66.29 | 61.44 | 60.60 | 58.08 |
| TDA [CVPR'24] | 58.97 | 49.12 | 80.89 | 70.04 | 44.87 | 41.49 | 64.76 | 58.50 | 68.07 | 62.06 | 62.37 | 54.79 |
| ZERO [NeurIPS'24] | 52.24 | 45.04 | 75.27 | 66.48 | 44.95 | 40.76 | 59.54 | 55.05 | 63.55 | 52.05 | 58.00 | 51.84 |
| RLCF [ICLR'24] | 54.27 | 46.18 | 76.54 | 68.98 | 45.08 | 38.58 | 63.51 | 53.75 | 66.23 | 56.57 | 59.85 | 51.87 |
| DPE [NeurIPS'24] | **60.21** | 47.46 | 80.76 | 69.96 | 48.07 | 43.50 | 65.80 | 60.78 | 68.04 | 62.37 | 63.71 | 55.43 |
| WATT [NeurIPS'24] | 57.86 | 49.83 | 78.05 | 70.24 | 45.80 | 40.49 | 65.37 | 55.63 | 68.69 | 57.72 | 61.77 | 54.05 |
| CLIPArTT [WACV'25] | 55.36 | 48.89 | 78.31 | 69.00 | 44.12 | 40.16 | 64.56 | 54.50 | 66.70 | 57.05 | 60.59 | 53.14 |
| O-TPT [CVPR'25] | 51.93 | 39.87 | 75.24 | 64.84 | 41.27 | 40.67 | 61.70 | 55.69 | 63.38 | 55.69 | 57.54 | 49.75 |
| SCAP [CVPR'25] | 59.05 | 47.00 | 78.08 | 73.11 | 45.86 | 41.03 | 65.31 | 54.19 | 66.30 | 58.85 | 62.08 | 53.83 |
| **L-TTA** (Ours) | 60.07 | **54.79** | **82.01** | **75.83** | **49.73** | **46.01** | **67.10** | 62.48 | **69.74** | **63.41** | **64.68** | **59.78** |

et al., 2021b), ImageNet-V2 (Recht et al., 2019), ImageNet-R (Hendrycks et al., 2021a) and ImageNet-S (Wang et al., 2019). ❷ Cross-Domain Benchmark (CDB), which evaluates the model's performance on ImageNet and 10 fine-grained datasets: Pets (Parkhi et al., 2012), SUN397 (Xiao et al., 2010), EuroSAT (Helber et al., 2019), Caltech101 (Fei-Fei et al., 2004), Cars (Krause et al., 2013), DTD (Cimpoi et al., 2014), UCF (Soomro et al., 2012), Flower102 (Nilsback & Zisserman, 2008), Food101 (Bossard et al., 2014), Aircraft (Maji et al., 2013). ❸ Corruption Benchmark (CB), which adds gaussian noise to images (with the variations $\iota$ ranging from 0.1 to 0.4) to mimic harsher scenarios. Experiments on other 16 corruption types (Hendrycks & Dietterich, 2019) with a severity of 5 can be found in Appendix J. To enable these datasets to show obvious long-tailed distribution (some of them may have already been slightly imbalanced), we conduct random sampling to manipulate the cardinality distribution into an exponentially decayed curve yielding specific imbalance ratio. Notably, if the calculated cardinality is less than the class cardinality itself, we simply keep that class unchanged. We treat the top-20% classes as head and others as tail.

▶ **Implementation Details.** Following previous studies, we adopt a pretrained CLIP for evaluation where the image encoder can be either ResNet-50 or ViT-B/16 (ViT-B/16 in default, the results on ResNet-50 are in **Appx. D**). Following ablation studies on ImageNet, we generate 15 augmented views for a image via random resized cropping. $\eta = 1$, $\lambda_1 = 6$, $\lambda_2 = 6$, $K = 0.3$, $\beta = 1$. We select AdamW as the optimizer with a weight decay of 1e-1 and eps of 1e-3. The imbalance ratio, defined as $\texttt{imb} = \max_i |\mathcal{C}_i| / \min_i |\mathcal{C}_i|$, is selected from $\{10, 20, 50\}$. All experiments are conducted on a single A100 GPU, and methods for comparison are reproduced with their provided hyperparameters. Besides accuracy (Acc.), we also report the macro-F1 (Mac.) to highlight each model's capability in balancing different classes. We compare with classic methods like TPT (Shu et al., 2022), C-TPT (Yoon et al., 2024), O-TPT (Sharifdeen et al., 2025); Training-free methods MTA (Zanella & Ben Ayed, 2024), TDA (Karmanov et al., 2024), ZERO (Farina et al., 2024), visual-adaptation methods: WATT (Osowiechi et al., 2024), CLIPArTT (Hakim et al., 2025) and other training-based methods like RLCF (Zhao et al., 2023b), DPE (Zhang et al., 2024a), SCAP (Zhang et al., 2025a). Details of baselines and prompts are in **Appx. B / E**.

Table 2: **Averaged Results on Long-tailed Cross-Domain Benchmark with an imbalance ratio of** $\{10, 20, 50\}$. Please refer to **Appx. C** for detailed results and head / tail accuracy. We conduct 5 runs for each experiment and average the results. The best results are in **Bold**.

| Methods | Caltech | | Pets | | Cars | | Flowers | | Food101 | | ImageNet | |
|---|---|---|---|---|---|---|---|---|---|---|---|---|
| | Acc. | Mac. | Acc. | Mac. | Acc. | Mac. | Acc. | Mac. | Acc. | Mac. | Acc. | Mac. |
| CLIP [ICML'21] | 93.10 | 89.95 | 83.51 | 83.29 | 61.83 | 60.09 | 65.67 | 60.67 | 82.34 | 78.24 | 63.43 | 56.66 |
| TPT [NeurIPS'22] | 93.82 | 90.99 | 83.51 | 83.10 | 63.18 | 61.41 | 66.94 | 60.97 | 83.49 | 79.85 | 65.35 | 59.21 |
| C-TPT [ICLR'24] | 93.41 | 90.69 | 85.41 | 85.08 | 63.25 | 61.57 | 67.66 | 62.08 | 82.41 | 78.38 | 64.98 | 58.47 |
| MTA [CVPR'24] | 93.48 | 90.97 | 84.36 | 84.33 | 63.78 | 62.41 | 65.32 | 60.24 | 83.61 | 80.17 | 66.87 | 61.87 |
| TDA [CVPR'24] | 94.33 | 90.69 | 86.16 | 83.47 | 68.15 | 59.60 | 71.77 | 62.13 | **85.94** | 80.68 | 68.70 | 63.10 |
| ZERO [NeurIPS'24] | 92.50 | 89.30 | 83.34 | 83.30 | 61.37 | 59.96 | 66.31 | 60.55 | 83.68 | 78.49 | 64.90 | 57.38 |
| RLCF [ICLR'24] | 88.46 | 86.90 | 83.39 | 81.58 | 61.86 | 58.18 | 66.45 | 61.81 | 80.55 | 79.38 | 68.37 | 60.88 |
| DPE [NeurIPS'24] | 94.85 | 91.80 | 90.09 | 85.85 | 68.88 | 61.14 | 73.95 | 65.54 | 84.11 | 78.68 | 68.54 | 63.36 |
| WATT [NeurIPS'24] | 93.33 | 90.35 | 86.02 | 84.91 | 61.95 | 61.03 | 66.32 | 60.87 | 80.94 | 77.31 | 69.37 | 62.01 |
| CLIPArTT [WACV'25] | 91.15 | 88.07 | 85.19 | 84.30 | 60.95 | 59.71 | 64.98 | 60.66 | 80.75 | 76.98 | 67.63 | 60.27 |
| O-TPT [CVPR'25] | 93.41 | 90.61 | 85.25 | 85.01 | 62.68 | 61.16 | 67.63 | 62.18 | 82.39 | 78.35 | 64.18 | 58.78 |
| SCAP [CVPR'25] | 91.40 | 86.96 | 86.15 | 77.26 | 64.50 | 62.35 | 68.10 | 62.43 | 82.40 | 80.25 | 67.92 | 62.21 |
| **L-TTA** (Ours) | **95.36** | **92.29** | **91.07** | **86.41** | **70.10** | **64.13** | **74.28** | **67.68** | 85.55 | **80.94** | **70.46** | **64.39** |

| Methods | Aircraft | | SUN397 | | DTD | | EuroSAT | | UCF101 | | ◆ Average ◆ | |
|---|---|---|---|---|---|---|---|---|---|---|---|---|
| | Acc. | Mac. | Acc. | Mac. | Acc. | Mac. | Acc. | Mac. | Acc. | Mac. | Acc. | Mac. |
| CLIP [ICML'21] | 14.39 | 16.78 | 60.17 | 54.25 | 42.89 | 35.70 | 37.29 | 30.17 | 65.25 | 58.18 | 60.90 | 56.72 |
| TPT [NeurIPS'22] | 14.73 | 17.09 | 63.97 | 57.08 | 43.46 | 36.61 | 37.12 | 30.05 | 66.71 | 60.04 | 62.03 | 57.86 |
| C-TPT [ICLR'24] | 17.42 | 18.76 | 63.22 | 56.65 | 43.62 | 37.28 | 37.76 | 29.53 | 67.46 | 59.10 | 62.42 | 58.37 |
| MTA [CVPR'24] | 17.81 | 18.60 | 62.46 | 56.59 | 43.45 | 36.94 | 40.45 | 30.95 | 67.82 | 61.39 | 62.67 | 58.59 |
| TDA [CVPR'24] | 23.04 | 18.97 | 65.81 | 58.68 | 44.06 | 34.53 | 53.48 | 46.39 | **71.13** | 61.25 | 66.60 | 60.20 |
| ZERO [NeurIPS'24] | 16.51 | 17.36 | 59.33 | 54.63 | 42.88 | 36.03 | 37.94 | 28.41 | 65.97 | 57.63 | 61.34 | 56.64 |
| RLCF [ICLR'24] | 16.08 | 17.17 | 59.13 | 52.75 | 41.80 | 36.77 | 40.49 | 29.03 | 65.75 | 53.75 | 61.12 | 56.20 |
| DPE [NeurIPS'24] | 24.32 | 21.38 | 68.26 | 61.18 | 47.55 | 39.82 | 55.21 | 45.85 | 69.53 | 60.38 | 67.75 | 61.24 |
| WATT [NeurIPS'24] | 17.74 | 18.27 | 60.19 | 55.05 | 45.60 | 37.49 | 45.09 | 33.38 | 68.09 | 59.38 | 63.15 | 58.19 |
| CLIPArTT [WACV'25] | 16.22 | 16.71 | 58.84 | 53.33 | 42.68 | 35.95 | 42.45 | 31.51 | 66.31 | 57.79 | 61.56 | 56.84 |
| O-TPT [CVPR'25] | 17.27 | 18.42 | 63.12 | 56.22 | 43.97 | 37.25 | 38.34 | 30.69 | 66.99 | 58.57 | 62.29 | 58.03 |
| SCAP [CVPR'25] | 22.85 | 19.01 | 63.48 | 61.14 | 42.71 | 36.37 | 46.85 | 38.89 | 66.31 | 62.11 | 63.90 | 59.23 |
| L-TTA (Ours) | **25.49** | **21.88** | **69.01** | **62.61** | **48.50** | **40.71** | **56.53** | **47.99** | 70.58 | **62.75** | **68.77** | **63.44** |

Table 3: **Averaged Results on Long-tailed Corruption Benchmark with** $\iota \in \{0.1, 0.2, 0.4\}$ **and** imb = 10. Please refer to **Appx. J** for results with other corruption types. We conduct 5 runs for each experiment and average the results. The best results are in **Bold**.

| Methods | Caltech | | Pets | | Cars | | Flowers | | Food101 | | ImageNet | |
|---|---|---|---|---|---|---|---|---|---|---|---|---|
| | Acc. | Mac. | Acc. | Mac. | Acc. | Mac. | Acc. | Mac. | Acc. | Mac. | Acc. | Mac. |
| CLIP [ICML'21] | 77.49 | 70.94 | 54.48 | 52.98 | 36.51 | 33.41 | 39.69 | 35.35 | 34.18 | 33.81 | 37.73 | 34.47 |
| TPT [NeurIPS'22] | 81.87 | 75.71 | 60.52 | **58.54** | 39.41 | 38.70 | 44.28 | 39.36 | 38.34 | 38.03 | 42.35 | 39.33 |
| C-TPT [ICLR'24] | 80.00 | 73.61 | 57.98 | 56.28 | 38.07 | 37.49 | 42.93 | 38.53 | 35.77 | 34.94 | 41.00 | 38.07 |
| MTA [CVPR'24] | 79.90 | 73.53 | 57.35 | 55.96 | 39.51 | 38.99 | 27.16 | 35.97 | 35.65 | 35.73 | 42.96 | 40.29 |
| TDA [CVPR'24] | 79.87 | 73.65 | 58.31 | 54.45 | 41.30 | 36.64 | 42.84 | 37.19 | 37.80 | 36.27 | 42.12 | 38.80 |
| ZERO [NeurIPS'24] | 77.74 | 73.99 | 55.75 | 53.74 | 38.97 | 33.60 | 38.78 | 36.75 | 35.08 | 33.95 | 38.93 | 35.49 |
| RLCF [ICLR'24] | 76.95 | 72.17 | 54.38 | 52.11 | 38.72 | 32.51 | 38.06 | 35.37 | 35.16 | 32.92 | 39.79 | 36.30 |
| DPE [NeurIPS'24] | 81.30 | 77.16 | 58.12 | 54.91 | 41.78 | 38.74 | 44.90 | 40.61 | 35.96 | 34.42 | 43.52 | 40.82 |
| WATT [NeurIPS'24] | 80.42 | 76.08 | 55.48 | 52.08 | 43.60 | 36.04 | 41.34 | 37.56 | 38.12 | 36.38 | 41.96 | 39.09 |
| CLIPArTT [WACV'25] | 79.06 | 74.57 | 56.65 | 53.84 | 41.43 | 35.08 | 40.83 | 37.41 | 36.79 | 35.40 | 42.45 | 38.52 |
| O-TPT [CVPR'25] | 80.13 | 73.82 | 58.53 | 56.77 | 37.38 | 36.79 | 42.62 | 38.20 | 35.72 | 34.90 | 40.90 | 39.26 |
| SCAP [CVPR'25] | 79.72 | 72.75 | 59.69 | 55.01 | 40.12 | 37.44 | 41.55 | 35.93 | 37.82 | 36.20 | 42.07 | 38.52 |
| L-TTA (Ours) | **82.45** | **78.35** | **61.98** | 58.53 | **43.48** | **39.46** | **46.68** | **42.72** | **39.55** | **37.92** | **46.11** | **43.10** |

| Methods | Aircraft | | SUN397 | | DTD | | EuroSAT | | UCF101 | | ◆ Average ◆ | |
|---|---|---|---|---|---|---|---|---|---|---|---|---|
| | Acc. | Mac. | Acc. | Mac. | Acc. | Mac. | Acc. | Mac. | Acc. | Mac. | Acc. | Mac. |
| CLIP [ICML'21] | 8.28 | 9.34 | 37.74 | 35.50 | 22.20 | 19.43 | 10.46 | 7.01 | 39.70 | 34.67 | 36.22 | 34.36 |
| TPT [NeurIPS'22] | 9.24 | 10.64 | 41.86 | 38.93 | 24.91 | 22.81 | 7.96 | 5.35 | 45.16 | 39.77 | 39.63 | 37.01 |
| C-TPT [ICLR'24] | 8.93 | 9.48 | 40.50 | 37.77 | 24.54 | 22.32 | 8.35 | 5.36 | 42.96 | 38.55 | 38.27 | 35.67 |
| MTA [CVPR'24] | 10.22 | 10.21 | 39.70 | 37.13 | 22.98 | 20.63 | 14.62 | 6.38 | 42.96 | 37.26 | 37.55 | 35.64 |
| TDA [CVPR'24] | 12.22 | 10.48 | 41.10 | 37.64 | 24.75 | 18.67 | 13.06 | 9.05 | 45.29 | 37.66 | 39.88 | 35.50 |
| ZERO [NeurIPS'24] | 10.52 | 9.07 | 40.72 | 36.54 | 23.32 | 20.08 | 11.40 | 7.18 | 42.84 | 35.46 | 37.64 | 34.17 |
| RLCF [ICLR'24] | 9.34 | 8.16 | 41.68 | 37.63 | 22.72 | 18.58 | 10.88 | 8.04 | 44.22 | 36.14 | 37.45 | 33.63 |
| DPE [NeurIPS'24] | 13.35 | 10.81 | 43.51 | 39.91 | 24.19 | 23.06 | 13.58 | 9.94 | 44.64 | 40.26 | 40.44 | 37.33 |
| WATT [NeurIPS'24] | 12.40 | 10.96 | 42.85 | 39.00 | 26.98 | 22.05 | 15.54 | 12.00 | 44.86 | 36.43 | 40.46 | 36.15 |
| CLIPArTT [WACV'25] | 11.49 | 10.08 | 41.55 | 38.44 | 24.59 | 21.33 | 15.26 | 11.18 | 43.73 | 35.09 | 39.39 | 35.54 |
| O-TPT [CVPR'25] | 9.06 | 9.79 | 39.55 | 36.81 | 24.18 | 22.69 | 7.89 | 6.15 | 40.36 | 36.55 | 37.85 | 35.61 |
| SCAP [CVPR'25] | 11.26 | 11.83 | 40.02 | 38.50 | 24.44 | 22.97 | 11.85 | 7.70 | 43.47 | 39.08 | 39.27 | 35.99 |
| L-TTA (Ours) | **15.65** | **14.57** | **45.01** | **41.24** | **30.94** | **27.43** | **17.42** | **13.94** | **47.21** | **42.46** | **43.31** | **39.97** |

## 4.1 RESULTS & DISCUSSIONS

**Results on the Long-tailed OOD Benchmark.** As demonstrated in Table 1, L-TTA mostly outperforms existing methods in both accuracy and macro-F1 under three imbalance settings. Going a step further, we find that existing methods consistently show performance degradation when the imbalance effects worsen. For example, focusing on the accuracy/macro-F1 of OOD average when changing imb from 10 to 50, we observe a drop of 1.38%/4.86% for TDA, 0.79%/4.72% for DPE, 2.29%/3.59% for SCAP. Interestingly, TPT and C-TPT show even more robustness since they do not involve temporally accumulated knowledge, as indicated by their relatively minor variations under the three imbalance settings. In contrast, L-TTA surpasses previous SOTAs by a significant margin (1.47%/1.70% in accuracy/macro-F1 for OOD average; 1.67%/1.35% for ImageNet); also, L-TTA yields a minor variation of 1.29% in macro-F1 when long-tail effects degrade. These results demonstrate the substantial robustness of L-TTA under long-tailed settings.

Table 4: **Comparisons of complexity.** Here HM is the harmonic mean of Accuracy and Macro-f1. \ means the model fails to provide valid outputs. - means the model did not finish within time budget.

| Methods | TPT | C-TPT | MTA | TDA | ZERO | RLCF | WATT | DPE | CLIPArTT | O-TPT | SCAP | L-TTA |
|---|---|---|---|---|---|---|---|---|---|---|---|---|
| Time (h) | 3.80 | 3.80 | 1.87 | 0.91 | 0.86 | 8.30 | 27.70 | 1.38 | 6.42 | 3.81 | 2.96 | 1.45h |
| Memory (G) | 17.94 | 17.94 | 1.29 | 0.89 | 1.68 | 19.84 | $1.54 \times n$ | 1.81 | 1.71 | 17.94 | 1.97 | 1.89 |
| HM on LT-CDB | 61.12 | 61.06 | 61.75 | 64.51 | 60.42 | 59.71 | 62.07 | 66.31 | 60.54 | 61.39 | 63.31 | 67.20 |
| HM on LT-CB | 40.04 | 38.57 | \ | 41.04 | 38.04 | - | - | 41.93 | - | 38.05 | 40.74 | 46.08 |

**Results on the Long-tailed Cross-Domain Benchmark.** As shown in Table 2, further comparisons on 10 fine-grained datasets and ImageNet prove the universality and robustness of L-TTA again. We observe that prototype-based methods like TDA and DPE generally perform more competitive than other methods. Turing to our L-TTA, it outperforms existing SOTAs for 10 of the 11 datasets and shows a significant enhancement in the averaged metrics. We also find the improvement in Macro-F1 (2.20%) significantly exceeds that of accuracy (1.02%). This demonstrates that L-TTA can handle diverse specific domains besides re-balancing and shows impressive adaptability.

**Results on the Long-tailed Corruption Benchmark.** We aim to mimic more realistic scenarios with this benchmark and report the comparison results in Table 3. We can observe that the performance gains on L-TTA is expanded, which outperforms existing SOTAs by 2.87% and 2.64% in averaged accuracy and macro-F1; moreover, by comparing Table 2 and Table 3, we further reveal that under noisy conditions, the superiority of previous prototype-based diminish substantially, nearly matching the baseline TPT; This may stem from their dependence on high-quality prototypes and incapacity to balance different classes. By contrast, our L-TTA utilizes synergistic prototype-sand active shortcut learning for rebalancing, exhibiting more stability under corrupted settings.

**Efficiency Study.** We show the results on ImageNet in Table 4 with `imb=10`. It is observed that training-free methods like ZERO, MTA and TDA perform suboptimal (especially with corrupted data), for their sensitivity to data quality. DPE yields the

Table 5: Results on other backbones. \: invalid outputs.

| Method | CLIP:ViT/L-14 | | CLIP:ViT/H-14 | | SigLIP-L/16 | | METACLIP-BigG | |
|---|---|---|---|---|---|---|---|---|
| | Acc. | Mac. | Acc. | Mac. | Acc. | Mac. | Acc. | Mac. |
| TPT | 67.16 | 66.08 | 67.57 | 66.45 | 68.09 | 65.56 | \ | \ |
| TDA | 69.75 | 66.95 | 72.12 | 67.56 | 72.40 | 70.46 | 75.76 | 71.78 |
| DPE | 71.03 | 67.13 | 73.00 | 69.07 | 75.50 | 71.23 | 76.74 | 73.32 |
| SCAP | 69.64 | 67.74 | 70.92 | 68.54 | 71.70 | 67.88 | \ | \ |
| L-TTA | 73.07 | 70.15 | 74.67 | 71.37 | 76.75 | 72.77 | 77.91 | 74.46 |

lowest complexity but performs suboptimal; For L-TTA, the main computational overhead is to update two prototypes and shortcuts: prototypes are updated in parallel, while optimizing shortcuts is free of gradient tracking throughout the backbone, thus L-TTA only incurs minor computation overhead. Statistically, L-TTA shows more effectiveness than the cumbersome SCAP, WATT and RLCF which requires the gradient propagation along the visual encoder. Meanwhile, it embraces improvement in both accuracy and macro-F1 (1.14% and 1.67%). Conclusively, L-TTA keeps the trade-off between performance and efficiency.

**Results on Other Backbones.** We further explore the adaptability of L-TTA on stronger backbones. We compare our L-TTA with competitive baselines on four additional backbones (ViT-L/14, ViT-H/14, SigLIP-L/16 (Zhai et al., 2023), and MetaCLIP-BigG (Xu et al., 2023; Chuang et al., 2025)), and report the averaged results over 10 fine-grained datasets in Table 5 (Per-dataset results are in **Appx. L**). The performance gains of our model remain salient (an average of 1.5% Acc. / 1.8% Mac.) across all backbones. This is because L-TTA effectively utilizes sample-level information and actively promotes continuous image-text alignment for re-balancing. We argue that these designs are essential for any backbone when faced with long-tailed TTA.

### 4.2 ABLATION STUDIES AND SENSITIVITY ANALYSIS

**Affinity Coefficients $\lambda_1$ and $\lambda_2$.** We report an ablation study of $\lambda_1$ and $\lambda_2$ on ImageNet with `imb` as 10 in Figure 4.a. Specifically, we find that lowering $\lambda_1$ and $\lambda_2$, i.e., the contributions of SyPs, generally yields poor performance; this indicates that our Synergistic Prototypes and Rebalancing Shortcuts undeniably contribute to the performance. However, we also observe that excessive $\lambda_1$ and $\lambda_2$ values lead to a slight degradation. As a trade-off, we set $\lambda_1$ and $\lambda_2$ as $\lambda_1 = 6$, $\lambda_2 = 6$.

**Different Components.** L-TTA consists of three components: Synergistic Prototypes (SyPs), Rebalancing Shortcuts (RS), and Balanced Entropy Minimization (BEM). We ablate each of them or their combinations under backbone ResNet50 or VIT-B/16 and report the results in Table 6. We find dropping DPs or EPs leads to a decrement of about -3.95% / -3.22% in macro-F1 and -3.77% /

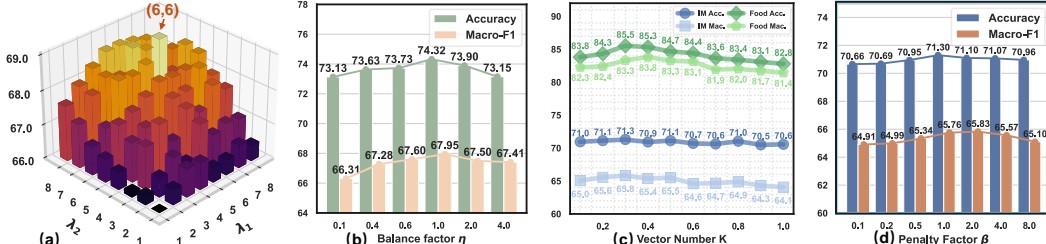

Figure 4: **The ablation studies** of (a) $\lambda_1$ and $\lambda_2$ evaluated on ImageNet; (b) balance factor $\eta$ evaluated on ImageNet; (c) hyper-class vector (expert) number $b$ in RSs evaluated on ImageNet and Food101; (d) Penalty factor $\beta$ in BEM evaluated on ImageNet and Food101.

-1.78% in macro-F1, indicating both DPs and EPs in the SyPs contribute to the performance. Also, RS is instrumental in the further improvement, without which we observe a considerable degradation. Furthermore, SyP+RS+BEM always achieves superior performance to other settings, which shows BEM is an eligible advancement of TTA under Long-tailed settings. In summary, all of the components function synergistically and contribute to the overall performance of L-TTA.

**The balance factor $\eta$.** We ablate the balance factor $\eta$ on ImageNet by varying its value in Figure 4.b. We find that combining our class re-allocation loss results in a performance enhancement of about 1.19%/1.64% in accuracy/macro-F1 compared with solely employing the entropy-minimization loss (i.e, $\eta = 0$). However, excessively large $\eta$ also leads to the performance degradation. We speculate that this potentially stems from an over-emphasis on cluster homogeneity, which introduces optimization challenges and consequently hinders adaptation with limited data-stream. Our experiment results demonstrate that setting $\eta = 1$ yields the best performance.

**Vector number $K$ in RS.** We alter $K$ from 0.1 to 1 on ImageNet and Food101 and report the results in Figure 4.c. Generally, we find that excessively small choices lead to performance degradation, which may result from the insufficient modeling of clusters in the dataset; In addition, we also observe a gradual degradation when $K$ keeps increasing, which may result from exacerbated training challenges. Our experiment results show that setting $K = 0.2$ yields the best performance.

Table 6: **Ablation studies** on components.

| Methods | RN50 | | VIT-B/16 | |
|---|---|---|---|---|
| | Acc. | Mac. | Acc. | Mac. |
| DP | 57.58 | 51.66 | 68.68 | 63.40 |
| DP+RS | 58.10 | 52.24 | 69.76 | 64.12 |
| EP | 56.94 | 51.07 | 67.54 | 62.20 |
| EP+RS | 57.43 | 51.86 | 68.03 | 62.77 |
| SyP(DP+EP)+RS | 59.59 | 53.04 | 70.94 | 65.17 |
| SyP+RS+BEM | **59.82** | **53.67** | **71.30** | **65.83** |

**Penalty factor $\beta$.** We vary $\beta$ from 0.1 to 8 and report the results in Figure 4.d. We find that both higher ($\geq 2$) and lower ($\leq 0.5$) choices generally result in the performance degradation; With appropriate parameter selection, our L-TTA outperforms both the approach leaning on class priors ($\beta$=8) and the method relying solely on original logits ($\beta$=0.1) by up to 0.64%/0.85% in accuracy/macro-F1. This validates the effectiveness of the additional penalty term we introduced in BEM. We find that setting $\beta = 1$ yields the best performance.

**Robustness to dynamic head/tail-class shifts.** This experiment investigates the model's robustness to dynamically changing head-tail classes. We vary the sampling probability $\epsilon$ for tail-class samples, where a larger

Table 7: **Ablation studies** on head/tail class shifts.

| Dataset | ImageNet | | | | Flowers | | | |
|---|---|---|---|---|---|---|---|---|
| $\epsilon$ | 0 | 1/3 | 2/3 | 0 | 1/3 | 2/3 | 1 |
| Acc. | 71.30 | 71.34 | 71.47 | 71.52 | 74.60 | 74.98 | 74.95 | 75.08 |
| Mac. | 65.72 | 65.76 | 65.86 | 65.93 | 68.62 | 68.85 | 68.98 | 69.01 |

$\epsilon$ increases their likelihood of appearing earlier in the stream. Results in Table 7 demonstrate L-TTA maintains stable performance across $\epsilon$, indicating the strong resilience of our proposed model.

## 5 CONCLUSION

This paper aims to extend Test-Time Adaptation (TTA) to harsher and realistic long-tailed scenarios. We propose our novel L-TTA, which consists of synergistic prototypes for enriching tail classes, rebalancing shortcuts to register knowledge and adapt models, and balanced entropy minimization as the first variant of EM targeting on long-tailed TTA tasks. L-TTA achieves superior performance over 15 datasets under various long-tailed ratios, while showing impressive efficiency.

## ACKNOWLEDGEMENT

The authors gratefully acknowledge the support from the National Natural Science Foundation of China (NSFC) under Grant Nos. 62402472, and 12227901. This work was also supported by the Natural Science Foundation of Jiangsu Province (No. BK20240461), the Project of Stable Support for Youth Team in Basic Research Field, CAS (No. YSBR-005), and the Academic Leaders Cultivation Program at USTC. The AI-driven experiments, simulations and model training were performed on the robotic AI-Scientist platform of Chinese Academy of Sciences.

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

# Appendix
# Long-tailed Test-Time Adaptation for Vision-Language Models

The Supplementary Material of the paper is organized as:

## A  PROOFS

### A.1  PROOF OF PROPOSITION 1

**Proposition 3** *For the long-tailed TTA, denote $\mathbf{z} = \{z_1, z_2, z_3.., z_C\}$ as the output logits, in which we assume $|\mathcal{C}_1| > |\mathcal{C}_2| > |\mathcal{C}_3|.. > |\mathcal{C}_C|$. $|\mathcal{C}_i|$ is the class cardinality of $z_i$. We split $\mathcal{C}$ into $\mathcal{C}_{\text{head}}$ and $\mathcal{C}_{\text{tail}}$ with certain measurements. Denote $\mathbb{H}(\cdot)$ as EM loss. The following holds true ($\delta a/\delta b$ equals $\nabla_b a$ in the proposition of the main paper):*

$$\mathbb{E}_{\mathcal{C}_{\text{head}}} \frac{\delta\mathbb{H}}{\delta z_i} < \mathbb{E}_{\mathcal{C}_{\text{tail}}} \frac{\delta\mathbb{H}}{\delta z_i} \tag{12}$$

**Proof.** We can separate the overall expectation into the top-1 expectation term and the non-top-1 expectation term. With the law of total expectation, we deduce:

$$\begin{aligned}
\mathbb{E}_{\mathcal{C}_{\text{head}}} \frac{\delta\mathbb{H}}{\delta z_i} &= \mathbb{E}_{\mathcal{C}_{\text{head}}}\left(\frac{\delta\mathbb{H}}{\delta z_i}\Big|\mathcal{M}_{\text{top1}}\right)\mathcal{P}_{\text{head}}(\mathcal{M}_{\text{top1}}) \\
&\quad + \mathbb{E}_{\mathcal{C}_{\text{head}}}\left(\frac{\delta\mathbb{H}}{\delta z_i}\Big|\mathcal{M}_{\text{non}}\right)\mathcal{P}_{\text{head}}(\mathcal{M}_{\text{non}}) \\
\mathbb{E}_{\mathcal{C}_{\text{tail}}} \frac{\delta\mathbb{H}}{\delta z_i} &= \mathbb{E}_{\mathcal{C}_{\text{tail}}}\left(\frac{\delta\mathbb{H}}{\delta z_i}\Big|\mathcal{M}_{\text{top1}}\right)\mathcal{P}_{\text{tail}}(\mathcal{M}_{\text{top1}}) \\
&\quad + \mathbb{E}_{\mathcal{C}_{\text{tail}}}\left(\frac{\delta\mathbb{H}}{\delta z_i}\Big|\mathcal{M}_{\text{non}}\right)\mathcal{P}_{\text{tail}}(\mathcal{M}_{\text{non}})
\end{aligned} \tag{13}$$

Where $\mathcal{P}(\mathcal{M}_{\text{top1/non}})$ represents the statistical probability of being top-1 / non-top1 of the entries. Under the long-tailed scenario, the following assumption holds true: $\mathcal{P}_{\text{head}}(\mathcal{M}_{\text{top1}}) > \mathcal{P}_{\text{tail}}(\mathcal{M}_{\text{top1}}), \mathcal{P}_{\text{head}}(\mathcal{M}_{\text{non}}) < \mathcal{P}_{\text{tail}}(\mathcal{M}_{\text{non}})$.

Next, we separately discover the gradient term for the top-1 logit entry, $s_1$, and non-top-1 logit entries. For the gradient of $\mathbb{H}$ with respect to $s_1$, we can deduce the specific gradient expressions as

follows, similar to (Wu et al., 2025):

$$\frac{\delta \mathbb{H}}{\delta z_i} = \frac{\delta}{\delta z_i}(-\sum_{i=1}^{L} p_l \log p_l)$$

$$= -\sum_{l=1}^{L}(\frac{\delta p_l}{\delta z_i} \log p_l + \delta p_i) \tag{14}$$

$$= p_i \log p_i + p_i - \sum_{l=1}^{L}(-p_l p_i \log p_l - p_l p_i)$$

We finally derive:

$$\mathbb{E}_{\mathcal{C}}(\frac{\delta \mathbb{H}}{\delta s_1}|\mathcal{M}_{\text{top1}}) = \mathbb{E}_{\mathcal{C}}(-\sum_{l=1}^{L} p_l p_1 \log \frac{p_1}{p_l}) < 0 \tag{15}$$

The last inequality can be easily deduced with the assumption that $s_1$ is the largest entry in the logits, since the last log term is always greater than 0.

Then consider non-top-1 logit entries. We ignore the specific correlations between different classes and treat them as identical elements. To facilitate the discussion, we redirect our focus to their summation $s_{\text{non}}$, $s_{\text{non}} = \sum_{i=2}^{L} z_i$. We make the following deduction:

$$\sum_{i=1}^{L} \frac{\delta \mathbb{H}}{\delta z_i} = \sum_{i=1}^{L}(-\sum_{l=1}^{L} p_l p_i \log \frac{p_i}{p_l}) = 0 \tag{16}$$

This shows that the gradient summation with respect to different entries is 0. Then, we can derive that:

$$\frac{\delta \mathbb{H}}{\delta s_{\text{rest}}} = \sum_{i=2}^{L} \frac{\delta \mathbb{H}/\delta z_i}{\delta s_{\text{rest}}/\delta z_i} = \sum_{i=1}^{L} \frac{\delta \mathbb{H}}{\delta z_i} - \frac{\delta \mathbb{H}}{\delta s_1} > 0 \tag{17}$$

Then we have:

$$\mathbb{E}_{\mathcal{C}}(\frac{\delta \mathbb{H}}{\delta z_i}|\mathcal{M}_{\text{non}}) = \mathbb{E}_{\mathcal{C}} \frac{\delta \mathbb{H}}{\delta z_i} = \mathbb{E}_{\mathcal{C}}(\frac{\delta \mathbb{H}/\delta s_{\text{rest}}}{\delta z_i/\delta s_{\text{rest}}}) > 0 \tag{18}$$

Consequently:

$$\mathbb{E}_{\mathcal{C}}(\frac{\delta \mathbb{H}}{\delta s_1}|\mathcal{M}_{\text{top1}}) < 0 < \mathbb{E}_{\mathcal{C}}(\frac{\delta \mathbb{H}}{\delta z_i}|\mathcal{M}_{\text{non}}) \tag{19}$$

This means that the gradient of entropy with respect to the top 1 entry is theoretically smaller than that with respect to a non-top 1 entry. Since head classes play as the top-1 entry more frequently, we can easily deduce that proposition 1 holds true.

## A.2 PROOF OF PROPOSITION 2

**Proposition 4** *For the long-tailed TTA, denote* $\mathbf{z} = \{z_1, z_2, z_3.., z_C\}$ *as the output logits, in which we assume* $|\mathcal{C}_1| > |\mathcal{C}_2| > |\mathcal{C}_3|.. > |\mathcal{C}_C|$. $|\mathcal{C}_i|$ *is the class cardinality of* $z_i$. *We split* $\mathcal{C}$ *into* $\mathcal{C}_{\text{head}}$ *and* $\mathcal{C}_{\text{tail}}$ *with certain measurements. Denote* $\mathbb{H}(\cdot)$ *as EM loss. The following holds true:*

$$|\mathbb{E}_{\mathcal{C}_{\text{head}}} \frac{\delta \mathbb{H}}{\delta z_i} - \mathbb{E}_{\mathcal{C}_{\text{tail}}} \frac{\delta \mathbb{H}}{\delta z_i}| > |\mathbb{E}_{\mathcal{C}_{\text{head}}} \frac{\delta \mathbb{H}'}{\delta z_i} - \mathbb{E}_{\mathcal{C}_{\text{tail}}} \frac{\delta \mathbb{H}'}{\delta z_i}| \tag{20}$$

**Proof.** Similarly, we have:

$$\mathbb{E}_{\mathcal{C}_{\text{head}}} \frac{\delta \mathbb{H}}{\delta z_i} = \mathbb{E}_{\mathcal{C}_{\text{head}}}(\frac{\delta \mathbb{H}}{\delta z_i}|\mathcal{M}_{\text{top1}})\mathcal{P}_{\text{head}}(\mathcal{M}_{\text{top1}}) + \mathbb{E}_{\mathcal{C}_{\text{head}}}(\frac{\delta \mathbb{H}}{\delta z_i}|\mathcal{M}_{\text{non}})\mathcal{P}_{\text{head}}(\mathcal{M}_{\text{non}})$$

$$\mathbb{E}_{\mathcal{C}_{\text{tail}}} \frac{\delta \mathbb{H}}{\delta z_i} = \mathbb{E}_{\mathcal{C}_{\text{tail}}}(\frac{\delta \mathbb{H}}{\delta z_i}|\mathcal{M}_{\text{top1}})\mathcal{P}_{\text{tail}}(\mathcal{M}_{\text{top1}}) + \mathbb{E}_{\mathcal{C}_{\text{tail}}}(\frac{\delta \mathbb{H}}{\delta z_i}|\mathcal{M}_{\text{non}})\mathcal{P}_{\text{tail}}(\mathcal{M}_{\text{non}}) \tag{21}$$

$$\mathbb{E}_{\mathcal{C}_{\text{head}}} \frac{\delta \mathbb{H}'}{\delta z_i} = \mathbb{E}_{\mathcal{C}_{\text{head}}}(\frac{\delta \mathbb{H}'}{\delta z_i}|\mathcal{M}_{\text{top1}})\mathcal{P}_{\text{head}}(\mathcal{M}_{\text{top1}}) + \mathbb{E}_{\mathcal{C}_{\text{head}}}(\frac{\delta \mathbb{H}'}{\delta z_i}|\mathcal{M}_{\text{non}})\mathcal{P}_{\text{head}}(\mathcal{M}_{\text{non}})$$

$$\mathbb{E}_{\mathcal{C}_{\text{tail}}} \frac{\delta \mathbb{H}'}{\delta z_i} = \mathbb{E}_{\mathcal{C}_{\text{tail}}}(\frac{\delta \mathbb{H}'}{\delta z_i}|\mathcal{M}_{\text{top1}})\mathcal{P}_{\text{tail}}(\mathcal{M}_{\text{top1}}) + \mathbb{E}_{\mathcal{C}_{\text{tail}}}(\frac{\delta \mathbb{H}'}{\delta z_i}|\mathcal{M}_{\text{non}})\mathcal{P}_{\text{tail}}(\mathcal{M}_{\text{non}}) \tag{22}$$

Returning to our BEM, it incorporates an additional prior term into the logits. Since BEM primarily exerts a stronger influence on uncertain categories, we could assert that EM and BEM yield the same behavior in confident (top-1) classes. Focusing on non-top1 classes, head classes receive greater negative gradient gains than tail classes under equivalent conditions. That is:

$$0 > \mathbb{E}_{\mathcal{C}_{\texttt{tail}}}(\frac{\delta\mathbb{H}'}{\delta z_m}|\mathcal{M}_{\texttt{non}}) - \mathbb{E}_{\mathcal{C}_{\texttt{tail}}}(\frac{\delta\mathbb{H}}{\delta z_m}|\mathcal{M}_{\texttt{non}}) > \mathbb{E}_{\mathcal{C}_{\texttt{head}}}(\frac{\delta\mathbb{H}'}{\delta z_m}|\mathcal{M}_{\texttt{non}}) - \mathbb{E}_{\mathcal{C}_{\texttt{head}}}(\frac{\delta\mathbb{H}}{\delta z_m}|\mathcal{M}_{\texttt{non}}) \quad (23)$$

Since the following two formula always hold true:

$$\mathbb{E}_{\mathcal{C}_{\texttt{tail}}}(\frac{\delta\mathbb{H}'}{\delta z_i}|\mathcal{M}_{\texttt{top1}}) > 0 > \mathbb{E}_{\mathcal{C}_{\texttt{tail}}}(\frac{\delta\mathbb{H}'}{\delta z_i}|\mathcal{M}_{\texttt{non}}), \ \ \mathbb{E}_{\mathcal{C}_{\texttt{head}}}\frac{\delta\mathbb{H}}{\delta z_i} - \mathbb{E}_{\mathcal{C}_{\texttt{tail}}}\frac{\delta\mathbb{H}}{\delta z_i} > 0 \quad (24)$$

We can derive the following formula:

$$\mathbb{E}_{\mathcal{C}_{\texttt{head}}}(\frac{\delta\mathbb{H}}{\delta z_m}) - \mathbb{E}_{\mathcal{C}_{\texttt{head}}}(\frac{\delta\mathbb{H}'}{\delta z_m}) \approx \mathcal{P}_{\texttt{head}}(\mathcal{M}_{\texttt{non}})(\mathbb{E}_{\mathcal{C}_{\texttt{tail}}}(\frac{\delta\mathbb{H}}{\delta z_m}|\mathcal{M}_{\texttt{non}}) - \mathbb{E}_{\mathcal{C}_{\texttt{head}}}(\frac{\delta\mathbb{H}'}{\delta z_m}|\mathcal{M}_{\texttt{non}})) > 0$$
$$\mathbb{E}_{\mathcal{C}_{\texttt{tail}}}(\frac{\delta\mathbb{H}}{\delta z_m}) - \mathbb{E}_{\mathcal{C}_{\texttt{tail}}}(\frac{\delta\mathbb{H}'}{\delta z_m}) \approx \mathcal{P}_{\texttt{tail}}(\mathcal{M}_{\texttt{non}})(\mathbb{E}_{\mathcal{C}_{\texttt{tail}}}(\frac{\delta\mathbb{H}}{\delta z_m}|\mathcal{M}_{\texttt{non}}) - \mathbb{E}_{\mathcal{C}_{\texttt{tail}}}(\frac{\delta\mathbb{H}'}{\delta z_m}|\mathcal{M}_{\texttt{non}})) > 0 \quad (25)$$

According to Equation 23, we can easily derive the following formula:

$$\mathbb{E}_{\mathcal{C}_{\texttt{head}}}(\frac{\delta\mathbb{H}}{\delta z_m}) - \mathbb{E}_{\mathcal{C}_{\texttt{head}}}(\frac{\delta\mathbb{H}'}{\delta z_m}) > \mathbb{E}_{\mathcal{C}_{\texttt{tail}}}(\frac{\delta\mathbb{H}}{\delta z_m}) - \mathbb{E}_{\mathcal{C}_{\texttt{tail}}}(\frac{\delta\mathbb{H}'}{\delta z_m}) > 0 \quad (26)$$

$$|\mathbb{E}_{\mathcal{C}_{\texttt{head}}}\frac{\delta\mathbb{H}}{\delta z_i} - \mathbb{E}_{\mathcal{C}_{\texttt{tail}}}\frac{\delta\mathbb{H}}{\delta z_i}| > |\mathbb{E}_{\mathcal{C}_{\texttt{head}}}\frac{\delta\mathbb{H}'}{\delta z_i} - \mathbb{E}_{\mathcal{C}_{\texttt{tail}}}\frac{\delta\mathbb{H}'}{\delta z_i}| \quad (27)$$

Here we proved Proposition 2.

## B    INTRODUCTION OF DATASETS AND METHODS

**Datasets:** We list the detailed statistics and introductions of datasets in Table 46; **Methods:** We list the introductions of methods used in our experiments as follows:

- TPT (Shu et al., 2022): TPT proposes to crop various augmented views from the original image, and select the most confident predictions to conduct entropy minimization.

- C-TPT (Yoon et al., 2024): C-TPT calibrates the uncertainty of model predictions by enforcing texture representations of different prompts to be closer to their centroids.

- MTA (Zanella & Ben Ayed, 2024): MTA is free of optimizing prompts. Instead, it searches the mode for different views and optimizes their inlierness. MTA can be adapted to API-based models.

- TDA (Karmanov et al., 2024): TDA proposes a positive cache to store discriminative class knowledge and a negative cache to store the noisy, spurious, or OOD semantics for different classes. TDA is training-free.

- DPE (Zhang et al., 2024a): DPE introduces dual prototype evolving to adapt at each step. It's free of back-propagation over vision and text encoders.

- O-TPT (Sharifdeen et al., 2025): O-TPT further enhances C-TPT by learning orthogonal text representations to obtain textual dispersion.

- SCAP (Zhang et al., 2025a): SCAP introduces supportive cliques to learn various attribute prompts for each class, accompanied by a graph retention technique to accumulate learned knowledge from previous batches. In the inference stage, it utilizes both the preserved attributes and the learned prompts for robust prediction.

- WATT (Osowiechi et al., 2024): WATT leverages diverse text prompt templates to generate pseudo labels for model updates, then employs weight averaging (either parallel or sequential) to consolidate learned information. The text embedding are ensembled during evaluation to enhance generalization across domain shifts, achieving strong performance even with episodic images without additional transformations or trainable modules.

- CLIPArTT (Hakim et al., 2025): CLIPArTT introduces a lightweight TTA method that generates instance-specific text prompts from top-K predictions and computes pseudo-labels via image-text similarity matrices. During training, it fine-tunes normalization layers using cross-entropy loss to enhance robustness across domain shifts.

- ZERO (Farina et al., 2024): ZERO is an ultra-efficient training-free TTA. It retains confident views and perform marginalization to get the final prediction without backpropagation, outperforming SOTA methods while being 10× faster and more memory-friendly.

- RLCF (Zhao et al., 2023b): RLCF is a universal TTA method that uses CLIP as a reward model to provide feedback. Concretely, it first samples task-specific candidates via beam search, then optimizes model parameters via REINFORCE algorithm with a baseline-adjusted CLIPScore reward to enhance zero-shot generalization across classification, retrieval, and captioning tasks.

## C  EXPERIMENT RESULTS ON CLIP-VIT-B/16

The full experiment results of Cross-domain Benchmark built upon **CLIP-ViT-B/16** backbone under the imbalanced ratio of $\{10, 20\}, \{50\}$ in shown in Table 11,12 respectively. The head / tail accuracy of the OOD benchmark is shown in Table 16. The head / tail accuracy of Cross-domain Benchmark is shown in Table 18, 19. The full experiment results of Corruption Benchmark in shown in Table 20, 21. As shown in the tables, L-TTA mostly outperforms existing methods, demonstrating its superiority.

## D  EXPERIMENT RESULTS ON CLIP-RESNET50

The full experiment results of Cross-domain Benchmark built upon **CLIP-ResNet50** backbone under the imbalanced ratio of $\{10, 20\}, \{50\}$ in shown in Table 13, 14 respectively. The full experiment results of OOD Benchmark built upon **CLIP-ResNet50** backbone under the imbalanced ratio of $\{10, 20, 50\}$ are shown in Table 15. As shown in the tables, L-TTA mostly outperforms existing methods, demonstrating its superiority.

## E  PROMPTS

We list the prompts used in experiments in Table 17.

## F  MORE ABLATION STUDIES

▶ **Number of Augmented Views.** we alter the number of augmented views in Figure 8.b (For representational convenience, we plot with the total count comprising both augmented views and the original image. For example, 4 corresponds to three augmented views and one original image). We find that the optimum lies in about 8; Excessively large or small choices, both leads to performance degradation, especially in macro-F1. This finding contradicts with the intuition that more augmented views lead to better performance from previous studies (Wu et al., 2025), for which we speculate that: although a larger number of views can more distinctly reflect various inter-class relationships, it also leads to an increase in OOD samples, thereby contaminating the DPs. Simultaneously, greater cropping uncertainty reduces the distinction between EPs, consequently diminishing their ability to indicate "the semantics least likely to occur in this category". Our experiments show that 16 (15 augmented views) is the optimal choice.

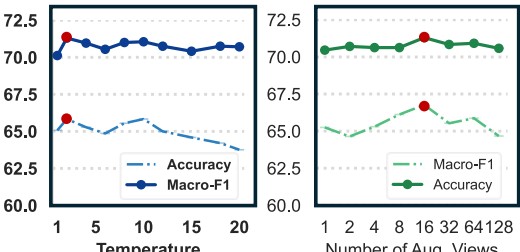

Table 8: **The ablation studies** of (a) temperature evaluated on ImageNet; (b) number of augmented views evaluated on ImageNet.

▶ **Temperature.** We alter the temperature from 1 to 20 in Figure 8.a. Interestingly, we find that excessively large and small temperature leads to degraded performance in macro-F1; for accuracy, we did not observe a clear trend. This phenomenon appears quite anomalous, contradicting the conventional wisdom that temperature adjustment can balance the learning of head and tail classes. However, we argue that: this occurs because temperature's functionality in the EM Loss lacks the clarity than when it in Cross-Entropy (CE) Loss. For CE loss, changing temperature (i.e., reweighting class contributions) explicitly balances the per-class contribution owing to the one-dimensional nature of logit gradient. In contrast, EM's gradient with respect to logits is high-dimensional, restricting the optimal temperature to a narrow range where deviations in either direction theoretically impair model efficacy. Our experiments show that choosing 2 yields the optimal performance.

## G  CAN CLASSIC LT METHODS BE WELL APPLIED TO LONG-TAILED TTA?

While this paper designs various techniques for TTA under long-tailed settings, we would like to also discover the performance of some classic methods, like balanced softmax and logit adjustment to this task. We first briefly introduce these methods as the following:

**Logit Adjustment (LA) (Menon et al., 2020):** LA is a classical LT method that directly adds the class prior, i.e, its proportion of data volume across all classes, to the logits. LA can be applied in both the training stage, as well as the inference stage (i.e, post-hoc adaptation). For both stages, LA can be formulated as:

Table 9: Comparison with classic LT methods.

| Methods | RN50 | | VIT-B/16 | |
|---|---|---|---|---|
| | Acc. | Mac. | Acc. | Mac. |
| MTA1 | 57.30 | 51.23 | 67.79 | 62.57 |
| MTA+BS | 57.31 | 51.28 | 67.87 | 62.33 |
| MTA+LA | 57.40 | 51.30 | 67.90 | 62.68 |
| MTA+BEM | **57.84** | **51.91** | **68.25** | **63.34** |
| DPE | 57.96 | 52.16 | 70.16 | 64.19 |
| DPE+BS | 57.92 | 52.18 | 70.19 | 64.23 |
| DPE+LA | 58.00 | 52.25 | 70.24 | 64.35 |
| DPE+BEM | **58.45** | **52.76** | **70.57** | **64.80** |
| Ours-BEM+BS | 59.50 | 52.76 | 70.75 | 65.12 |
| Ours-BEM+LA | 59.57 | 53.14 | 71.01 | 65.25 |
| Ours | **59.82** | **53.67** | **71.30** | **65.83** |

$$\mathbb{P} = \sigma(\boldsymbol{z}), \ \ \boldsymbol{z} = \boldsymbol{z} - \tau \cdot \log(\boldsymbol{\pi}), \boldsymbol{\pi} = \frac{|\mathcal{C}|}{\sum_i |\mathcal{C}_i|} \tag{28}$$

Where $\tau$ is the temperature. Notably, the original paper only discuss LA's effects with supervised cross-entropy loss. It indicates that LA enlarges the prediction gap between dominant classes and rare ones so that the model would better distinguish them.

**Balanced Softmax (BS) (Ren et al., 2020):** BS proposes to directly combine the class cardinality into softmax to achieve the head-tail balancing. It can be formulated as:

$$\mathbb{P}_t = \frac{|\mathcal{C}_t| \exp(\boldsymbol{z}_t)}{\sum_j |\mathcal{C}_j| \exp(\boldsymbol{z}_j)} \tag{29}$$

The equation 29 is nearly similar with equation 28, where the only difference lies in the signal of the appended term and the temperature. In our experiments, We optimized the temperature to its optimal value.

So, does these techniques really perform well on our long-tailed TTA tasks? we report the ablation results in Table 9 with RN50 and VIT-B/16 as the backbone respectively. We find that the performance gains from incorporating either LA or BS were quite limited. For instance, with MTA, using LA showed almost no improvement, while BS provided only modest gains of approximately 0.10% in accuracy and 0.08% in Macro-F1, RN50. a enhancement within the standard variations. This validates our intuitive explanations in the main paper that merely adding class prior is not adequate for TTA tasks. In contrast, our BEM yields considerable performance improvement: For MTA, we observe a improvement of about 0.64% and 0.68% WITH RN50, 0.46% / 0.77% with VIT-B/16. This also reflect that our BEM is widely applicable and shows great compatibility.

**Some uni-modal LT-TTA methods.** We note that some uni-modal TTA methods have indeed begun addressing long-tail challenges. For instance, SAR (Niu et al., 2022)'s framework includes label shift among its three practical TTA scenarios. SAR then employs batch-agnostic norms, noisy samples filtering and sharpness-aware entropy minimization to deal with it; DELTA (Zhao et al.,

2023a) introduces Test-time Batch Renormalization (TBR) to refine normalization statistics, and Dynamic Online re-weighTing (DOT) to mitigates class bias in long-tailed TTA. However, these methods unanimously fails on the VLM-specific failure modes as introduced in our Figure 1.

## H THE USE OF LARGE LANGUAGE MODELS (LLMS)

LLMs are only for polishing the writing of this paper.

## I REPRODUCIBILITY

The code is available.

## J EXPERIMENTS ON LONG-TAILED CORRUPTION BENCHMARK

We conduct experiments on other corruption types besides gaussian blur, which are first defined by ImageNet-C: *gaussian_noise, shot_noise, impulse_noise, speckle_noise, defocus_blur, glass_blur, motion_blur, zoom_blur, snow, frost, fog, brightness, contrast, jpeg_compression, pixelate, saturate, elastic_transform*. The results are shown in Table 24, 25, 26, 27, 28, 29, 30, 31, 32, 33, 34, 35, 36, 37, 38, 39. The averaged results are shown in Table 23.

As evidenced by these results, L-TTA's performance advantage not only persists but becomes more pronounced under corruptions. Furthermore, we have three key observations:(a): Training-free methods generally struggle on this benchmark. For instance, MTA fails on 12 of 16 corruption types, and the advantages of TDA and ZERO are substantially diminished. These methods rely on the assumption that features form reliable, consistent clusters—an assumption violated by corruptions that fragment distributions and disrupt cross-modal alignment. (b): The gains of unimodal, training-required methods (e.g., SCAP) degrade, converging to the level of classic TPT. This stems from their lack of genuine cross-modal co-adaptation, causing them to amplify modality misalignments under corruption. (c) L-TTA delivers extensive and consistent improvements. This is because that: instead of relying on distributional assumptions, L-TTA leverages instance-specific cues (SyPs and BEM) and employs proactive modality-shared learning (RSs) to continuously counteract corruption-induced biases within and across modalities.

## K EXPERIMENT RESULTS ON BALANCED DATASETS

The experiment results on balanced datasets are in Table 45. The results show that L-TTA can also outperforms existing methods on balanced datasets.

## L EXPERIMENTS ON LARGER BACKBONES

The experiment results on larger backbones for TPT, TDA, DPE, SCAP and L-TTA are shown in Table 40, 41, 42, 43, 44. The performance improvement of our model is consistent across different backbone architectures, with only minor fluctuations observed. This is because L-TTA effectively leverages sample-level information and actively promotes continuous image-text alignment, which are generally beneficial regardless of the underlying backbone in long-tailed test-time adaptation.

## M TUNING EFFICIENCY

Our method involves tuning several additional parameters for different datasets, but this process is more of an optional refinement choice rather than a necessary step that critically affects model behavior. We explain this from two perspectives: First, as shown in Figure 4, varying $\eta$, $K$, and $\beta$ within a broad range ($\times 0.5$–$2$) results in only minor performance changes of about 0.2%. Furthermore,

Table 10: Ablation studies on Per-dataset Hyper-parameter FineTuning.

| Method | LT-CDB | | LT-OODB | |
|---|---|---|---|---|
| | Acc. | Mac. | Acc. | Mac. |
| w/o PHFT | 68.52 | 63.12 | 64.88 | 59.98 |
| L-TTA | 68.77 | 63.44 | 65.19 | 60.49 |

Table 11: **Results on Long-tailed Cross-domain Benchmark under the imbalance ratio of 10 (upper one) and 20 (lower one)**. We conduct 5 runs for each experiment under each imbalance setting and report the Accuracy (Acc.) and F1-Macro (Mac.). All the methods are built upon **CLIP-ViT-B/16** backbone. The best results are in **Bold**.

| Methods | Caltech | | Pets | | Cars | | Flowers | | Food101 | |
| --- | --- | --- | --- | --- | --- | --- | --- | --- | --- | --- |
| | Acc. | Mac. | Acc. | Mac. | Acc. | Mac. | Acc. | Mac. | Acc. | Mac. |
| CLIP [ICML'21] | 93.17 | 90.47 | 84.68 | 85.23 | 62.58 | 62.21 | 65.93 | 62.69 | 82.77 | 81.76 |
| TPT [NeurIPS'22] | 93.97 | 91.61 | 84.50 | 84.32 | 63.97 | 63.37 | 67.40 | 63.02 | 83.89 | 82.92 |
| C-TPT [ICLR'24] | 93.47 | 91.19 | 86.17 | 86.49 | 63.75 | 63.33 | 68.44 | 64.54 | 82.79 | 81.59 |
| MTA [CVPR'24] | 93.18 | 91.54 | 85.08 | 86.03 | 64.52 | 64.09 | 65.96 | 62.73 | 83.99 | **83.11** |
| TDA [CVPR'24] | 94.06 | 90.96 | 87.71 | 86.01 | 68.17 | 63.06 | 71.70 | 64.76 | 86.23 | **83.82** |
| DPE [NeurIPS'24] | 94.91 | **92.56** | 91.18 | 89.33 | 68.45 | 64.74 | 74.36 | 68.70 | 84.87 | 82.74 |
| O-TPT [CVPR'25] | 93.51 | 91.08 | 85.96 | 86.40 | 63.40 | 63.08 | 68.37 | 64.53 | 82.72 | 81.47 |
| SCAP [CVPR'25] | 91.52 | 86.99 | 86.26 | 80.21 | 65.70 | 63.81 | 70.50 | 65.01 | 83.09 | 82.94 |
| L-TTA (Ours) | **95.12** | 92.46 | **91.62** | **90.22** | **70.33** | **65.54** | **74.91** | **68.99** | **85.53** | 83.33 |

| Methods | Aircraft | | SUN397 | | DTD | | EuroSAT | | UCF101 | |
| --- | --- | --- | --- | --- | --- | --- | --- | --- | --- | --- |
| | Acc. | Mac. | Acc. | Mac. | Acc. | Mac. | Acc. | Mac. | Acc. | Mac. |
| CLIP [ICML'21] | 16.56 | 18.05 | 60.62 | 57.93 | 43.52 | 36.91 | 37.74 | 31.39 | 64.78 | 60.67 |
| TPT [NeurIPS'22] | 16.75 | 17.71 | 64.13 | 60.48 | 44.30 | 38.03 | 37.69 | 31.76 | 66.75 | 62.86 |
| C-TPT [ICLR'24] | 19.17 | 19.58 | 63.67 | 60.40 | 44.61 | 39.35 | 37.72 | 32.27 | 66.93 | 61.75 |
| MTA [CVPR'24] | 20.14 | 20.07 | 62.89 | 60.21 | 43.99 | 38.01 | 39.52 | 32.79 | 67.47 | 63.43 |
| TDA [CVPR'24] | 20.83 | 19.95 | 66.59 | 62.60 | 46.30 | 36.86 | 57.02 | 48.25 | **71.09** | 64.26 |
| DPE [NeurIPS'24] | 25.23 | 22.20 | 68.85 | 65.10 | 48.36 | 43.21 | 56.77 | 49.47 | 70.19 | 64.81 |
| O-TPT [CVPR'25] | 18.78 | 18.68 | 64.26 | 60.70 | 45.08 | 38.87 | 38.37 | 32.75 | 66.09 | 61.14 |
| SCAP [CVPR'25] | 22.92 | 19.09 | 63.73 | 61.29 | 44.84 | 37.70 | 52.48 | 45.53 | 68.54 | **66.71** |
| L-TTA (Ours) | **27.02** | **24.14** | **69.79** | **65.44** | **51.63** | **47.50** | **57.07** | **48.67** | 70.77 | 64.86 |

| Methods | Caltech | | Pets | | Cars | | Flowers | | Food101 | |
| --- | --- | --- | --- | --- | --- | --- | --- | --- | --- | --- |
| | Acc. | Mac. | Acc. | Mac. | Acc. | Mac. | Acc. | Mac. | Acc. | Mac. |
| CLIP [ICML'21] | 92.98 | 90.37 | 83.80 | 83.25 | 62.10 | 60.78 | 66.18 | 62.37 | 82.34 | 79.01 |
| TPT [NeurIPS'22] | 93.50 | 91.10 | 83.71 | 84.06 | 63.37 | 61.93 | 67.17 | 62.40 | 83.46 | 81.47 |
| C-TPT [ICLR'24] | 93.19 | 90.88 | 85.50 | 86.09 | 63.37 | 61.81 | 67.62 | 63.36 | 82.30 | 79.89 |
| MTA [CVPR'24] | 93.58 | 91.47 | 84.05 | 84.72 | 64.43 | **63.64** | 64.41 | 60.93 | 83.59 | 81.56 |
| TDA [CVPR'24] | 94.14 | 90.97 | 87.06 | 84.06 | 67.77 | 60.61 | 72.80 | 63.88 | **85.96** | **82.00** |
| DPE [NeurIPS'24] | 94.67 | 91.92 | 90.04 | 87.20 | 68.46 | 61.72 | 74.17 | 66.67 | 83.90 | 79.70 |
| O-TPT [CVPR'25] | 93.15 | 90.85 | 85.35 | 85.90 | 62.80 | 61.33 | 67.70 | 63.37 | 82.32 | 79.87 |
| SCAP [CVPR'25] | 91.32 | 86.92 | 86.47 | 78.73 | 63.90 | 61.61 | 68.27 | 62.14 | 83.07 | 82.91 |
| L-TTA (Ours) | **95.28** | **92.94** | **91.91** | **87.30** | **70.44** | 63.46 | **74.89** | **67.22** | 85.65 | 80.53 |

| Methods | Aircraft | | SUN397 | | DTD | | EuroSAT | | UCF101 | |
| --- | --- | --- | --- | --- | --- | --- | --- | --- | --- | --- |
| | Acc. | Mac. | Acc. | Mac. | Acc. | Mac. | Acc. | Mac. | Acc. | Mac. |
| CLIP [ICML'21] | 15.48 | 15.81 | 60.18 | 54.92 | 43.37 | 36.94 | 37.22 | 30.12 | 64.72 | 58.34 |
| TPT [NeurIPS'22] | 15.55 | 17.17 | 64.04 | 57.91 | 43.76 | 37.91 | 37.12 | 30.63 | 65.82 | 59.16 |
| C-TPT [ICLR'24] | 17.68 | 19.07 | 63.07 | 57.25 | 43.37 | 37.78 | 37.56 | 30.22 | 67.09 | 59.43 |
| MTA [CVPR'24] | 18.16 | 18.85 | 62.57 | 57.17 | 43.76 | 38.01 | 40.31 | 31.31 | 67.23 | 61.84 |
| TDA [CVPR'24] | 25.38 | 20.26 | 65.40 | 58.79 | 44.72 | 33.66 | 55.34 | 53.73 | 70.39 | 61.49 |
| DPE [NeurIPS'24] | 25.77 | 21.35 | 67.89 | 61.91 | **47.79** | 40.51 | 56.07 | 47.04 | 70.32 | 62.45 |
| O-TPT [CVPR'25] | 17.48 | 18.75 | 62.86 | 56.99 | 43.95 | **37.96** | 38.05 | 31.21 | 66.82 | 59.35 |
| SCAP [CVPR'25] | 22.83 | 19.00 | 63.74 | 61.33 | 41.61 | 35.70 | 44.64 | 35.72 | 68.09 | 62.70 |
| L-TTA (Ours) | **26.67** | **21.85** | **68.48** | **62.40** | 47.02 | 37.43 | **57.72** | **47.41** | **71.37** | **62.82** |

we report an additional experiment in Table 10, which compares the results using identical hyperparameters (tuned on ImageNet in the above) and parameters tuned for each dataset. The results show that L-TTA remains consistently SOTA under this fixed configuration, with performance fluctuations within only 0.3%.

Table 12: **Results on Long-tailed Cross-domain Benchmark under the imbalance ratio of 50**. We conduct 5 runs for each experiment under each imbalance setting and report the Accuracy (Acc.) and F1-Macro (Mac.). All the methods are built upon **CLIP-ViT-B/16** backbone. The best results are marked in **Bold**.

| Methods | Caltech Acc. | Mac. | Pets Acc. | Mac. | Cars Acc. | Mac. | Flowers Acc. | Mac. | Food101 Acc. | Mac. |
|---|---|---|---|---|---|---|---|---|---|---|
| CLIP [ICML'21] | 93.15 | 89.01 | 82.05 | 81.40 | 60.80 | 57.28 | 64.91 | 56.94 | 81.92 | 73.94 |
| TPT [NeurIPS'22] | 93.99 | 90.27 | 82.31 | 80.91 | 62.19 | 58.94 | 66.25 | 57.48 | 83.11 | 75.16 |
| C-TPT [ICLR'24] | 93.57 | 90.00 | 84.55 | 82.65 | 62.63 | 59.58 | 66.92 | 58.35 | 82.13 | 73.66 |
| MTA [CVPR'24] | 93.68 | 89.90 | 83.95 | 82.25 | 62.39 | 59.49 | 65.58 | 57.06 | 83.26 | 75.84 |
| TDA [CVPR'24] | 94.79 | 90.14 | 83.72 | 80.34 | 68.50 | 55.13 | 70.82 | 57.76 | **85.64** | 76.21 |
| DPE [NeurIPS'24] | 94.97 | 90.93 | 89.06 | 81.01 | **69.74** | 56.97 | 73.32 | 61.26 | 83.57 | 73.59 |
| O-TPT [CVPR'25] | 93.57 | 89.90 | 84.43 | 82.74 | 61.84 | 59.08 | 66.81 | 58.65 | 82.13 | 73.72 |
| SCAP [CVPR'24] | 91.36 | 86.97 | 85.71 | 72.84 | 63.89 | **61.62** | 65.54 | 60.13 | 83.05 | 82.90 |
| L-TTA (Ours) | **95.69** | **91.47** | **89.68** | **84.70** | 69.52 | 60.45 | **73.04** | **62.84** | 84.30 | **76.97** |

| Methods | Aircraft Acc. | Mac. | SUN397 Acc. | Mac. | DTD Acc. | Mac. | EuroSAT Acc. | Mac. | UCF101 Acc. | Mac. |
|---|---|---|---|---|---|---|---|---|---|---|
| CLIP [ICML'21] | 11.12 | 16.47 | 59.71 | 49.90 | 41.78 | 33.26 | 36.91 | 28.99 | 66.26 | 55.54 |
| TPT [NeurIPS'22] | 11.90 | 16.38 | 63.75 | 52.86 | 42.33 | 33.90 | 36.56 | 27.77 | 67.56 | 58.10 |
| C-TPT [ICLR'24] | 15.40 | 17.64 | 62.91 | 52.29 | 42.89 | 34.72 | 38.01 | 26.10 | 68.36 | 56.13 |
| MTA [CVPR'24] | 15.12 | 16.88 | 61.91 | 52.40 | 42.61 | 34.80 | 41.52 | 28.74 | 68.76 | 58.91 |
| TDA [CVPR'24] | **22.91** | 16.71 | 65.44 | 54.64 | 43.17 | 33.08 | 48.07 | 45.19 | 71.92 | 58.01 |
| DPE [NeurIPS'24] | 21.95 | **20.60** | 68.03 | 56.52 | 46.51 | 35.75 | 52.78 | 41.03 | 68.07 | 53.87 |
| O-TPT [CVPR'25] | 15.54 | 17.82 | 62.25 | 50.96 | 42.89 | 34.93 | 38.60 | 28.11 | 68.06 | 55.22 |
| SCAP [CVPR'24] | 22.80 | 18.94 | 62.97 | **60.80** | 41.67 | 35.71 | 43.43 | 35.41 | 62.31 | 56.91 |
| L-TTA (Ours) | 22.79 | 19.64 | **68.56** | 59.76 | **46.84** | **37.21** | **54.79** | **45.30** | **69.61** | **59.58** |

Table 13: **Results on Long-tailed Cross-domain Benchmark under the imbalance ratio of 10 (upper one) and 20 (lower one)**. We conduct 5 runs for each experiment under each imbalance setting and report the Accuracy (Acc.) and F1-Macro (Mac.). All the methods are built upon **CLIP-ResNet50** backbone. The best results are marked in **Bold**. SCAP (Zhang et al., 2025a) is not included here, because there's no implementation to other backbones rather than CLIP-VIT-B/16.

| Methods | Caltech Acc. | Mac. | Pets Acc. | Mac. | Cars Acc. | Mac. | Flowers Acc. | Mac. | Food101 Acc. | Mac. |
|---|---|---|---|---|---|---|---|---|---|---|
| TPT [NeurIPS'22] | 86.73 | 83.31 | 82.07 | 81.69 | 56.81 | 55.49 | 60.79 | 56.17 | 74.27 | 72.67 |
| C-TPT [ICLR'24] | 86.56 | 82.74 | 82.41 | 82.15 | 54.92 | 53.94 | 62.26 | 58.02 | 74.51 | 72.84 |
| MTA [CVPR'24] | 85.64 | 82.68 | 82.20 | 82.35 | 57.09 | 55.67 | 60.48 | 56.21 | 74.23 | 72.70 |
| TDA [CVPR'24] | 89.43 | 85.35 | 86.74 | 84.73 | 57.22 | 52.34 | 67.94 | 60.28 | 77.75 | 74.54 |
| DPE [NeurIPS'24] | 89.71 | 85.82 | 86.32 | 84.31 | 60.10 | 55.28 | 68.28 | 62.47 | 75.96 | 73.45 |
| O-TPT [CVPR'25] | 86.77 | 83.03 | 82.20 | 81.88 | 54.54 | 53.60 | 62.44 | 58.09 | 74.56 | 72.94 |
| L-TTA (Ours) | **90.89** | **86.83** | **87.35** | **85.08** | **60.98** | **55.91** | **69.50** | **62.66** | **78.35** | **75.34** |

| Methods | Aircraft Acc. | Mac. | SUN397 Acc. | Mac. | DTD Acc. | Mac. | EuroSAT Acc. | Mac. | UCF101 Acc. | Mac. |
|---|---|---|---|---|---|---|---|---|---|---|
| TPT [NeurIPS'22] | 12.39 | 14.22 | 60.45 | 56.46 | 39.15 | 33.33 | 17.58 | 15.90 | 61.37 | 54.09 |
| C-TPT [ICLR'24] | 13.09 | 14.74 | 60.71 | 56.80 | 39.15 | 32.74 | 19.86 | 15.69 | 61.59 | 54.74 |
| MTA [CVPR'24] | 12.86 | 14.35 | 59.71 | 56.07 | 39.62 | 33.93 | 20.24 | 16.54 | 61.92 | 55.16 |
| TDA [CVPR'24] | 15.42 | 13.39 | 61.66 | 56.52 | 45.24 | 37.55 | 50.00 | **45.14** | 66.49 | 57.36 |
| DPE [NeurIPS'24] | 17.83 | 15.76 | 62.89 | **59.25** | 49.29 | 44.02 | 49.34 | 38.19 | 62.89 | 56.06 |
| O-TPT [CVPR'25] | 13.17 | 14.81 | 60.44 | 56.46 | 39.15 | 32.79 | 30.78 | 28.05 | 61.75 | 55.04 |
| L-TTA (Ours) | **19.85** | **18.31** | **63.45** | 59.12 | **50.95** | **45.66** | **50.57** | 45.10 | **67.06** | **58.10** |

| Methods | Caltech Acc. | Mac. | Pets Acc. | Mac. | Cars Acc. | Mac. | Flowers Acc. | Mac. | Food101 Acc. | Mac. |
|---|---|---|---|---|---|---|---|---|---|---|
| TPT [NeurIPS'22] | 86.22 | 82.83 | 81.07 | 79.61 | 55.46 | 52.73 | 59.85 | 54.32 | 73.02 | 69.72 |
| C-TPT [ICLR'24] | 86.43 | 82.69 | 81.84 | 81.33 | 54.11 | 52.54 | 63.61 | 58.75 | 73.77 | 70.61 |
| MTA [CVPR'24] | 85.35 | 82.65 | 81.58 | 80.37 | 55.73 | 53.15 | 61.04 | 55.57 | 73.65 | 70.78 |
| TDA [CVPR'24] | 88.34 | 84.17 | 85.27 | 80.63 | 56.49 | 49.20 | 68.33 | 58.93 | 77.30 | **72.34** |
| DPE [NeurIPS'24] | 90.39 | 86.25 | 87.31 | 84.58 | 59.50 | 52.39 | 66.99 | 60.41 | 75.43 | 70.23 |
| O-TPT [CVPR'25] | 86.45 | 82.63 | 81.67 | 80.57 | 53.52 | 51.63 | 61.91 | 56.89 | 73.74 | 70.56 |
| L-TTA (Ours) | **90.52** | **87.42** | **87.61** | **85.87** | **60.58** | **53.10** | **69.04** | **60.67** | **78.29** | 71.61 |

| Methods | Aircraft Acc. | Mac. | SUN397 Acc. | Mac. | DTD Acc. | Mac. | EuroSAT Acc. | Mac. | UCF101 Acc. | Mac. |
|---|---|---|---|---|---|---|---|---|---|---|
| TPT [NeurIPS'22] | 11.11 | 13.23 | 60.34 | 53.74 | 38.92 | 31.22 | 16.92 | 15.39 | 61.49 | 52.89 |
| C-TPT [ICLR'24] | 10.33 | 12.64 | 60.52 | 54.26 | 39.53 | 31.60 | 17.86 | 16.66 | 62.01 | 53.31 |
| MTA [CVPR'24] | 11.11 | 13.09 | 59.11 | 52.98 | 39.30 | 33.46 | 18.68 | 16.37 | 63.16 | 54.19 |
| TDA [CVPR'24] | 15.34 | 13.13 | 61.06 | 53.73 | 42.61 | 32.67 | 39.00 | **37.77** | 66.86 | **55.10** |
| DPE [NeurIPS'24] | 15.25 | 13.58 | 63.12 | 56.85 | 45.68 | **40.75** | 45.99 | 35.90 | 62.67 | 52.50 |
| O-TPT [CVPR'25] | 10.62 | 12.66 | 60.27 | 54.28 | 40.11 | 32.09 | 29.11 | 26.46 | 61.94 | 53.11 |
| L-TTA (Ours) | **19.40** | **15.00** | **63.94** | **56.06** | **48.56** | 40.54 | **50.08** | 35.91 | **67.45** | 55.02 |

Table 14: **Results on Long-tailed Cross-domain Benchmark under the imbalance ratio of 50**. We conduct 5 runs for each experiment under each imbalance setting and report the Accuracy (Acc.) and F1-Macro (Mac.). All the methods are built upon **CLIP-ResNet50** backbone. The best results are marked in **Bold**. SCAP (Zhang et al., 2025a) is not included here, because there're no implementation to other backbones rather than CLIP-VIT-B/16.

| Methods | Caltech | | Pets | | Cars | | Flowers | | Food101 | |
|---|---|---|---|---|---|---|---|---|---|---|
| | Acc. | Mac. | Acc. | Mac. | Acc. | Mac. | Acc. | Mac. | Acc. | Mac. |
| TPT [NeurIPS'22] | 85.63 | 79.80 | 80.32 | 75.45 | 55.09 | 50.23 | 60.78 | 52.72 | 72.97 | 63.26 |
| C-TPT [ICLR'24] | 85.99 | 80.02 | 81.07 | 75.80 | 52.59 | 48.58 | 60.45 | 52.82 | 72.25 | 63.29 |
| MTA [CVPR'24] | 84.15 | 79.27 | 79.57 | 75.34 | 54.54 | **49.46** | 61.32 | 51.48 | 73.23 | 63.82 |
| TDA [CVPR'24] | 87.17 | 82.48 | 83.97 | 76.63 | 57.29 | 45.11 | 67.73 | 53.15 | 76.85 | **64.63** |
| DPE [NeurIPS'24] | 89.90 | 82.91 | 85.09 | 78.65 | 58.42 | 45.03 | 66.32 | 54.41 | 74.68 | 63.48 |
| O-TPT [CVPR'25] | 85.84 | 79.81 | 80.44 | 74.93 | 52.20 | 47.80 | 63.67 | 54.11 | 73.20 | 63.23 |
| L-TTA (Ours) | **91.30** | **84.43** | **85.21** | **79.30** | **59.44** | 49.35 | **69.84** | **55.84** | **77.28** | 63.95 |

| Methods | Aircraft | | SUN397 | | DTD | | EuroSAT | | UCF101 | |
|---|---|---|---|---|---|---|---|---|---|---|
| | Acc. | Mac. | Acc. | Mac. | Acc. | Mac. | Acc. | Mac. | Acc. | Mac. |
| TPT [NeurIPS'22] | 8.40 | 13.59 | 60.66 | 48.88 | 38.99 | 29.12 | 13.00 | 13.47 | 63.47 | 49.20 |
| C-TPT [ICLR'24] | 7.98 | 11.37 | 60.46 | 49.96 | 40.11 | 29.93 | 13.76 | 14.64 | 64.27 | 51.39 |
| MTA [CVPR'24] | 8.54 | 12.02 | 58.75 | 47.89 | 38.71 | 29.71 | 14.39 | 15.88 | 64.17 | 51.62 |
| TDA [CVPR'24] | 14.26 | 11.16 | 60.29 | 49.01 | 41.78 | 25.78 | 31.03 | 30.82 | 69.23 | 53.17 |
| DPE [NeurIPS'24] | 17.76 | 13.66 | 61.77 | 50.94 | 42.06 | 29.13 | 49.55 | **32.97** | 66.53 | 49.12 |
| O-TPT [CVPR'25] | 8.26 | 11.63 | 60.49 | 49.94 | 40.66 | 29.59 | 24.04 | 23.96 | 64.47 | 51.73 |
| L-TTA (Ours) | **18.48** | **15.88** | **62.70** | **51.73** | **42.86** | **30.69** | **51.34** | 31.86 | **69.59** | **54.26** |

Table 15: **Results on Long-tailed OOD Benchmark**. We conduct 5 runs for each experiment and report the accuracy (Acc.) and macro-F1 (Mac.). All the methods are built upon **CLIP-RN50**. The "OOD Average" is calculated by averaging four OOD datasets excluding ImageNet. The best results are marked in **Bold**. **SCAP (Zhang et al., 2025a) is not included because of the above reason.**

| Methods | ImageNet-A | | ImageNet-R | | ImageNet-S | | ImageNet-V2 | | ImageNet | | OOD average | |
|---|---|---|---|---|---|---|---|---|---|---|---|---|
| | Acc | Mac. | Acc | Mac. | Acc | Mac. | Acc | Mac. | Acc | Mac. | Acc | Mac. |
| | | | | | imb=10 | | | | | | | |
| TPT [NeurIPS'22] | 25.00 | 23.17 | 59.72 | 57.79 | 32.59 | 31.81 | 55.72 | 54.28 | 56.57 | 50.50 | 43.26 | 41.76 |
| C-TPT [ICLR'24] | 23.54 | 21.86 | 58.13 | 56.53 | 32.01 | 31.35 | 55.75 | 52.00 | 56.44 | 50.62 | 42.36 | 40.44 |
| MTA [CVPR'24] | 26.39 | 25.10 | 58.05 | 56.30 | 34.22 | 35.78 | 55.67 | 54.50 | 57.30 | 51.23 | 44.04 | 42.92 |
| TDA [CVPR'24] | 30.62 | 27.95 | 63.00 | 60.08 | 35.58 | 33.31 | 56.68 | 50.53 | 58.13 | 52.88 | 46.47 | 42.97 |
| DPE [NeurIPS'24] | 30.38 | 28.21 | 63.70 | 61.03 | 36.87 | 35.53 | 58.39 | 53.48 | 57.96 | 52.16 | 47.34 | 44.56 |
| O-TPT [CVPR'25] | 22.75 | 21.30 | 57.55 | 55.84 | 33.55 | 30.69 | 55.04 | 52.47 | 56.81 | 50.77 | 42.22 | 40.08 |
| L-TTA (Ours) | **32.96** | **30.57** | **64.88** | **61.92** | **37.13** | **35.93** | **59.07** | **54.98** | **59.82** | **53.67** | **48.51** | **45.85** |
| | | | | | imb=20 | | | | | | | |
| TPT [NeurIPS'22] | 24.63 | 22.51 | 59.57 | 55.51 | 31.96 | 30.55 | 55.83 | 54.69 | 56.06 | 50.81 | 43.00 | 40.82 |
| C-TPT [ICLR'24] | 22.96 | 21.45 | 57.94 | 54.15 | 31.36 | 30.11 | 55.79 | 52.06 | 54.37 | 50.27 | 42.01 | 39.44 |
| MTA [CVPR'24] | 27.53 | 25.38 | 58.79 | 57.83 | 35.35 | 34.04 | 54.22 | 53.05 | 57.94 | 51.02 | 43.97 | 42.58 |
| TDA [CVPR'24] | 30.83 | 27.88 | 63.32 | 58.08 | 34.93 | 32.13 | 56.58 | 51.16 | 58.39 | 52.65 | 46.42 | 42.31 |
| DPE [NeurIPS'24] | 31.31 | 28.75 | 63.27 | 58.96 | 35.07 | 33.84 | 57.83 | 50.81 | 59.02 | 53.51 | 46.87 | 43.09 |
| O-TPT [CVPR'25] | 22.40 | 20.70 | 57.47 | 53.81 | 31.33 | 32.98 | 56.06 | 52.29 | 55.76 | 51.14 | 41.82 | 39.95 |
| L-TTA (Ours) | **31.53** | **29.20** | **66.47** | **60.27** | **38.47** | **35.29** | **58.05** | **54.88** | **59.25** | **53.17** | **48.63** | **44.91** |
| | | | | | imb=50 | | | | | | | |
| TPT [NeurIPS'22] | 24.28 | 19.74 | 60.08 | 49.66 | 31.19 | 27.53 | 55.03 | 53.00 | 55.71 | 50.82 | 42.65 | 37.48 |
| C-TPT [ICLR'24] | 22.09 | 18.07 | 58.37 | 48.30 | 32.31 | 26.84 | 56.03 | 51.75 | 53.93 | 49.75 | 42.20 | 36.24 |
| MTA [CVPR'24] | 27.63 | 25.38 | 58.79 | 57.83 | 35.35 | **34.04** | 54.22 | 53.05 | 55.25 | 50.59 | 44.00 | 42.58 |
| TDA [CVPR'24] | 30.20 | 23.71 | 63.19 | 50.54 | 34.10 | 30.10 | 56.83 | 49.00 | 56.14 | 52.86 | 46.08 | 38.34 |
| DPE [NeurIPS'24] | 30.76 | 24.92 | 64.81 | 52.10 | 35.05 | 31.07 | 56.42 | 47.15 | 57.22 | 53.62 | 46.76 | 38.81 |
| O-TPT [CVPR'25] | 21.38 | 17.72 | 57.88 | 47.87 | 30.83 | 26.81 | 54.95 | 53.84 | 56.02 | 50.00 | 41.26 | 36.56 |
| L-TTA (Ours) | **30.64** | **25.46** | **65.58** | **58.42** | **36.11** | 33.93 | **57.73** | **54.80** | **59.31** | **53.10** | **47.52** | **43.40** |

Table 16: **Head / Tail accuracy on Long-tailed OOD Benchmark**. We conduct 5 runs for each experiment and report the accuracy (Acc.) and macro-F1 (Mac.). All the methods are built upon **CLIP-VIT-B/16**. The "OOD Average" is calculated by averaging four OOD datasets excluding ImageNet. The best results are marked in **Bold**.

| imb=10 | ImageNet-A | | ImageNet-R | | ImageNet-S | | ImageNet-V2 | | ImageNet | | Average | |
|---|---|---|---|---|---|---|---|---|---|---|---|---|
| | Head | Tail | Head | Tail | Head | Tail | Head | Tail | Head | Tail | Head | Tail |
| TPT [NeurIPS'22] | 53.27 | 50.71 | 78.91 | 72.27 | 48.04 | 36.52 | 63.11 | 55.36 | 66.61 | 63.50 | 61.99 | 55.67 |
| C-TPT [ICLR'24] | 52.17 | 47.78 | 77.59 | 73.70 | 48.34 | 38.25 | 62.12 | 56.41 | 66.10 | 64.87 | 61.26 | 56.20 |
| MTA [CVPR'24] | 61.07 | 53.73 | 80.38 | 75.22 | 50.34 | 43.17 | 64.74 | 59.20 | 68.08 | 66.19 | 64.92 | 59.50 |
| TDA [CVPR'24] | 64.47 | 57.31 | 85.58 | 78.31 | 51.00 | 41.28 | 68.45 | 64.49 | 70.45 | 64.92 | 67.99 | 61.26 |
| DPE [NeurIPS'24] | 64.31 | 56.79 | 85.13 | 78.22 | 52.35 | 42.45 | 69.71 | 63.61 | 71.36 | 66.87 | 68.57 | 61.59 |
| O-TPT [CVPR'25] | 54.67 | 50.64 | 78.15 | 72.22 | 48.66 | 35.10 | 63.19 | 56.42 | 67.14 | 66.53 | 62.36 | 56.18 |
| SCAP [CVPR'25] | 62.46 | 55.72 | 83.72 | 72.45 | 51.86 | 38.54 | 68.95 | 63.43 | 65.30 | 65.84 | 66.46 | 59.20 |
| L-TTA (Ours) | **65.30** | **56.90** | **86.00** | **79.28** | **53.51** | **44.00** | **70.02** | **65.30** | **72.04** | **68.81** | **69.37** | **62.86** |

| imb=20 | ImageNet-A | | ImageNet-R | | ImageNet-S | | ImageNet-V2 | | ImageNet | | Average | |
|---|---|---|---|---|---|---|---|---|---|---|---|---|
| | Head | Tail | Head | Tail | Head | Tail | Head | Tail | Head | Tail | Head | Tail |
| TPT [NeurIPS'22] | 52.74 | 47.92 | 77.58 | 74.01 | 48.17 | 38.49 | 61.13 | 55.47 | 66.39 | 62.80 | 61.20 | 55.74 |
| C-TPT [ICLR'24] | 52.03 | 46.09 | 76.52 | 73.75 | 48.21 | 37.54 | 63.69 | 55.92 | 62.35 | 61.16 | 60.56 | 54.89 |
| MTA [CVPR'24] | 61.07 | 53.73 | 80.38 | 74.22 | 49.09 | 43.87 | 63.12 | 58.89 | 68.04 | 62.87 | 64.34 | 58.72 |
| TDA [CVPR'24] | 64.72 | 56.69 | 85.33 | 77.70 | 50.71 | 41.66 | 66.05 | **66.01** | 68.30 | 62.33 | 67.02 | 60.88 |
| DPE [NeurIPS'24] | 63.49 | 56.82 | 85.00 | 77.66 | 52.24 | 42.91 | 68.12 | 64.25 | 70.72 | 65.61 | 67.91 | 61.45 |
| O-TPT [CVPR'25] | 54.28 | 50.42 | 78.16 | 73.93 | 48.49 | 36.95 | 63.54 | 55.07 | 66.09 | 62.55 | 62.11 | 55.78 |
| SCAP [CVPR'25] | 62.33 | 54.31 | 83.60 | 76.51 | 51.66 | 36.50 | 68.55 | 62.17 | 65.28 | 65.10 | 66.28 | 58.92 |
| L-TTA (Ours) | **65.25** | **57.12** | **86.64** | **78.48** | **53.66** | 43.75 | **68.76** | 65.34 | **71.84** | **68.50** | **69.23** | **62.64** |

| imb=50 | ImageNet-A | | ImageNet-R | | ImageNet-S | | ImageNet-V2 | | ImageNet | | Average | |
|---|---|---|---|---|---|---|---|---|---|---|---|---|
| | Head | Tail | Head | Tail | Head | Tail | Head | Tail | Head | Tail | Head | Tail |
| TPT [NeurIPS'22] | 52.47 | 37.25 | 77.49 | 72.77 | 46.89 | 37.06 | 60.49 | 52.59 | 65.90 | 59.07 | 60.65 | 51.75 |
| C-TPT [ICLR'24] | 53.81 | 35.00 | 76.35 | 72.69 | 46.12 | 36.76 | 60.59 | 50.90 | 63.72 | 59.37 | 60.12 | 50.94 |
| MTA [CVPR'24] | 61.07 | 53.73 | 80.37 | 75.22 | 50.39 | 41.17 | 64.74 | 59.15 | 65.96 | 60.41 | 64.51 | 57.94 |
| TDA [CVPR'24] | 64.50 | 56.52 | 85.41 | 78.33 | 50.24 | 42.22 | 68.15 | **63.54** | 66.51 | 63.06 | 66.96 | 60.73 |
| DPE [NeurIPS'24] | 63.16 | 55.50 | 85.01 | 78.83 | 49.54 | **43.36** | **68.89** | 62.75 | 67.03 | 65.97 | 66.73 | 61.28 |
| O-TPT [CVPR'25] | 52.88 | 51.33 | 78.02 | 72.65 | 48.04 | 35.09 | 61.10 | 50.46 | 62.28 | 60.98 | 60.46 | 54.10 |
| SCAP [CVPR'25] | 62.31 | 56.05 | 82.24 | 76.30 | 51.23 | 37.35 | 68.30 | 61.29 | 64.02 | 59.59 | 65.62 | 58.12 |
| L-TTA (Ours) | **64.57** | **57.93** | **86.81** | **78.95** | **51.75** | 42.57 | 68.15 | 63.44 | **67.84** | **67.17** | **67.82** | **62.01** |

Table 17: The list of prompts. Notably, following DPE (Zhang et al., 2024a), we also employ the CuPL (Pratt et al., 2023) methods to further expand prompt pools.

| Datasets | Prompts |
|---|---|
| Caltech101 | "a photo of a {CLASS}." |
| Pets | "a photo of a {CLASS}, a type of food." |
| Cars | "a photo of a {CLASS}." |
| Flowers | "a photo of a {CLASS}, a type of flower." |
| Food101 | "itap of a {CLASS}, a type of food." |
| SUN397 | "a photo of a {CLASS}." |
| DTD | "{CLASS} texture." |
| EuroSAT | "a centered satellie photo of {CLASS}." |
| UCF101 | "a photo of a person doing {CLASS}." |
| Aircraft | "a photo of a {CLASS}, a type of aircraft." |
| ImageNet | "a bad photo of the {CLASS}." |
| ImageNet-V | "a origami of the {CLASS}." |
| ImageNet-S | "a photo of the large {CLASS}." |
| ImageNet-A | "a {CLASS} in a video game." |
| ImageNet-R | "art of the {CLASS}"
"a photo of the small {CLASS}" |

Table 18: **Head / Tail Accuracy on Long-tailed Cross-domain Benchmark under the imbalance ratio of 10 (upper one) and 20 (lower one)**. We conduct 5 runs for each experiment under each imbalance setting. All the methods are built upon **CLIP-ViT-B/16** backbone. The best results are marked in **Bold**.

| Methods | Caltech | | Pets | | Cars | | Flowers | | Food101 | |
|---|---|---|---|---|---|---|---|---|---|---|
| | Head | Tail | Head | Tail | Head | Tail | Head | Tail | Head | Tail |
| TPT [NeurIPS'22] | 95.74 | 89.39 | 83.02 | 81.44 | 64.53 | 58.89 | 69.17 | 64.58 | 85.71 | 77.30 |
| C-TPT [ICLR'24] | 94.91 | 90.61 | 86.66 | 85.27 | 65.59 | 60.34 | 70.84 | 65.36 | 85.53 | 76.88 |
| MTA [CVPR'24] | 95.37 | 91.74 | 87.01 | 82.50 | 67.39 | 61.33 | 67.53 | 63.41 | 86.73 | 79.85 |
| TDA [CVPR'24] | 97.07 | 88.44 | 91.71 | **89.02** | **71.25** | 66.52 | 73.40 | 67.32 | 88.20 | 81.68 |
| DPE [NeurIPS'24] | **96.89** | 91.15 | 94.08 | 86.01 | 70.39 | 66.01 | 77.24 | 70.59 | 86.07 | 81.91 |
| O-TPT [CVPR'25] | 93.74 | 88.79 | 84.23 | 80.86 | 62.78 | 58.47 | 69.97 | 65.10 | 85.06 | 76.43 |
| SCAP [CVPR'25] | 95.03 | 86.26 | 87.39 | 85.38 | 65.19 | 63.43 | 72.93 | 62.31 | 86.82 | 81.88 |
| L-TTA (Ours) | 96.36 | **92.96** | 94.28 | 88.36 | 71.15 | **69.71** | 78.34 | 71.04 | 88.49 | 84.78 |

| Methods | Aircraft | | SUN397 | | DTD | | EuroSAT | | UCF101 | |
|---|---|---|---|---|---|---|---|---|---|---|
| | Head | Tail | Head | Tail | Head | Tail | Head | Tail | Head | Tail |
| TPT [NeurIPS'22] | 19.40 | 10.30 | 65.80 | 59.96 | 45.67 | 40.60 | 40.12 | 31.38 | 70.11 | 55.71 |
| C-TPT [ICLR'24] | 23.17 | 9.29 | 64.03 | 63.09 | 44.56 | 44.32 | 38.97 | 36.77 | 71.30 | 56.85 |
| MTA [CVPR'24] | 26.19 | 11.34 | 63.76 | 62.60 | 46.35 | 41.91 | 39.10 | 39.85 | 70.18 | 64.04 |
| TDA [CVPR'24] | 24.24 | 17.03 | 67.54 | 67.13 | 45.47 | 44.15 | 58.17 | 57.08 | 73.84 | 67.10 |
| DPE [NeurIPS'24] | 26.87 | 22.45 | 69.46 | 67.84 | 49.70 | 45.59 | 57.50 | **57.62** | **72.67** | 66.46 |
| O-TPT [CVPR'25] | 21.21 | 10.04 | 61.32 | 59.07 | 46.56 | 41.07 | 40.35 | 32.01 | 71.09 | 55.80 |
| SCAP [CVPR'25] | 24.53 | 21.45 | 65.13 | 63.19 | 47.22 | 40.20 | 53.94 | 47.77 | 72.47 | 58.76 |
| L-TTA (Ours) | **29.83** | **22.71** | **70.27** | **70.16** | **54.99** | **47.59** | **58.33** | 56.69 | 72.39 | **68.26** |

| Methods | Caltech | | Pets | | Cars | | Flowers | | Food101 | |
|---|---|---|---|---|---|---|---|---|---|---|
| | Head | Tail | Head | Tail | Head | Tail | Head | Tail | Head | Tail |
| TPT [NeurIPS'22] | 95.78 | 87.98 | 85.50 | 79.39 | 64.57 | 59.71 | 69.51 | 61.80 | 85.89 | 76.85 |
| C-TPT [ICLR'24] | 95.29 | 88.10 | 85.94 | 85.01 | 64.47 | 60.23 | 69.67 | 62.96 | 84.31 | 79.06 |
| MTA [CVPR'24] | 95.42 | 89.75 | 85.10 | 83.00 | 64.89 | 62.97 | 65.94 | 63.96 | 85.46 | 80.41 |
| TDA [CVPR'24] | 95.17 | **93.10** | 90.29 | 80.33 | 67.96 | 67.83 | 74.70 | 67.21 | 87.19 | 83.87 |
| DPE [NeurIPS'24] | 96.63 | 91.07 | 93.77 | 83.42 | 69.49 | 68.66 | 77.76 | 66.09 | 85.19 | 81.99 |
| O-TPT [CVPR'25] | 94.75 | 88.06 | 83.91 | 79.91 | 65.17 | 56.43 | 73.32 | 59.12 | 85.60 | 72.48 |
| SCAP [CVPR'25] | 93.03 | 88.18 | 87.25 | 85.94 | 65.30 | 63.48 | 73.12 | 57.20 | 86.90 | 75.91 |
| L-TTA (Ours) | **97.55** | 91.24 | 94.93 | 85.97 | 71.05 | 69.85 | 79.70 | 69.98 | 87.23 | 83.71 |

| Methods | Aircraft | | SUN397 | | DTD | | EuroSAT | | UCF101 | |
|---|---|---|---|---|---|---|---|---|---|---|
| | Head | Tail | Head | Tail | Head | Tail | Head | Tail | Head | Tail |
| TPT [NeurIPS'22] | 19.11 | 7.77 | 64.75 | 62.00 | 45.74 | 39.00 | 39.18 | 32.10 | 70.01 | 56.08 |
| C-TPT [ICLR'24] | 21.00 | 9.25 | 63.90 | 60.19 | 43.95 | 41.86 | 39.26 | 33.83 | 71.49 | 58.15 |
| MTA [CVPR'24] | 22.24 | 9.65 | 62.85 | 62.14 | 45.77 | 40.12 | 40.65 | 36.64 | 70.05 | 60.67 |
| TDA [CVPR'24] | 26.36 | 21.63 | 66.14 | 65.60 | 45.16 | 41.76 | 57.16 | 48.93 | 72.35 | 65.10 |
| DPE [NeurIPS'24] | 27.89 | 21.17 | 68.45 | 67.61 | 47.55 | 45.77 | 56.56 | 54.55 | 72.16 | 65.95 |
| O-TPT [CVPR'25] | 19.92 | 11.38 | 63.27 | 61.07 | 46.95 | 36.63 | 40.92 | 31.62 | 70.01 | 55.06 |
| SCAP [CVPR'25] | 24.26 | 21.37 | 64.20 | 63.13 | 43.60 | 40.35 | 46.32 | 39.49 | 72.69 | 58.69 |
| L-TTA (Ours) | **29.64** | **22.00** | **68.66** | **68.48** | **47.95** | **46.39** | **57.76** | **55.75** | **73.07** | **67.85** |

Table 19: **Head / Tail Accuracy on Long-tailed Cross-domain Benchmark under the imbalance ratio of 50**. We conduct 5 runs for each experiment under each imbalance setting. All the methods are built upon **CLIP-VIT-B/16** backbone. The best results are marked in **Bold**.

| Methods | Caltech | | Pets | | Cars | | Flowers | | Food101 | |
|---|---|---|---|---|---|---|---|---|---|---|
| | Head | Tail | Head | Tail | Head | Tail | Head | Tail | Head | Tail |
| TPT [NeurIPS'22] | 94.55 | 90.16 | 84.16 | 77.95 | 66.05 | 53.65 | 68.94 | 59.86 | 85.78 | 75.02 |
| C-TPT [ICLR'24] | 94.41 | 90.99 | 85.63 | 83.23 | 66.57 | 53.53 | 69.21 | 61.51 | 83.63 | 76.01 |
| MTA [CVPR'24] | 95.20 | 90.59 | 83.97 | 83.54 | 65.26 | 55.88 | 67.19 | 62.10 | 86.33 | 77.49 |
| TDA [CVPR'24] | 96.35 | 91.10 | 86.40 | 78.93 | 69.25 | 68.02 | 75.46 | 60.55 | 85.20 | 79.70 |
| DPE [NeurIPS'24] | 96.80 | 90.78 | 90.25 | 83.64 | 70.51 | 68.47 | 76.56 | 69.32 | 85.67 | 81.09 |
| O-TPT [CVPR'25] | 96.33 | 87.59 | 86.98 | 79.32 | 64.81 | 55.69 | 68.94 | 59.48 | 83.97 | 79.20 |
| SCAP [CVPR'25] | 94.03 | 86.02 | 86.71 | 83.50 | 64.41 | 63.42 | 66.93 | 62.31 | 82.85 | 74.90 |
| L-TTA (Ours) | **97.80** | **91.74** | **91.14** | **86.44** | **71.50** | **69.54** | **77.91** | **70.31** | **87.51** | **82.36** |

| Methods | Aircraft | | SUN397 | | DTD | | EuroSAT | | UCF101 | |
|---|---|---|---|---|---|---|---|---|---|---|
| | Head | Tail | Head | Tail | Head | Tail | Head | Tail | Head | Tail |
| TPT [NeurIPS'22] | 13.85 | 8.13 | 64.78 | 60.05 | 43.45 | 39.69 | 38.13 | 31.46 | 71.14 | 60.80 |
| C-TPT [ICLR'24] | 18.10 | 8.82 | 62.84 | 62.65 | 43.88 | 39.36 | 40.29 | 31.35 | 71.78 | 63.13 |
| MTA [CVPR'24] | 17.63 | 9.73 | 61.86 | 61.38 | 44.09 | 40.29 | 43.15 | 38.43 | 71.95 | 64.14 |
| TDA [CVPR'24] | 25.98 | 15.03 | 66.18 | 63.89 | 45.35 | 39.72 | 50.49 | 41.18 | 72.39 | 64.60 |
| DPE [NeurIPS'24] | 26.47 | 17.09 | 68.27 | 67.53 | 46.76 | 43.18 | 54.00 | 50.44 | 70.19 | 65.66 |
| O-TPT [CVPR'25] | 15.38 | 10.58 | 63.13 | 61.18 | 43.73 | 37.41 | 40.25 | 32.81 | 71.30 | 61.30 |
| SCAP [CVPR'25] | 25.12 | 17.26 | 63.08 | 63.13 | 42.91 | 40.20 | 48.46 | 35.50 | 69.58 | 51.87 |
| L-TTA (Ours) | **26.62** | **20.77** | **68.43** | **68.10** | **48.88** | **43.64** | **57.25** | **52.04** | **72.74** | **67.94** |

Table 20: **Detailed Accuracy on Long-tailed Noise Benchmark under the severity of 0.1 (upper one) and 0.2 (lower one)**. We conduct 5 runs for each experiment under each imbalance setting. All the methods are built upon **CLIP-ViT-B/16** backbone. The best results are marked in **Bold**.

| Methods | Average | | Caltech | | Pets | | Cars | | Flowers | | Food101 | |
|---|---|---|---|---|---|---|---|---|---|---|---|---|
| | Acc. | Mac. | Acc. | Mac. | Acc. | Mac. | Acc. | Mac. | Acc. | Mac. | Acc. | Mac. |
| TPT [NeurIPS'22] | 54.51 | 52.16 | 92.38 | 88.79 | 78.94 | 78.47 | 57.02 | 57.01 | 61.46 | 56.41 | 69.71 | 67.91 |
| C-TPT [ICLR'24] | 53.61 | 51.08 | 91.54 | 87.83 | 77.97 | 77.16 | 55.29 | 55.38 | 61.52 | 57.03 | 66.03 | 64.15 |
| MTA [CVPR'24] | 54.38 | 51.66 | 91.75 | 87.96 | 78.24 | 77.90 | 57.99 | 57.72 | 59.87 | 54.95 | 67.97 | 66.62 |
| TDA [CVPR'24] | 56.27 | 52.26 | 91.46 | 86.91 | 81.40 | 79.61 | 59.43 | 53.83 | 65.11 | 57.79 | 69.21 | 66.53 |
| DPE [NeurIPS'24] | 57.33 | 53.56 | 92.86 | 89.90 | 83.27 | 80.15 | 61.63 | 57.14 | 66.52 | 60.66 | 67.52 | 61.66 |
| O-TPT [CVPR'25] | 53.53 | 51.48 | 91.62 | 87.84 | 78.24 | 77.42 | 54.70 | 54.82 | 61.65 | 57.08 | 66.00 | 64.18 |
| SCAP [CVPR'25] | 55.32 | 51.45 | 91.01 | 87.63 | 82.79 | 76.96 | 56.52 | 55.10 | 59.30 | 54.10 | 70.35 | 63.88 |
| L-TTA (Ours) | **59.69** | **55.92** | **92.91** | **90.51** | **85.44** | **82.03** | **62.76** | **58.40** | **68.13** | **61.14** | **71.50** | **67.98** |

| Methods | Aircraft | | SUN397 | | DTD | | EuroSAT | | UCF101 | | ImageNet | |
|---|---|---|---|---|---|---|---|---|---|---|---|---|
| | Acc. | Mac. | Acc. | Mac. | Acc. | Mac. | Acc. | Mac. | Acc. | Mac. | Head | Tail |
| TPT [NeurIPS'22] | 12.54 | 15.24 | 58.50 | 55.09 | 39.00 | 35.37 | 12.41 | 9.36 | 59.39 | 53.78 | 58.22 | 56.36 |
| C-TPT [ICLR'24] | 14.18 | 15.50 | 58.09 | 54.44 | 37.28 | 33.90 | 12.24 | 9.05 | 58.29 | 53.23 | 57.25 | 54.25 |
| MTA [CVPR'24] | 15.69 | 15.49 | 58.43 | 54.84 | 37.75 | 34.24 | 13.93 | 8.95 | 58.57 | 52.85 | 58.00 | 56.73 |
| TDA [CVPR'24] | 19.78 | 17.00 | 60.48 | 56.51 | 38.06 | 31.11 | 11.92 | 13.92 | 63.68 | 55.50 | 58.46 | 56.19 |
| DPE [NeurIPS'24] | 19.50 | 16.14 | 62.56 | 57.61 | 39.02 | 36.02 | 15.06 | 14.79 | 63.47 | 57.04 | 59.20 | 58.07 |
| O-TPT [CVPR'25] | 14.57 | 16.29 | 57.66 | 54.17 | 37.12 | 33.77 | 12.73 | 9.90 | 58.18 | 53.30 | 56.35 | 57.50 |
| SCAP [CVPR'25] | 19.81 | 18.07 | 57.21 | 55.74 | 36.88 | 35.12 | 14.78 | 7.30 | 62.67 | 57.49 | 57.20 | 54.54 |
| L-TTA (Ours) | **22.48** | **20.51** | **62.99** | **59.02** | **45.86** | **40.15** | **18.10** | **16.68** | **65.77** | **59.15** | **60.64** | **59.55** |

| Methods | Average | | Caltech | | Pets | | Cars | | Flowers | | Food101 | |
|---|---|---|---|---|---|---|---|---|---|---|---|---|
| | Acc. | Mac. | Acc. | Mac. | Acc. | Mac. | Acc. | Mac. | Acc. | Mac. | Acc. | Mac. |
| TPT [NeurIPS'22] | 42.83 | 40.13 | 86.94 | 81.21 | 67.96 | 65.79 | 44.11 | 43.34 | 50.15 | 44.26 | 39.13 | 39.33 |
| C-TPT [ICLR'24] | 41.48 | 38.56 | 85.51 | 79.58 | 66.57 | 63.94 | 42.90 | 42.52 | 47.88 | 42.64 | 35.71 | 35.20 |
| MTA [CVPR'24] | 38.09 | 38.72 | 84.67 | 78.83 | 66.29 | 64.48 | 45.38 | 44.70 | 4.74 | 40.08 | 34.87 | 36.27 |
| TDA [CVPR'24] | 42.92 | 38.21 | 85.89 | 79.90 | 67.24 | 63.57 | 47.45 | 42.01 | 46.83 | 40.52 | 37.12 | 35.98 |
| DPE [NeurIPS'24] | 43.82 | 40.77 | 86.99 | 82.57 | 65.64 | 63.34 | 47.33 | 44.76 | 49.61 | 44.73 | 36.30 | 36.48 |
| O-TPT [CVPR'25] | 41.18 | 38.55 | 85.64 | 79.83 | 67.26 | 64.83 | 42.02 | 41.48 | 47.33 | 42.09 | 35.56 | 35.12 |
| SCAP [CVPR'25] | 42.87 | 39.48 | 84.99 | 77.83 | 67.70 | 62.42 | 46.11 | 42.30 | 49.03 | 41.88 | 37.05 | 38.00 |
| L-TTA (Ours) | **46.66** | **43.36** | **88.07** | **84.25** | **69.60** | **66.80** | **49.29** | **44.51** | **51.14** | **46.73** | **39.38** | **38.64** |

| Methods | Aircraft | | SUN397 | | DTD | | EuroSAT | | UCF101 | | ImageNet | |
|---|---|---|---|---|---|---|---|---|---|---|---|---|
| | Acc. | Mac. | Acc. | Mac. | Acc. | Mac. | Acc. | Mac. | Acc. | Mac. | Acc. | Mac. |
| TPT [NeurIPS'22] | 10.52 | 12.03 | 47.94 | 44.47 | 23.71 | 22.45 | 7.97 | 4.95 | 49.94 | 43.85 | 42.80 | 39.75 |
| C-TPT [ICLR'24] | 8.65 | 8.88 | 46.25 | 42.94 | 24.64 | 22.70 | 9.36 | 4.93 | 47.52 | 41.54 | 41.24 | 39.30 |
| MTA [CVPR'24] | 10.44 | 10.88 | 45.80 | 42.23 | 22.15 | 20.25 | 13.28 | 5.93 | 47.19 | 40.43 | 44.17 | 41.88 |
| TDA [CVPR'24] | 12.53 | 11.77 | 46.25 | 41.87 | 22.40 | 16.31 | 12.08 | 8.07 | 50.34 | 41.04 | 43.01 | 39.28 |
| DPE [NeurIPS'24] | 14.57 | 12.07 | 48.61 | 43.73 | 25.74 | 25.43 | 14.80 | 10.58 | 48.01 | 42.42 | 44.38 | 42.31 |
| O-TPT [CVPR'25] | 8.72 | 9.25 | 45.23 | 41.92 | 24.49 | 24.35 | 7.62 | 4.67 | 47.14 | 41.17 | 42.00 | 39.35 |
| SCAP [CVPR'25] | 9.60 | 11.84 | 45.55 | 43.07 | 25.79 | 23.63 | 15.68 | 11.09 | 47.61 | 41.75 | 42.48 | 40.52 |
| L-TTA (Ours) | **16.40** | **15.60** | **50.37** | **45.55** | **31.51** | **30.12** | **17.49** | **14.18** | **52.72** | **46.38** | **47.26** | **44.15** |

Table 21: **Detailed Accuracy on Long-tailed Noise Benchmark under the severity ratio of 0.4**. We conduct 5 runs for each experiment under each imbalance setting. All the methods are built upon **CLIP-ViT-B/16** backbone. The best results are marked in **Bold**.

| Methods | Average | | Caltech | | Pets | | Cars | | Flowers | | Food101 | |
|---|---|---|---|---|---|---|---|---|---|---|---|---|
| | Acc. | Mac. | Acc. | Mac. | Acc. | Mac. | Acc. | Mac. | Acc. | Mac. | Acc. | Mac. |
| TPT [NeurIPS'22] | 21.54 | 18.75 | 66.30 | 57.13 | 34.67 | 31.35 | 17.09 | 15.74 | 21.22 | 17.41 | 6.19 | 6.85 |
| C-TPT [ICLR'24] | 19.74 | 17.37 | 62.95 | 53.42 | 29.39 | 27.74 | 16.01 | 14.57 | 19.38 | 15.92 | 5.56 | 5.47 |
| MTA [CVPR'24] | 20.17 | 16.55 | 63.29 | 53.80 | 27.51 | 25.51 | 15.15 | 14.56 | 16.88 | 12.89 | 4.12 | 4.29 |
| TDA [CVPR'24] | 20.44 | 16.02 | 62.27 | 54.15 | 26.30 | 20.17 | 17.03 | 14.07 | 16.59 | 13.25 | 7.06 | 6.29 |
| DPE [NeurIPS'24] | 20.18 | 17.66 | 64.06 | 59.02 | 25.46 | 21.23 | 16.37 | 14.32 | 18.58 | 16.43 | 4.05 | 5.11 |
| O-TPT [CVPR'25] | 18.83 | 16.81 | 63.12 | 53.78 | 30.09 | 28.06 | 15.42 | 14.08 | 18.89 | 15.43 | 5.59 | 5.41 |
| SCAP [CVPR'25] | 19.63 | 17.05 | 63.17 | 52.79 | 28.58 | 25.66 | 17.72 | 14.91 | 16.31 | 11.81 | 6.05 | 6.71 |
| L-TTA (Ours) | **23.60** | **20.65** | **66.36** | **60.28** | **30.89** | **26.76** | **18.39** | **15.48** | **20.76** | **20.28** | **7.76** | **7.15** |

| Methods | Aircraft | | SUN397 | | DTD | | EuroSAT | | UCF101 | | ImageNet | |
|---|---|---|---|---|---|---|---|---|---|---|---|---|
| | Acc. | Mac. | Acc. | Mac. | Acc. | Mac. | Acc. | Mac. | Acc. | Mac. | Acc. | Mac. |
| TPT [NeurIPS'22] | 4.67 | 4.65 | 19.13 | 17.22 | 12.01 | 10.60 | 3.51 | 1.75 | 26.15 | 21.67 | 26.04 | 21.88 |
| C-TPT [ICLR'24] | 3.97 | 4.06 | 17.15 | 15.94 | 11.70 | 10.35 | 3.45 | 2.09 | 23.07 | 20.87 | 24.50 | 20.67 |
| MTA [CVPR'24] | 4.52 | 4.25 | 14.87 | 14.31 | 9.04 | 7.41 | 16.65 | 4.25 | 23.13 | 18.51 | 26.70 | 22.25 |
| TDA [CVPR'24] | 4.36 | 2.67 | 16.56 | 14.53 | 12.79 | 8.58 | 15.18 | 5.16 | 21.86 | 16.45 | 24.88 | 20.94 |
| DPE [NeurIPS'24] | 5.99 | 4.21 | 19.37 | 18.40 | 7.80 | 7.72 | 10.88 | 4.46 | 22.44 | 21.33 | 26.99 | 22.07 |
| O-TPT [CVPR'25] | 3.89 | 3.84 | 15.77 | 14.34 | 10.92 | 9.96 | 3.32 | 3.89 | 15.77 | 15.19 | 24.34 | 20.92 |
| SCAP [CVPR'25] | 4.38 | 5.57 | 17.30 | 16.70 | 10.66 | 10.17 | 5.08 | 4.71 | 20.13 | 17.99 | 26.52 | 20.50 |
| L-TTA (Ours) | **8.07** | **7.59** | **21.68** | **19.15** | **15.44** | **12.01** | **16.67** | **10.95** | **23.13** | **21.84** | **30.42** | **25.61** |

Table 22: **Ablation study on the Per-dataset Hyper-parameter FineTuning (PHFT).**

| Method | Caltech | | Pets | | Cars | | Flowers | | Food101 | |
|---|---|---|---|---|---|---|---|---|---|---|
| | Acc. | Mac. | Acc. | Mac. | Acc. | Mac. | Acc. | Mac. | Acc. | Mac. |
| w/o PHFT | 94.98 | 92.43 | 91.50 | 89.85 | 69.91 | 65.02 | 74.79 | 68.78 | 85.27 | 83.16 |
| L-TTA | 95.12 | 92.46 | 91.62 | 90.22 | 70.33 | 65.54 | 74.91 | 68.99 | 85.53 | 83.33 |
| Method | Aircraft | | SUN397 | | DTD | | EuroSAT | | UCF101 | |
| | Acc. | Mac. | Acc. | Mac. | Acc. | Mac. | Acc. | Mac. | Acc. | Mac. |
| w/o PHFT | 26.63 | 23.31 | 69.52 | 64.40 | 51.39 | 46.63 | 56.82 | 47.82 | 69.56 | 63.58 |
| L-TTA | 27.02 | 24.14 | 69.99 | 64.68 | 51.63 | 47.50 | 57.07 | 48.27 | 70.77 | 63.86 |
| Method | ImageNet | | ImageNet-A | | ImageNet-R | | ImageNet-S | | ImageNet-V2 | |
| | Acc. | Mac. | Acc. | Mac. | Acc. | Mac. | Acc. | Mac. | Acc. | Mac. |
| w/o PHFT | 72.12 | 65.27 | 61.56 | 55.26 | 82.48 | 78.09 | 49.98 | 45.45 | 68.60 | 63.87 |
| L-TTA | 71.30 | 65.83 | 61.78 | 55.97 | 82.86 | 78.56 | 50.25 | 45.99 | 68.99 | 64.19 |

Table 23: **Averaged Accuracy on Long-tailed Corruption Benchmark of 16 corruption types:** *gaussian_noise, shot_noise, impulse_noise, speckle_noise, defocus_blur, glass_blur, motion_blur, zoom_blur, snow, frost, fog, brightness, contrast, jpeg_compression, pixelate, saturate, elastic_transform*. We conduct 5 runs for each experiment under each imbalance setting. All the methods are built upon **CLIP-VIT-B/16** backbone. The best results are marked in **Bold**.

| Method | Caltech | | Pets | | Cars | | Flowers | | Food101 | |
|---|---|---|---|---|---|---|---|---|---|---|
| | Acc. | Mac. | Acc. | Mac. | Acc. | Mac. | Acc. | Mac. | Acc. | Mac. |
| CLIP [ICML'21] | 72.29 | 67.18 | 58.13 | 57.24 | 35.53 | 35.18 | 42.03 | 38.57 | 48.43 | 47.12 |
| TPT [NeurIPS'22] | 73.59 | 68.26 | 60.40 | 59.31 | 37.10 | 36.47 | 43.38 | 39.89 | 50.03 | 48.67 |
| C-TPT [ICLR'24] | 72.04 | 67.71 | 59.48 | 58.94 | 35.80 | 35.25 | 43.26 | 39.83 | 49.13 | 47.69 |
| MTA [CVPR'24] | \ | \ | \ | \ | \ | \ | \ | \ | \ | \ |
| TDA [CVPR'24] | 73.85 | 68.36 | 60.97 | 58.57 | 40.29 | 36.40 | 46.65 | 40.73 | 51.54 | 49.31 |
| ZERO [NeurIPS'24] | 70.29 | 65.25 | 57.17 | 55.22 | 36.34 | 34.81 | 42.46 | 37.70 | 48.35 | 46.15 |
| DPE [NeurIPS'24] | 73.95 | 70.00 | 61.21 | 59.25 | 40.56 | 38.49 | 46.50 | 42.24 | 51.52 | 49.78 |
| O-TPT [CVPR'25] | 72.02 | 67.23 | 58.70 | 58.32 | 35.44 | 34.89 | 42.98 | 39.55 | 48.09 | 46.46 |
| SCAP [CVPR'25] | 72.82 | 68.25 | 60.13 | 60.74 | 37.44 | 36.46 | 44.87 | 41.64 | 50.07 | 48.47 |
| L-TTA (Ours) | 75.08 | 72.51 | 64.56 | 62.99 | 44.76 | 41.11 | 52.27 | 47.66 | 54.23 | 52.56 |
| Method | Aircraft | | SUN397 | | DTD | | EuroSAT | | UCF101 | |
| | Acc. | Mac. | Acc. | Mac. | Acc. | Mac. | Acc. | Mac. | Acc. | Mac. |
| CLIP [ICML'21] | 9.93 | 10.17 | 39.37 | 37.08 | 25.64 | 21.68 | 25.47 | 18.72 | 41.74 | 38.00 |
| TPT [NeurIPS'22] | 11.26 | 11.56 | 41.11 | 38.66 | 27.45 | 23.39 | 27.30 | 21.03 | 43.34 | 39.58 |
| C-TPT [ICLR'24] | 10.54 | 10.02 | 38.83 | 36.10 | 25.90 | 21.95 | 23.09 | 18.01 | 41.39 | 37.35 |
| MTA [CVPR'24] | \ | \ | \ | \ | \ | \ | \ | \ | \ | \ |
| TDA [CVPR'24] | 12.70 | 10.35 | 43.03 | 39.42 | 27.50 | 21.26 | 28.93 | 25.11 | 46.91 | 40.53 |
| ZERO [NeurIPS'24] | 10.48 | 9.59 | 39.44 | 36.74 | 25.60 | 21.32 | 24.40 | 19.43 | 42.98 | 38.45 |
| DPE [NeurIPS'24] | 14.88 | 12.95 | 44.17 | 41.10 | 29.51 | 26.16 | 31.39 | 26.41 | 42.00 | 37.65 |
| O-TPT [CVPR'25] | 10.62 | 10.00 | 38.26 | 35.33 | 25.29 | 20.91 | 22.73 | 17.44 | 41.06 | 36.72 |
| SCAP [CVPR'25] | 11.19 | 11.00 | 41.81 | 39.19 | 28.07 | 23.59 | 29.86 | 24.27 | 45.25 | 40.54 |
| L-TTA (Ours) | 18.07 | 16.31 | 45.82 | 43.13 | 35.38 | 30.85 | 38.80 | 33.36 | 49.37 | 43.97 |

Table 24: **Detailed Accuracy on Long-tailed Corruption Benchmark of *shot noise*.** We conduct 5 runs for each experiment under each imbalance setting. All the methods are built upon **CLIP-VIT-B/16** backbone. The best results are marked in **Bold**.

| Method | Caltech | | Pets | | Cars | | Flowers | | Food101 | |
|---|---|---|---|---|---|---|---|---|---|---|
| | Acc. | Mac. | Acc. | Mac. | Acc. | Mac. | Acc. | Mac. | Acc. | Mac. |
| CLIP [ICML'21] | 91.04 | 86.94 | 77.48 | 76.30 | 55.31 | 55.65 | 57.73 | 53.79 | 66.34 | 64.71 |
| TPT [NeurIPS'22] | 92.72 | 89.65 | 82.21 | 81.92 | 60.35 | 60.45 | 63.98 | 59.06 | 73.52 | 72.09 |
| C-TPT [ICLR'24] | 88.28 | 84.70 | 83.03 | 81.90 | 61.96 | 60.51 | 61.10 | 56.45 | 76.14 | 74.88 |
| MTA [CVPR'24] | \ | \ | \ | \ | \ | \ | \ | \ | \ | \ |
| TDA [CVPR'24] | 91.79 | 87.29 | 81.47 | 78.12 | 61.99 | 56.52 | 64.19 | 57.47 | 70.69 | 67.92 |
| ZERO [NeurIPS'24] | 90.37 | 86.76 | 77.97 | 76.60 | 55.34 | 55.21 | 57.55 | 52.75 | 68.56 | 65.53 |
| DPE [NeurIPS'24] | 92.56 | 90.10 | 83.90 | 81.77 | 63.68 | 59.18 | 66.13 | 60.58 | 70.38 | 68.30 |
| O-TPT [CVPR'25] | 90.37 | 86.54 | 78.74 | 78.54 | 55.45 | 55.37 | 58.17 | 54.63 | 65.24 | 63.65 |
| SCAP [CVPR'25] | 90.78 | 86.96 | 82.09 | 82.42 | 58.65 | 56.54 | 61.62 | 58.53 | 66.79 | 62.73 |
| L-TTA (Ours) | 92.48 | 89.04 | 84.10 | 82.47 | 62.87 | 57.28 | 70.98 | 66.21 | 70.83 | 68.46 |
| Method | Aircraft | | SUN397 | | DTD | | EuroSAT | | UCF101 | |
| | Acc. | Mac. | Acc. | Mac. | Acc. | Mac. | Acc. | Mac. | Acc. | Mac. |
| CLIP [ICML'21] | 12.86 | 14.64 | 56.72 | 53.85 | 34.32 | 30.72 | 16.00 | 13.12 | 56.92 | 51.61 |
| TPT [NeurIPS'22] | 14.19 | 15.60 | 61.59 | 58.01 | 40.41 | 36.74 | 19.95 | 17.26 | 62.86 | 57.87 |
| C-TPT [ICLR'24] | 8.03 | 7.29 | 54.92 | 51.78 | 38.10 | 34.37 | 16.94 | 14.98 | 57.58 | 51.88 |
| MTA [CVPR'24] | \ | \ | \ | \ | \ | \ | \ | \ | \ | \ |
| TDA [CVPR'24] | 20.09 | 16.66 | 61.06 | 56.56 | 37.29 | 29.78 | 23.24 | 20.18 | 65.70 | 56.88 |
| ZERO [NeurIPS'24] | 13.56 | 15.02 | 56.66 | 53.03 | 34.79 | 29.68 | 14.02 | 10.27 | 56.92 | 51.05 |
| DPE [NeurIPS'24] | 20.87 | 19.53 | 64.10 | 60.57 | 43.84 | 40.89 | 23.33 | 20.53 | 11.86 | 14.37 |
| O-TPT [CVPR'25] | 15.28 | 14.97 | 56.06 | 52.07 | 35.57 | 29.02 | 16.58 | 10.51 | 55.27 | 49.53 |
| SCAP [CVPR'25] | 12.15 | 11.18 | 60.12 | 56.04 | 38.43 | 31.99 | 19.51 | 12.77 | 62.81 | 57.03 |
| L-TTA (Ours) | 19.21 | 19.29 | 65.20 | 60.93 | 47.55 | 43.38 | 23.37 | 20.78 | 66.16 | 59.57 |

Table 25: **Detailed Accuracy on Long-tailed Corruption Benchmark of *impulse noise*. We conduct 5 runs for each experiment under each imbalance setting. All the methods are built upon CLIP-VIT-B/16 backbone. The best results are marked in Bold.**

| Method | Caltech | | Pets | | Cars | | Flowers | | Food101 | |
|---|---|---|---|---|---|---|---|---|---|---|
| | Acc. | Mac. | Acc. | Mac. | Acc. | Mac. | Acc. | Mac. | Acc. | Mac. |
| CLIP [ICML'21] | 78.57 | 71.12 | 60.88 | 60.22 | 31.70 | 31.16 | 40.73 | 36.08 | 36.22 | 36.13 |
| TPT [NeurIPS'22] | 81.62 | 69.68 | 75.26 | 71.82 | 32.67 | 31.98 | 49.66 | 44.76 | 40.99 | 40.87 |
| C-TPT [ICLR'24] | 83.15 | 83.77 | 74.57 | 72.71 | 31.85 | 31.74 | 47.76 | 44.98 | 42.85 | 40.54 |
| MTA [CVPR'24] | 82.41 | 80.28 | 76.46 | 70.34 | 30.70 | 30.01 | 41.10 | 37.09 | 35.47 | 30.05 |
| TDA [CVPR'24] | 80.98 | 73.08 | 65.58 | 61.11 | 36.25 | 31.67 | 44.76 | 38.94 | 40.94 | 38.93 |
| ZERO [NeurIPS'24] | 79.24 | 71.14 | 62.13 | 62.01 | 33.04 | 32.28 | 41.53 | 36.24 | 36.56 | 36.78 |
| DPE [NeurIPS'24] | 81.47 | 79.20 | 56.63 | 55.13 | 33.62 | 30.70 | 42.36 | 41.07 | 37.88 | 38.02 |
| O-TPT [CVPR'25] | 79.42 | 72.26 | 63.10 | 61.88 | 32.07 | 30.84 | 41.10 | 36.52 | 36.10 | 34.81 |
| SCAP [CVPR'25] | 78.47 | 71.23 | 56.94 | 57.48 | 30.70 | 27.84 | 44.79 | 38.49 | 36.28 | 35.23 |
| L-TTA (Ours) | 86.36 | 80.76 | 61.49 | 59.55 | 35.31 | 32.13 | 52.72 | 48.94 | 38.86 | 37.64 |

| Method | Aircraft | | SUN397 | | DTD | | EuroSAT | | UCF101 | |
|---|---|---|---|---|---|---|---|---|---|---|
| | Acc. | Mac. | Acc. | Mac. | Acc. | Mac. | Acc. | Mac. | Acc. | Mac. |
| CLIP [ICML'21] | 7.72 | 8.91 | 35.77 | 34.22 | 17.47 | 15.91 | 6.70 | 3.96 | 38.30 | 30.67 |
| TPT [NeurIPS'22] | 9.28 | 11.12 | 42.19 | 39.79 | 29.01 | 25.69 | 16.18 | 15.47 | 43.52 | 37.04 |
| C-TPT [ICLR'24] | 11.85 | 11.96 | 44.45 | 41.54 | 30.11 | 25.79 | 19.95 | 10.94 | 43.02 | 37.29 |
| MTA [CVPR'24] | 7.57 | 5.96 | 50.42 | 46.65 | 25.98 | 20.23 | 18.94 | 12.47 | 51.85 | 42.59 |
| TDA [CVPR'24] | 9.27 | 8.05 | 41.38 | 37.45 | 19.81 | 13.77 | 12.66 | 8.23 | 43.41 | 34.93 |
| ZERO [NeurIPS'24] | 8.50 | 9.04 | 34.65 | 33.37 | 17.00 | 15.97 | 4.68 | 2.99 | 38.68 | 30.90 |
| DPE [NeurIPS'24] | 14.49 | 14.94 | 40.47 | 38.92 | 17.78 | 18.91 | 11.02 | 5.18 | 11.70 | 12.46 |
| O-TPT [CVPR'25] | 8.18 | 7.22 | 35.04 | 32.06 | 18.41 | 14.41 | 6.31 | 4.30 | 37.58 | 32.04 |
| SCAP [CVPR'25] | 10.19 | 10.85 | 46.89 | 46.73 | 24.44 | 19.16 | 8.72 | 7.90 | 46.42 | 42.09 |
| L-TTA (Ours) | 10.90 | 10.02 | 50.99 | 48.78 | 29.46 | 26.97 | 24.88 | 19.26 | 48.30 | 34.36 |

Table 26: **Detailed Accuracy on Long-tailed Corruption Benchmark of *speckle noise*. We conduct 5 runs for each experiment under each imbalance setting. All the methods are built upon CLIP-VIT-B/16 backbone. The best results are marked in Bold.**

| Method | Caltech | | Pets | | Cars | | Flowers | | Food101 | |
|---|---|---|---|---|---|---|---|---|---|---|
| | Acc. | Mac. | Acc. | Mac. | Acc. | Mac. | Acc. | Mac. | Acc. | Mac. |
| CLIP [ICML'21] | 89.16 | 84.61 | 73.94 | 71.96 | 51.38 | 51.72 | 48.81 | 44.67 | 53.38 | 51.15 |
| TPT [NeurIPS'22] | 89.63 | 85.33 | 74.62 | 72.82 | 52.80 | 52.59 | 47.73 | 43.67 | 53.04 | 50.80 |
| C-TPT [ICLR'24] | 88.38 | 82.57 | 72.42 | 70.28 | 53.10 | 52.43 | 44.09 | 39.84 | 51.21 | 50.92 |
| MTA [CVPR'24] | \ | \ | \ | \ | \ | \ | \ | \ | \ | \ |
| TDA [CVPR'24] | 90.28 | 85.53 | 75.92 | 73.29 | 58.54 | 53.30 | 55.45 | 49.75 | 57.99 | 55.04 |
| ZERO [NeurIPS'24] | 88.78 | 84.16 | 74.21 | 71.62 | 51.03 | 51.24 | 49.60 | 45.34 | 55.58 | 52.73 |
| DPE [NeurIPS'24] | 89.29 | 87.05 | 77.86 | 77.08 | 59.69 | 55.22 | 58.53 | 53.85 | 57.32 | 55.87 |
| O-TPT [CVPR'25] | 88.20 | 83.79 | 74.84 | 73.41 | 52.00 | 51.79 | 50.64 | 46.82 | 52.79 | 50.30 |
| SCAP [CVPR'25] | 87.98 | 80.85 | 76.48 | 75.59 | 55.30 | 54.21 | 54.64 | 50.28 | 59.64 | 57.70 |
| L-TTA (Ours) | 92.53 | 88.63 | 80.97 | 79.31 | 61.52 | 55.77 | 61.16 | 54.77 | 57.79 | 56.27 |

| Method | Aircraft | | SUN397 | | DTD | | EuroSAT | | UCF101 | |
|---|---|---|---|---|---|---|---|---|---|---|
| | Acc. | Mac. | Acc. | Mac. | Acc. | Mac. | Acc. | Mac. | Acc. | Mac. |
| CLIP [ICML'21] | 11.15 | 12.38 | 52.99 | 50.00 | 29.33 | 26.21 | 18.51 | 13.37 | 53.08 | 47.38 |
| TPT [NeurIPS'22] | 13.41 | 14.46 | 53.59 | 50.28 | 28.05 | 24.60 | 18.36 | 13.24 | 52.69 | 46.31 |
| C-TPT [ICLR'24] | 12.95 | 14.64 | 51.65 | 49.93 | 25.15 | 24.24 | 16.48 | 15.74 | 50.05 | 48.69 |
| MTA [CVPR'24] | \ | \ | \ | \ | \ | \ | \ | \ | \ | \ |
| TDA [CVPR'24] | 14.56 | 12.55 | 57.06 | 52.55 | 31.20 | 24.56 | 22.04 | 13.66 | 61.30 | 52.22 |
| ZERO [NeurIPS'24] | 11.61 | 12.90 | 53.63 | 49.76 | 29.80 | 26.15 | 16.00 | 10.60 | 53.95 | 48.01 |
| DPE [NeurIPS'24] | 21.26 | 19.74 | 61.27 | 57.90 | 34.95 | 33.37 | 26.86 | 18.68 | 60.91 | 53.58 |
| O-TPT [CVPR'25] | 11.85 | 11.52 | 52.50 | 48.57 | 29.80 | 25.46 | 18.78 | 11.25 | 53.02 | 46.96 |
| SCAP [CVPR'25] | 13.12 | 12.75 | 57.14 | 50.15 | 30.62 | 29.85 | 24.19 | 12.64 | 60.95 | 52.05 |
| L-TTA (Ours) | 17.50 | 16.68 | 61.54 | 57.17 | 39.44 | 37.74 | 27.42 | 20.47 | 60.82 | 54.97 |

Table 27: **Detailed Accuracy on Long-tailed Corruption Benchmark of *defocus blur*.** We conduct 5 runs for each experiment under each imbalance setting. All the methods are built upon **CLIP-VIT-B/16** backbone. The best results are marked in **Bold**.

| Method | Caltech | | Pets | | Cars | | Flowers | | Food101 | |
|---|---|---|---|---|---|---|---|---|---|---|
| | Acc. | Mac. | Acc. | Mac. | Acc. | Mac. | Acc. | Mac. | Acc. | Mac. |
| CLIP [ICML'21] | 44.96 | 37.04 | 19.46 | 17.69 | 3.12 | 2.33 | 12.78 | 9.48 | 9.88 | 8.04 |
| TPT [NeurIPS'22] | 45.86 | 37.79 | 18.83 | 17.13 | 3.82 | 2.95 | 12.78 | 9.54 | 10.35 | 8.14 |
| C-TPT [ICLR'24] | 41.77 | 34.51 | 18.35 | 18.26 | 3.21 | 2.43 | 13.09 | 9.84 | 11.21 | 8.92 |
| MTA [CVPR'24] | \ | \ | \ | \ | \ | \ | \ | \ | \ | \ |
| TDA [CVPR'24] | 45.69 | 37.67 | 18.81 | 16.62 | 3.51 | 2.31 | 13.94 | 9.58 | 11.43 | 9.69 |
| ZERO [NeurIPS'24] | 41.82 | 32.53 | 14.77 | 11.13 | 2.59 | 2.51 | 13.70 | 10.72 | 12.03 | 9.57 |
| DPE [NeurIPS'24] | 48.89 | 43.20 | 17.35 | 16.97 | 3.24 | 2.28 | 8.36 | 8.84 | 11.25 | 9.40 |
| O-TPT [CVPR'25] | 42.70 | 35.26 | 18.42 | 18.32 | 3.27 | 2.44 | 12.97 | 9.71 | 11.14 | 8.72 |
| SCAP [CVPR'25] | 40.90 | 38.33 | 19.58 | 19.91 | 3.99 | 3.80 | 10.79 | 10.98 | 12.94 | 12.12 |
| L-TTA (Ours) | 49.67 | 44.18 | 25.12 | 25.01 | 14.65 | 6.48 | 24.85 | 22.09 | 18.04 | 17.50 |
| Method | Aircraft | | SUN397 | | DTD | | EuroSAT | | UCF101 | |
| | Acc. | Mac. | Acc. | Mac. | Acc. | Mac. | Acc. | Mac. | Acc. | Mac. |
| CLIP [ICML'21] | 1.79 | 0.95 | 9.42 | 8.16 | 6.71 | 5.88 | 22.95 | 10.26 | 11.59 | 10.37 |
| TPT [NeurIPS'22] | 2.49 | 1.85 | 8.71 | 7.31 | 7.07 | 6.14 | 23.83 | 11.42 | 13.02 | 11.58 |
| C-TPT [ICLR'24] | 1.95 | 0.64 | 7.65 | 6.93 | 5.15 | 4.24 | 10.48 | 5.74 | 10.05 | 8.69 |
| MTA [CVPR'24] | \ | \ | \ | \ | \ | \ | \ | \ | \ | \ |
| TDA [CVPR'24] | 1.64 | 0.70 | 11.02 | 8.27 | 7.96 | 5.13 | 22.24 | 12.79 | 14.98 | 10.20 |
| ZERO [NeurIPS'24] | 2.49 | 1.60 | 10.40 | 9.79 | 5.77 | 4.38 | 20.01 | 8.76 | 14.34 | 12.07 |
| DPE [NeurIPS'24] | 3.58 | 3.47 | 12.61 | 11.80 | 12.95 | 11.79 | 29.23 | 19.24 | 8.42 | 7.58 |
| O-TPT [CVPR'25] | 2.26 | 0.69 | 7.79 | 7.04 | 4.84 | 3.85 | 11.11 | 6.68 | 10.00 | 8.68 |
| SCAP [CVPR'25] | 1.29 | 2.02 | 10.15 | 9.42 | 12.34 | 10.44 | 23.88 | 10.86 | 12.36 | 11.02 |
| L-TTA (Ours) | 8.26 | 6.77 | 12.34 | 11.52 | 18.30 | 14.61 | 32.00 | 20.74 | 23.56 | 18.16 |

Table 28: **Detailed Accuracy on Long-tailed Corruption Benchmark of *glass blur*.** We conduct 5 runs for each experiment under each imbalance setting. All the methods are built upon **CLIP-VIT-B/16** backbone. The best results are marked in **Bold**.

| Method | Caltech | | Pets | | Cars | | Flowers | | Food101 | |
|---|---|---|---|---|---|---|---|---|---|---|
| | Acc. | Mac. | Acc. | Mac. | Acc. | Mac. | Acc. | Mac. | Acc. | Mac. |
| CLIP [ICML'21] | 77.77 | 70.02 | 59.97 | 57.35 | 22.61 | 21.05 | 37.06 | 32.76 | 37.96 | 37.38 |
| TPT [NeurIPS'22] | 79.16 | 72.51 | 59.80 | 57.05 | 23.08 | 21.31 | 37.12 | 32.82 | 38.84 | 38.48 |
| C-TPT [ICLR'24] | 79.07 | 72.14 | 61.50 | 59.05 | 22.69 | 21.42 | 39.27 | 34.77 | 37.76 | 36.35 |
| MTA [CVPR'24] | \ | \ | \ | \ | \ | \ | \ | \ | \ | \ |
| TDA [CVPR'24] | 80.45 | 72.20 | 64.40 | 60.96 | 28.08 | 23.53 | 43.24 | 36.96 | 43.00 | 40.56 |
| ZERO [NeurIPS'24] | 78.32 | 70.68 | 62.82 | 60.00 | 24.65 | 22.71 | 37.49 | 32.66 | 41.67 | 40.86 |
| DPE [NeurIPS'24] | 82.08 | 74.47 | 62.18 | 59.13 | 25.21 | 22.11 | 41.01 | 39.35 | 40.33 | 40.27 |
| O-TPT [CVPR'25] | 78.57 | 71.63 | 61.57 | 59.02 | 22.85 | 21.36 | 39.39 | 34.78 | 37.78 | 36.31 |
| SCAP [CVPR'25] | 80.22 | 75.69 | 62.15 | 81.13 | 22.45 | 21.78 | 40.25 | 35.62 | 39.08 | 38.08 |
| L-TTA (Ours) | 82.76 | 78.39 | 67.61 | 64.78 | 31.24 | 30.26 | 43.39 | 38.27 | 43.18 | 42.22 |
| Method | Aircraft | | SUN397 | | DTD | | EuroSAT | | UCF101 | |
| | Acc. | Mac. | Acc. | Mac. | Acc. | Mac. | Acc. | Mac. | Acc. | Mac. |
| CLIP [ICML'21] | 6.78 | 6.91 | 35.16 | 32.69 | 17.63 | 14.20 | 27.79 | 21.09 | 37.42 | 33.72 |
| TPT [NeurIPS'22] | 6.87 | 7.33 | 36.41 | 34.02 | 18.28 | 14.65 | 28.54 | 22.38 | 37.86 | 34.32 |
| C-TPT [ICLR'24] | 5.38 | 4.93 | 32.93 | 29.78 | 17.78 | 13.66 | 13.26 | 13.94 | 37.20 | 32.49 |
| MTA [CVPR'24] | \ | \ | \ | \ | \ | \ | \ | \ | \ | \ |
| TDA [CVPR'24] | 5.84 | 4.92 | 38.52 | 34.19 | 18.41 | 11.17 | 30.89 | 32.22 | 43.78 | 36.52 |
| ZERO [NeurIPS'24] | 8.26 | 8.58 | 37.30 | 34.27 | 19.50 | 17.13 | 24.76 | 18.10 | 40.98 | 36.20 |
| DPE [NeurIPS'24] | 10.67 | 10.20 | 40.51 | 39.13 | 21.06 | 21.82 | 35.60 | 26.93 | 47.49 | 42.51 |
| O-TPT [CVPR'25] | 5.14 | 4.84 | 32.70 | 29.58 | 17.47 | 13.56 | 14.24 | 14.40 | 37.14 | 32.23 |
| SCAP [CVPR'25] | 6.20 | 7.19 | 38.94 | 34.62 | 20.85 | 15.11 | 36.39 | 28.28 | 42.64 | 37.50 |
| L-TTA (Ours) | 14.22 | 12.95 | 42.11 | 40.49 | 23.89 | 21.42 | 41.61 | 32.12 | 42.01 | 39.29 |

Table 29: **Detailed Accuracy on Long-tailed Corruption Benchmark of *motion blur*.** We conduct 5 runs for each experiment under each imbalance setting. All the methods are built upon **CLIP-VIT-B/16** backbone. The best results are marked in **Bold**.

| Method | Caltech | | Pets | | Cars | | Flowers | | Food101 | |
|---|---|---|---|---|---|---|---|---|---|---|
| | Acc. | Mac. | Acc. | Mac. | Acc. | Mac. | Acc. | Mac. | Acc. | Mac. |
| CLIP [ICML'21] | 44.96 | 37.04 | 19.46 | 17.69 | 3.12 | 2.33 | 12.78 | 9.48 | 9.88 | 8.04 |
| TPT [NeurIPS'22] | 45.28 | 37.16 | 19.80 | 17.91 | 3.98 | 2.36 | 12.58 | 9.17 | 10.18 | 8.52 |
| C-TPT [ICLR'24] | 41.77 | 34.51 | 18.35 | 18.26 | 3.21 | 2.43 | 13.09 | 9.84 | 11.21 | 8.92 |
| MTA [CVPR'24] | \ | \ | \ | \ | \ | \ | \ | \ | \ | \ |
| TDA [CVPR'24] | 45.45 | 37.46 | 19.64 | 17.04 | 3.76 | 2.48 | 14.23 | 9.60 | 11.35 | 9.67 |
| ZERO [NeurIPS'24] | 41.77 | 30.56 | 13.90 | 11.27 | 4.61 | 3.60 | 13.70 | 10.72 | 12.03 | 9.56 |
| DPE [NeurIPS'24] | 45.28 | 40.16 | 17.77 | 15.11 | 1.17 | 20.45 | 7.98 | 9.18 | 10.67 | 8.80 |
| O-TPT [CVPR'25] | 42.70 | 35.26 | 18.42 | 18.32 | 3.27 | 2.44 | 12.97 | 9.71 | 11.14 | 8.72 |
| SCAP [CVPR'25] | 44.83 | 37.17 | 16.44 | 15.89 | 5.02 | 3.74 | 14.85 | 12.22 | 9.85 | 8.31 |
| L-TTA (Ours) | 48.64 | 42.46 | 25.95 | 22.78 | 11.27 | 10.13 | 16.66 | 15.62 | 18.52 | 17.36 |
| Method | Aircraft | | SUN397 | | DTD | | EuroSAT | | UCF101 | |
| | Acc. | Mac. | Acc. | Mac. | Acc. | Mac. | Acc. | Mac. | Acc. | Mac. |
| CLIP [ICML'21] | 1.79 | 0.95 | 9.39 | 8.15 | 6.71 | 5.88 | 22.95 | 10.26 | 11.59 | 10.37 |
| TPT [NeurIPS'22] | 2.19 | 1.15 | 11.25 | 8.94 | 8.38 | 7.77 | 24.25 | 11.38 | 11.75 | 10.53 |
| C-TPT [ICLR'24] | 1.95 | 0.64 | 7.65 | 6.93 | 5.15 | 4.24 | 10.48 | 5.74 | 10.05 | 8.69 |
| MTA [CVPR'24] | \ | \ | \ | \ | \ | \ | \ | \ | \ | \ |
| TDA [CVPR'24] | 2.49 | 1.03 | 11.11 | 8.35 | 8.11 | 4.37 | 25.10 | 13.97 | 13.34 | 9.28 |
| ZERO [NeurIPS'24] | 2.02 | 0.98 | 7.89 | 6.75 | 7.48 | 4.10 | 19.34 | 9.02 | 8.92 | 8.09 |
| DPE [NeurIPS'24] | 3.27 | 2.78 | 8.92 | 9.46 | 15.44 | 15.44 | 28.50 | 18.60 | 13.71 | 11.84 |
| O-TPT [CVPR'25] | 2.26 | 0.69 | 7.79 | 7.04 | 4.84 | 3.85 | 11.11 | 6.68 | 10.00 | 8.68 |
| SCAP [CVPR'25] | 1.89 | 1.11 | 9.72 | 9.83 | 7.69 | 7.70 | 20.75 | 16.20 | 14.84 | 12.64 |
| L-TTA (Ours) | 5.90 | 4.60 | 15.25 | 13.00 | 23.82 | 20.60 | 30.63 | 24.80 | 16.75 | 13.18 |

Table 30: **Detailed Accuracy on Long-tailed Corruption Benchmark of *zoom blur*.** We conduct 5 runs for each experiment under each imbalance setting. All the methods are built upon **CLIP-VIT-B/16** backbone. The best results are marked in **Bold**.

| Method | Caltech | | Pets | | Cars | | Flowers | | Food101 | |
|---|---|---|---|---|---|---|---|---|---|---|
| | Acc. | Mac. | Acc. | Mac. | Acc. | Mac. | Acc. | Mac. | Acc. | Mac. |
| CLIP [ICML'21] | 92.97 | 89.91 | 84.50 | 84.36 | 58.85 | 59.60 | 62.57 | 58.29 | 78.11 | 77.24 |
| TPT [NeurIPS'22] | 93.66 | 90.49 | 85.11 | 84.98 | 59.21 | 59.89 | 63.50 | 59.21 | 78.96 | 78.20 |
| C-TPT [ICLR'24] | 92.76 | 89.92 | 84.02 | 84.94 | 57.49 | 58.19 | 64.04 | 59.54 | 77.15 | 76.09 |
| MTA [CVPR'24] | \ | \ | \ | \ | \ | \ | \ | \ | \ | \ |
| TDA [CVPR'24] | 93.66 | 90.48 | 88.48 | 86.35 | 64.88 | 59.01 | 69.34 | 61.76 | 81.53 | 78.85 |
| ZERO [NeurIPS'24] | 86.36 | 79.71 | 81.54 | 79.57 | 59.20 | 54.50 | 63.38 | 56.50 | 78.23 | 76.59 |
| DPE [NeurIPS'24] | 93.54 | 90.12 | 87.37 | 86.90 | 66.19 | 61.74 | 70.55 | 63.98 | 81.05 | 78.86 |
| O-TPT [CVPR'25] | 92.80 | 89.98 | 84.02 | 85.01 | 57.67 | 58.21 | 64.10 | 59.66 | 76.99 | 75.79 |
| SCAP [CVPR'25] | 94.03 | 91.36 | 86.26 | 86.60 | 60.91 | 61.19 | 69.14 | 66.77 | 81.09 | 79.29 |
| L-TTA (Ours) | 94.17 | 90.65 | 90.35 | 89.38 | 68.76 | 62.55 | 71.16 | 68.30 | 82.58 | 79.80 |
| Method | Aircraft | | SUN397 | | DTD | | EuroSAT | | UCF101 | |
| | Acc. | Mac. | Acc. | Mac. | Acc. | Mac. | Acc. | Mac. | Acc. | Mac. |
| CLIP [ICML'21] | 16.60 | 17.90 | 60.77 | 58.04 | 37.75 | 32.79 | 38.84 | 33.96 | 64.40 | 60.83 |
| TPT [NeurIPS'22] | 16.83 | 18.12 | 61.86 | 59.10 | 39.53 | 34.43 | 40.35 | 35.39 | 65.48 | 61.77 |
| C-TPT [ICLR'24] | 15.90 | 15.59 | 58.27 | 54.22 | 37.91 | 33.09 | 32.25 | 31.90 | 63.35 | 58.09 |
| MTA [CVPR'24] | \ | \ | \ | \ | \ | \ | \ | \ | \ | \ |
| TDA [CVPR'24] | 21.03 | 18.36 | 64.42 | 60.50 | 40.87 | 33.26 | 42.15 | 43.20 | 69.51 | 62.53 |
| ZERO [NeurIPS'24] | 17.00 | 14.65 | 61.84 | 59.36 | 38.76 | 34.56 | 40.56 | 39.37 | 65.81 | 61.48 |
| DPE [NeurIPS'24] | 22.43 | 20.37 | 65.28 | 61.33 | 48.36 | 43.54 | 45.50 | 42.52 | 72.33 | 65.27 |
| O-TPT [CVPR'25] | 15.98 | 15.92 | 58.06 | 54.17 | 38.38 | 33.26 | 39.91 | 34.68 | 63.46 | 58.12 |
| SCAP [CVPR'25] | 19.48 | 19.83 | 62.23 | 57.68 | 42.10 | 34.50 | 43.60 | 36.28 | 65.46 | 60.64 |
| L-TTA (Ours) | 27.05 | 25.40 | 66.44 | 63.05 | 50.69 | 46.08 | 47.68 | 44.74 | 71.87 | 68.67 |

Table 31: **Detailed Accuracy on Long-tailed Corruption Benchmark of** *snow*. We conduct 5 runs for each experiment under each imbalance setting. All the methods are built upon **CLIP-VIT-B/16** backbone. The best results are marked in **Bold**.

| Method | Caltech | | Pets | | Cars | | Flowers | | Food101 | |
|---|---|---|---|---|---|---|---|---|---|---|
| | Acc. | Mac. | Acc. | Mac. | Acc. | Mac. | Acc. | Mac. | Acc. | Mac. |
| CLIP [ICML'21] | 88.99 | 85.45 | 77.00 | 77.05 | 50.75 | 50.27 | 55.29 | 50.69 | 72.44 | 71.13 |
| TPT [NeurIPS'22] | 89.22 | 85.10 | 77.19 | 77.30 | 50.61 | 49.91 | 55.53 | 50.89 | 71.87 | 70.11 |
| C-TPT [ICLR'24] | 89.54 | 86.24 | 78.25 | 78.81 | 48.55 | 47.67 | 56.21 | 51.19 | 71.21 | 69.79 |
| MTA [CVPR'24] | \ | | \ | | \ | | \ | | \ | |
| TDA [CVPR'24] | 91.95 | 88.46 | 83.34 | 80.42 | 57.54 | 51.94 | 60.36 | 52.44 | 76.58 | 73.83 |
| ZERO [NeurIPS'24] | 84.55 | 82.13 | 78.89 | 75.68 | 52.88 | 51.90 | 58.13 | 49.01 | 68.56 | 67.16 |
| DPE [NeurIPS'24] | 91.58 | 88.10 | 85.55 | 82.03 | 58.18 | 51.99 | 61.54 | 53.48 | 76.40 | 74.95 |
| O-TPT [CVPR'25] | 89.66 | 86.14 | 78.67 | 79.10 | 48.97 | 48.02 | 56.33 | 51.27 | 71.17 | 69.57 |
| SCAP [CVPR'25] | 90.97 | 84.56 | 82.96 | 82.19 | 49.40 | 49.76 | 60.15 | 56.73 | 74.38 | 73.14 |
| L-TTA (Ours) | 93.79 | 88.74 | 88.19 | 85.46 | 62.29 | 62.97 | 66.62 | 58.61 | 78.44 | 75.86 |
| Method | Aircraft | | SUN397 | | DTD | | EuroSAT | | UCF101 | |
| | Acc. | Mac. | Acc. | Mac. | Acc. | Mac. | Acc. | Mac. | Acc. | Mac. |
| CLIP [ICML'21] | 11.77 | 12.91 | 54.84 | 51.84 | 33.23 | 29.23 | 35.63 | 31.27 | 59.29 | 55.42 |
| TPT [NeurIPS'22] | 11.79 | 12.54 | 53.99 | 51.51 | 33.30 | 29.57 | 35.24 | 31.28 | 57.12 | 53.17 |
| C-TPT [ICLR'24] | 12.70 | 13.15 | 51.84 | 48.12 | 31.51 | 26.39 | 32.20 | 28.96 | 57.75 | 52.70 |
| MTA [CVPR'24] | \ | | \ | | \ | | \ | | \ | |
| TDA [CVPR'24] | 15.97 | 13.60 | 60.16 | 55.80 | 34.48 | 28.81 | 42.82 | 38.00 | 67.34 | 59.52 |
| ZERO [NeurIPS'24] | 11.78 | 10.22 | 52.50 | 48.84 | 33.10 | 26.00 | 38.65 | 39.13 | 62.90 | 56.35 |
| DPE [NeurIPS'24] | 16.12 | 14.08 | 62.20 | 56.86 | 34.36 | 28.42 | 45.51 | 41.01 | 66.69 | 59.19 |
| O-TPT [CVPR'25] | 12.78 | 13.24 | 51.71 | 48.03 | 31.36 | 26.21 | 32.42 | 29.11 | 57.53 | 52.61 |
| SCAP [CVPR'25] | 10.36 | 10.45 | 53.66 | 50.85 | 30.28 | 25.57 | 39.06 | 35.27 | 62.32 | 56.20 |
| L-TTA (Ours) | 19.58 | 18.13 | 55.12 | 52.36 | 36.08 | 31.38 | 48.95 | 44.26 | 67.09 | 62.53 |

Table 32: **Detailed Accuracy on Long-tailed Corruption Benchmark of** *frost*. We conduct 5 runs for each experiment under each imbalance setting. All the methods are built upon **CLIP-VIT-B/16** backbone. The best results are marked in **Bold**.

| Method | Caltech | | Pets | | Cars | | Flowers | | Food101 | |
|---|---|---|---|---|---|---|---|---|---|---|
| | Acc. | Mac. | Acc. | Mac. | Acc. | Mac. | Acc. | Mac. | Acc. | Mac. |
| CLIP [ICML'21] | 14.40 | 10.74 | 7.99 | 7.17 | 1.32 | 0.76 | 4.34 | 2.61 | 2.63 | 0.90 |
| TPT [NeurIPS'22] | 15.46 | 12.32 | 7.88 | 7.80 | 0.79 | 0.22 | 4.07 | 2.41 | 1.68 | 0.28 |
| C-TPT [ICLR'24] | 13.81 | 10.51 | 8.34 | 8.18 | 1.43 | 0.93 | 4.40 | 2.53 | 1.94 | 0.73 |
| MTA [CVPR'24] | \ | | \ | | \ | | \ | | \ | |
| TDA [CVPR'24] | 18.58 | 12.40 | 6.94 | 7.00 | 1.32 | 1.04 | 4.27 | 2.19 | 2.15 | 1.04 |
| ZERO [NeurIPS'24] | 14.81 | 11.43 | 2.10 | 7.57 | 1.15 | 1.08 | 3.18 | 1.44 | 2.80 | 1.59 |
| DPE [NeurIPS'24] | 14.82 | 11.43 | 7.57 | 7.08 | 1.35 | 0.84 | 3.18 | 1.45 | 2.66 | 1.01 |
| O-TPT [CVPR'25] | 14.27 | 10.85 | 8.34 | 8.22 | 1.47 | 1.04 | 4.40 | 2.48 | 1.98 | 0.72 |
| SCAP [CVPR'25] | 16.54 | 11.24 | 8.95 | 8.81 | 2.55 | 1.67 | 5.21 | 3.19 | 2.21 | 1.79 |
| L-TTA (Ours) | 25.67 | 20.23 | 12.22 | 11.65 | 8.78 | 8.29 | 16.12 | 15.13 | 5.89 | 5.50 |
| Method | Aircraft | | SUN397 | | DTD | | EuroSAT | | UCF101 | |
| | Acc. | Mac. | Acc. | Mac. | Acc. | Mac. | Acc. | Mac. | Acc. | Mac. |
| CLIP [ICML'21] | 1.48 | 0.83 | 2.60 | 1.83 | 11.54 | 6.70 | 8.95 | 3.25 | 4.23 | 3.83 |
| TPT [NeurIPS'22] | 1.79 | 0.90 | 3.34 | 2.38 | 9.29 | 4.87 | 10.21 | 4.89 | 4.69 | 4.84 |
| C-TPT [ICLR'24] | 1.79 | 0.70 | 2.60 | 2.31 | 7.96 | 5.15 | 7.92 | 2.35 | 3.19 | 3.93 |
| MTA [CVPR'24] | \ | | \ | | \ | | \ | | \ | |
| TDA [CVPR'24] | 2.41 | 1.34 | 3.05 | 1.99 | 14.98 | 7.69 | 6.67 | 4.10 | 6.51 | 4.00 |
| ZERO [NeurIPS'24] | 2.16 | 1.14 | 3.05 | 1.43 | 10.09 | 7.25 | 8.48 | 3.08 | 4.66 | 4.41 |
| DPE [NeurIPS'24] | 3.34 | 2.69 | 3.03 | 2.44 | 8.73 | 7.01 | 5.95 | 4.80 | 7.00 | 5.82 |
| O-TPT [CVPR'25] | 1.79 | 0.60 | 2.50 | 2.14 | 8.27 | 5.42 | 4.79 | 2.09 | 3.08 | 3.86 |
| SCAP [CVPR'25] | 2.70 | 2.15 | 2.84 | 2.12 | 10.51 | 6.39 | 5.20 | 4.96 | 7.75 | 4.70 |
| L-TTA (Ours) | 8.96 | 8.03 | 3.63 | 2.74 | 20.23 | 18.71 | 19.50 | 15.49 | 6.03 | 4.04 |

Table 33: **Detailed Accuracy on Long-tailed Corruption Benchmark of *fog*.** We conduct 5 runs for each experiment under each imbalance setting. All the methods are built upon **CLIP-VIT-B/16** backbone. The best results are marked in **Bold**.

| Method | Caltech | | Pets | | Cars | | Flowers | | Food101 | |
|---|---|---|---|---|---|---|---|---|---|---|
| | Acc. | Mac. | Acc. | Mac. | Acc. | Mac. | Acc. | Mac. | Acc. | Mac. |
| CLIP [ICML'21] | 20.39 | 13.55 | 5.35 | 3.87 | 0.55 | 0.15 | 5.14 | 3.65 | 2.26 | 1.24 |
| TPT [NeurIPS'22] | 19.59 | 13.09 | 6.49 | 5.43 | 0.88 | 0.16 | 4.96 | 3.48 | 3.60 | 2.15 |
| C-TPT [ICLR'24] | 18.29 | 12.88 | 4.66 | 3.16 | 0.55 | 0.11 | 4.83 | 2.87 | 3.31 | 1.86 |
| MTA [CVPR'24] | \ | \ | \ | \ | \ | \ | \ | \ | \ | \ |
| TDA [CVPR'24] | 19.84 | 14.74 | 5.48 | 2.68 | 0.46 | 0.17 | 5.15 | 2.81 | 2.48 | 1.68 |
| ZERO [NeurIPS'24] | 23.65 | 16.61 | 6.32 | 5.18 | 0.58 | 0.24 | 4.71 | 2.77 | 1.84 | 1.04 |
| DPE [NeurIPS'24] | 19.28 | 18.10 | 4.16 | 3.50 | 0.64 | 0.14 | 2.36 | 2.30 | 2.87 | 2.30 |
| O-TPT [CVPR'25] | 18.92 | 13.30 | 4.66 | 3.31 | 0.57 | 0.18 | 4.83 | 3.01 | 3.37 | 1.86 |
| SCAP [CVPR'25] | 18.43 | 16.73 | 4.99 | 4.17 | 1.56 | 1.37 | 4.34 | 1.03 | 3.00 | 1.08 |
| L-TTA (Ours) | 31.91 | 26.47 | 10.56 | 9.37 | 10.79 | 10.18 | 16.67 | 15.12 | 5.52 | 5.45 |

| Method | Aircraft | | SUN397 | | DTD | | EuroSAT | | UCF101 | |
|---|---|---|---|---|---|---|---|---|---|---|
| | Acc. | Mac. | Acc. | Mac. | Acc. | Mac. | Acc. | Mac. | Acc. | Mac. |
| CLIP [ICML'21] | 0.31 | 0.13 | 1.99 | 1.22 | 1.87 | 1.67 | 9.93 | 3.96 | 1.15 | 0.88 |
| TPT [NeurIPS'22] | 1.05 | 1.63 | 3.48 | 2.53 | 2.84 | 2.63 | 9.78 | 3.68 | 2.40 | 1.72 |
| C-TPT [ICLR'24] | 0.70 | 0.14 | 1.74 | 1.29 | 1.40 | 1.18 | 13.47 | 4.03 | 1.59 | 1.27 |
| MTA [CVPR'24] | \ | \ | \ | \ | \ | \ | \ | \ | \ | \ |
| TDA [CVPR'24] | 0.78 | 0.17 | 2.63 | 1.36 | 4.84 | 2.61 | 18.29 | 6.85 | 2.33 | 1.55 |
| ZERO [NeurIPS'24] | 0.49 | 0.39 | 3.00 | 1.32 | 4.44 | 2.15 | 9.94 | 5.06 | 1.12 | 0.63 |
| DPE [NeurIPS'24] | 1.56 | 1.20 | 3.82 | 2.10 | 4.28 | 3.00 | 20.60 | 11.37 | 2.07 | 1.29 |
| O-TPT [CVPR'25] | 1.09 | 0.22 | 1.77 | 1.27 | 1.40 | 0.99 | 12.30 | 4.22 | 1.54 | 1.13 |
| SCAP [CVPR'25] | 1.58 | 0.60 | 1.88 | 1.68 | 4.34 | 2.84 | 14.08 | 8.09 | 1.55 | 1.40 |
| L-TTA (Ours) | 8.87 | 7.22 | 3.09 | 2.89 | 18.66 | 15.32 | 19.03 | 13.78 | 3.11 | 0.29 |

Table 34: **Detailed Accuracy on Long-tailed Corruption Benchmark of *brightness*.** We conduct 5 runs for each experiment under each imbalance setting. All the methods are built upon **CLIP-VIT-B/16** backbone. The best results are marked in **Bold**.

| Method | Caltech | | Pets | | Cars | | Flowers | | Food101 | |
|---|---|---|---|---|---|---|---|---|---|---|
| | Acc. | Mac. | Acc. | Mac. | Acc. | Mac. | Acc. | Mac. | Acc. | Mac. |
| CLIP [ICML'21] | 91.04 | 88.18 | 82.90 | 82.62 | 57.91 | 58.49 | 60.73 | 57.73 | 79.32 | 78.15 |
| TPT [NeurIPS'22] | 95.54 | 92.37 | 86.88 | 84.13 | 60.90 | 60.94 | 62.25 | 59.35 | 81.24 | 79.20 |
| C-TPT [ICLR'24] | 90.87 | 87.74 | 82.90 | 83.31 | 58.33 | 58.78 | 62.57 | 59.46 | 78.80 | 77.51 |
| MTA [CVPR'24] | 86.13 | 84.94 | 78.33 | 80.08 | 65.19 | 56.37 | 61.77 | 58.89 | 76.33 | 73.52 |
| TDA [CVPR'24] | 93.09 | 89.87 | 85.70 | 83.50 | 64.98 | 60.49 | 68.03 | 59.70 | 82.05 | 79.72 |
| ZERO [NeurIPS'24] | 90.49 | 87.16 | 82.27 | 81.98 | 58.65 | 58.86 | 61.77 | 57.54 | 74.28 | 70.82 |
| DPE [NeurIPS'24] | 91.40 | 90.03 | 87.93 | 84.99 | 66.42 | 63.56 | 69.30 | 62.3 | 83.08 | 79.59 |
| O-TPT [CVPR'25] | 90.41 | 87.31 | 83.04 | 83.52 | 58.26 | 58.73 | 62.08 | 59.14 | 78.76 | 77.41 |
| SCAP [CVPR'25] | 91.25 | 88.63 | 85.69 | 85.16 | 59.67 | 57.76 | 63.69 | 60.46 | 79.82 | 78.07 |
| L-TTA (Ours) | 64.46 | 91.79 | 90.34 | 88.97 | 66.13 | 60.81 | 72.68 | 66.63 | 85.88 | 81.64 |

| Method | Aircraft | | SUN397 | | DTD | | EuroSAT | | UCF101 | |
|---|---|---|---|---|---|---|---|---|---|---|
| | Acc. | Mac. | Acc. | Mac. | Acc. | Mac. | Acc. | Mac. | Acc. | Mac. |
| CLIP [ICML'21] | 15.82 | 17.00 | 59.27 | 56.56 | 40.87 | 33.71 | 37.92 | 28.61 | 59.78 | 55.60 |
| TPT [NeurIPS'22] | 18.26 | 19.48 | 61.53 | 59.07 | 43.54 | 36.10 | 42.04 | 33.15 | 62.51 | 58.27 |
| C-TPT [ICLR'24] | 18.32 | 18.40 | 59.85 | 55.68 | 41.97 | 35.91 | 36.74 | 28.96 | 60.88 | 55.16 |
| MTA [CVPR'24] | 15.42 | 19.42 | 61.18 | 58.13 | 35.77 | 29.19 | 38.88 | 31.67 | 58.92 | 54.65 |
| TDA [CVPR'24] | 20.64 | 16.72 | 64.61 | 60.45 | 39.16 | 31.80 | 35.73 | 30.90 | 60.92 | 53.99 |
| ZERO [NeurIPS'24] | 17.16 | 11.59 | 58.29 | 52.10 | 40.90 | 33.86 | 38.91 | 30.95 | 60.03 | 52.80 |
| DPE [NeurIPS'24] | 22.61 | 18.15 | 65.06 | 58.43 | 37.09 | 31.25 | 40.92 | 33.08 | 64.46 | 57.45 |
| O-TPT [CVPR'25] | 17.54 | 17.85 | 59.38 | 55.25 | 42.12 | 36.57 | 37.81 | 30.06 | 60.82 | 55.01 |
| SCAP [CVPR'25] | 18.12 | 18.86 | 62.39 | 59.36 | 46.00 | 43.24 | 61.82 | 55.79 | 65.77 | 61.52 |
| L-TTA (Ours) | 24.07 | 22.99 | 67.15 | 63.26 | 47.74 | 40.50 | 67.34 | 62.04 | 67.33 | 62.04 |

Table 35: **Detailed Accuracy on Long-tailed Corruption Benchmark of *contrast*.** We conduct 5 runs for each experiment under each imbalance setting. All the methods are built upon **CLIP-VIT-B/16** backbone. The best results are marked in **Bold**.

| Method | Caltech | | Pets | | Cars | | Flowers | | Food101 | |
|---|---|---|---|---|---|---|---|---|---|---|
| | Acc. | Mac. | Acc. | Mac. | Acc. | Mac. | Acc. | Mac. | Acc. | Mac. |
| CLIP [ICML'21] | 92.34 | 89.92 | 84.16 | 84.66 | 59.14 | 59.52 | 63.18 | 60.15 | 79.37 | 78.52 |
| TPT [NeurIPS'22] | 95.33 | 92.39 | 86.66 | 87.64 | 62.92 | 62.83 | 64.46 | 61.93 | 82.00 | 81.40 |
| C-TPT [ICLR'24] | 92.72 | 90.13 | 83.25 | 83.94 | 59.36 | 59.62 | 65.20 | 61.91 | 78.56 | 77.79 |
| MTA [CVPR'24] | 87.71 | 86.33 | 80.15 | 78.29 | 60.04 | 58.24 | 63.88 | 59.91 | 72.86 | 70.31 |
| TDA [CVPR'24] | 94.02 | 91.20 | 85.91 | 84.90 | 67.05 | 61.85 | 69.55 | 61.55 | 82.72 | 80.54 |
| ZERO [NeurIPS'24] | 90.34 | 87.69 | 83.80 | 80.92 | 60.84 | 57.57 | 62.76 | 56.36 | 76.20 | 74.08 |
| DPE [NeurIPS'24] | 94.47 | 92.30 | 86.71 | 85.37 | 67.97 | 62.19 | 72.15 | 64.80 | 83.60 | 82.47 |
| O-TPT [CVPR'25] | 92.09 | 89.49 | 83.11 | 83.90 | 59.23 | 59.44 | 64.83 | 61.58 | 78.45 | 77.64 |
| SCAP [CVPR'25] | 92.39 | 90.20 | 85.38 | 83.94 | 62.94 | 62.53 | 65.05 | 60.07 | 82.16 | 78.35 |
| L-TTA (Ours) | 95.90 | 93.29 | 88.75 | 87.65 | 69.00 | 64.25 | 75.50 | 66.37 | 85.16 | 82.87 |

| Method | Aircraft | | SUN397 | | DTD | | EuroSAT | | UCF101 | |
|---|---|---|---|---|---|---|---|---|---|---|
| | Acc. | Mac. | Acc. | Mac. | Acc. | Mac. | Acc. | Mac. | Acc. | Mac. |
| CLIP [ICML'21] | 18.55 | 18.24 | 60.89 | 57.83 | 43.21 | 36.75 | 41.78 | 31.87 | 61.37 | 56.71 |
| TPT [NeurIPS'22] | 19.50 | 19.53 | 64.55 | 61.23 | 44.39 | 38.43 | 43.71 | 34.05 | 64.18 | 59.95 |
| C-TPT [ICLR'24] | 19.10 | 18.09 | 60.74 | 56.66 | 42.12 | 34.88 | 39.38 | 32.64 | 61.98 | 56.93 |
| MTA [CVPR'24] | 20.71 | 17.50 | 62.03 | 60.86 | 36.12 | 30.38 | 38.42 | 33.91 | 58.78 | 52.04 |
| TDA [CVPR'24] | 22.04 | 17.80 | 66.79 | 63.05 | 44.15 | 34.76 | 46.45 | 45.78 | 67.44 | 59.77 |
| ZERO [NeurIPS'24] | 18.90 | 18.48 | 62.18 | 59.80 | 41.24 | 33.48 | 35.25 | 28.56 | 62.98 | 56.16 |
| DPE [NeurIPS'24] | 25.45 | 21.10 | 67.04 | 64.18 | 47.74 | 41.94 | 48.22 | 46.43 | 68.55 | 60.25 |
| O-TPT [CVPR'25] | 18.55 | 18.19 | 60.53 | 56.34 | 41.97 | 34.04 | 39.90 | 32.88 | 61.98 | 56.78 |
| SCAP [CVPR'25] | 20.96 | 20.29 | 64.22 | 59.37 | 43.73 | 37.36 | 40.54 | 32.04 | 63.88 | 59.71 |
| L-TTA (Ours) | 28.76 | 24.30 | 67.78 | 65.84 | 48.25 | 42.43 | 63.84 | 60.96 | 70.21 | 63.95 |

Table 36: **Detailed Accuracy on Long-tailed Corruption Benchmark of *jpeg compression*.** We conduct 5 runs for each experiment under each imbalance setting. All the methods are built upon **CLIP-VIT-B/16** backbone. The best results are marked in **Bold**.

| Method | Caltech | | Pets | | Cars | | Flowers | | Food101 | |
|---|---|---|---|---|---|---|---|---|---|---|
| | Acc. | Mac. | Acc. | Mac. | Acc. | Mac. | Acc. | Mac. | Acc. | Mac. |
| CLIP [ICML'21] | 90.21 | 86.50 | 72.69 | 74.16 | 55.65 | 53.72 | 59.69 | 54.92 | 68.83 | 67.05 |
| TPT [NeurIPS'22] | 91.07 | 87.13 | 74.70 | 75.94 | 59.97 | 57.85 | 62.99 | 58.66 | 72.91 | 71.54 |
| C-TPT [ICLR'24] | 91.04 | 87.23 | 75.12 | 76.88 | 53.25 | 51.74 | 61.65 | 57.24 | 68.30 | 66.26 |
| MTA [CVPR'24] | 89.20 | 86.46 | 68.46 | 62.90 | 52.99 | 50.40 | 56.37 | 54.41 | 66.51 | 64.79 |
| TDA [CVPR'24] | 92.32 | 88.39 | 81.05 | 79.25 | 62.72 | 56.91 | 66.34 | 58.98 | 72.16 | 69.51 |
| ZERO [NeurIPS'24] | 80.74 | 86.37 | 82.68 | 76.13 | 57.87 | 53.47 | 60.39 | 55.70 | 66.50 | 58.42 |
| DPE [NeurIPS'24] | 92.99 | 88.04 | 84.56 | 80.77 | 63.35 | 58.46 | 68.01 | 59.72 | 74.28 | 70.31 |
| O-TPT [CVPR'25] | 91.04 | 87.44 | 75.47 | 77.04 | 53.62 | 52.12 | 61.41 | 56.98 | 68.20 | 66.02 |
| SCAP [CVPR'25] | 91.95 | 88.23 | 78.72 | 80.74 | 54.04 | 54.67 | 64.06 | 61.25 | 70.99 | 70.49 |
| L-TTA (Ours) | 92.70 | 89.85 | 84.38 | 81.96 | 66.21 | 60.40 | 71.28 | 64.83 | 77.88 | 76.48 |

| Method | Aircraft | | SUN397 | | DTD | | EuroSAT | | UCF101 | |
|---|---|---|---|---|---|---|---|---|---|---|
| | Acc. | Mac. | Acc. | Mac. | Acc. | Mac. | Acc. | Mac. | Acc. | Mac. |
| CLIP [ICML'21] | 14.11 | 12.88 | 54.39 | 51.57 | 39.00 | 33.36 | 26.78 | 21.83 | 60.77 | 56.57 |
| TPT [NeurIPS'22] | 18.39 | 17.55 | 56.86 | 54.00 | 41.83 | 35.73 | 29.45 | 27.87 | 62.83 | 57.89 |
| C-TPT [ICLR'24] | 16.52 | 15.05 | 53.48 | 49.46 | 36.97 | 31.61 | 26.97 | 18.28 | 58.74 | 53.63 |
| MTA [CVPR'24] | 12.30 | 11.81 | 51.12 | 50.23 | 39.22 | 26.95 | 23.62 | 20.40 | 53.26 | 50.91 |
| TDA [CVPR'24] | 18.93 | 14.23 | 60.94 | 55.91 | 42.90 | 36.01 | 32.09 | 32.00 | 67.92 | 60.17 |
| ZERO [NeurIPS'24] | 15.09 | 15.48 | 56.39 | 52.49 | 39.43 | 34.08 | 28.72 | 23.34 | 61.71 | 55.21 |
| DPE [NeurIPS'24] | 20.68 | 14.98 | 61.25 | 56.04 | 44.82 | 38.40 | 34.87 | 34.14 | 68.26 | 61.98 |
| O-TPT [CVPR'25] | 16.13 | 14.84 | 53.28 | 49.30 | 36.66 | 30.86 | 26.24 | 18.20 | 58.79 | 53.66 |
| SCAP [CVPR'25] | 16.08 | 15.01 | 58.49 | 55.88 | 43.53 | 32.34 | 32.04 | 35.84 | 61.78 | 54.86 |
| L-TTA (Ours) | 24.86 | 20.38 | 62.53 | 58.97 | 47.55 | 42.80 | 39.57 | 36.04 | 70.28 | 64.50 |

Table 37: **Detailed Accuracy on Long-tailed Corruption Benchmark of *pixelate*.** We conduct 5 runs for each experiment under each imbalance setting. All the methods are built upon **CLIP-VIT-B/16** backbone. The best results are marked in **Bold**.

| Method | Caltech | | Pets | | Cars | | Flowers | | Food101 | |
|---|---|---|---|---|---|---|---|---|---|---|
| | Acc. | Mac. | Acc. | Mac. | Acc. | Mac. | Acc. | Mac. | Acc. | Mac. |
| CLIP [ICML'21] | 56.05 | 46.02 | 34.89 | 30.86 | 3.78 | 2.73 | 22.75 | 20.66 | 16.88 | 14.83 |
| TPT [NeurIPS'22] | 55.70 | 45.89 | 35.67 | 31.41 | 3.48 | 2.07 | 22.89 | 20.96 | 18.02 | 16.04 |
| C-TPT [ICLR'24] | 56.30 | 47.61 | 36.62 | 31.40 | 3.91 | 2.65 | 24.71 | 23.44 | 16.82 | 14.88 |
| MTA [CVPR'24] | \ | \ | \ | \ | \ | \ | \ | \ | \ | \ |
| TDA [CVPR'24] | 57.64 | 46.05 | 36.09 | 32.08 | 5.87 | 3.77 | 25.46 | 22.16 | 20.90 | 18.08 |
| ZERO [NeurIPS'24] | 52.10 | 40.14 | 29.23 | 26.16 | 4.20 | 4.20 | 19.74 | 16.13 | 17.96 | 15.50 |
| DPE [NeurIPS'24] | 58.29 | 46.25 | 38.74 | 34.87 | 6.00 | 5.34 | 28.15 | 25.62 | 22.31 | 20.05 |
| O-TPT [CVPR'25] | 56.63 | 47.55 | 36.62 | 31.54 | 3.95 | 2.68 | 24.65 | 23.46 | 16.82 | 14.82 |
| SCAP [CVPR'25] | 58.00 | 48.56 | 40.05 | 35.86 | 3.82 | 3.46 | 26.68 | 24.23 | 21.47 | 20.53 |
| L-TTA (Ours) | 60.71 | 52.24 | 40.84 | 38.90 | 12.98 | 10.61 | 28.60 | 26.20 | 25.30 | 22.08 |
| Method | Aircraft | | SUN397 | | DTD | | EuroSAT | | UCF101 | |
| | Acc. | Mac. | Acc. | Mac. | Acc. | Mac. | Acc. | Mac. | Acc. | Mac. |
| CLIP [ICML'21] | 2.73 | 2.35 | 16.67 | 14.40 | 7.80 | 4.26 | 12.41 | 8.36 | 21.70 | 17.47 |
| TPT [NeurIPS'22] | 3.15 | 3.22 | 15.60 | 13.48 | 7.85 | 4.35 | 12.03 | 7.57 | 23.16 | 18.81 |
| C-TPT [ICLR'24] | 3.12 | 2.32 | 15.64 | 13.47 | 8.42 | 5.13 | 15.87 | 10.27 | 21.04 | 16.04 |
| MTA [CVPR'24] | \ | \ | \ | \ | \ | \ | \ | \ | \ | \ |
| TDA [CVPR'24] | 3.97 | 3.66 | 18.65 | 15.79 | 7.64 | 3.91 | 22.46 | 18.36 | 26.42 | 19.95 |
| ZERO [NeurIPS'24] | 2.22 | 2.01 | 13.23 | 12.78 | 7.75 | 4.57 | 15.53 | 9.63 | 23.01 | 16.19 |
| DPE [NeurIPS'24] | 4.96 | 4.54 | 20.75 | 18.14 | 9.00 | 5.50 | 21.86 | 17.49 | 28.89 | 21.82 |
| O-TPT [CVPR'25] | 3.20 | 2.34 | 15.59 | 13.42 | 8.74 | 5.39 | 14.43 | 10.09 | 20.88 | 15.94 |
| SCAP [CVPR'25] | 5.41 | 4.82 | 17.65 | 15.46 | 9.26 | 7.18 | 24.87 | 19.60 | 22.32 | 15.78 |
| L-TTA (Ours) | 11.85 | 9.61 | 26.84 | 21.41 | 16.62 | 11.36 | 33.47 | 26.29 | 32.00 | 25.65 |

Table 38: **Detailed Accuracy on Long-tailed Corruption Benchmark of *saturate*.** We conduct 5 runs for each experiment under each imbalance setting. All the methods are built upon **CLIP-VIT-B/16** backbone. The best results are marked in **Bold**.

| Method | Caltech | | Pets | | Cars | | Flowers | | Food101 | |
|---|---|---|---|---|---|---|---|---|---|---|
| | Acc. | Mac. | Acc. | Mac. | Acc. | Mac. | Acc. | Mac. | Acc. | Mac. |
| CLIP [ICML'21] | 92.88 | 90.40 | 85.27 | 85.97 | 61.34 | 61.35 | 63.79 | 60.70 | 80.97 | 80.17 |
| TPT [NeurIPS'22] | 93.52 | 90.85 | 87.09 | 87.70 | 62.54 | 62.65 | 64.42 | 60.92 | 81.15 | 80.40 |
| C-TPT [ICLR'24] | 93.43 | 90.73 | 84.99 | 86.17 | 61.56 | 61.37 | 64.34 | 61.03 | 80.00 | 79.08 |
| MTA [CVPR'24] | \ | \ | \ | \ | \ | \ | \ | \ | \ | \ |
| TDA [CVPR'24] | 94.15 | 91.15 | 88.02 | 86.68 | 67.61 | 62.87 | 70.19 | 62.79 | 84.90 | 82.54 |
| ZERO [NeurIPS'24] | 90.85 | 88.77 | 83.20 | 80.57 | 61.70 | 56.49 | 63.59 | 54.20 | 80.00 | 79.74 |
| DPE [NeurIPS'24] | 94.75 | 91.43 | 90.81 | 88.16 | 68.90 | 63.81 | 72.06 | 63.37 | 86.38 | 85.37 |
| O-TPT [CVPR'25] | 93.22 | 90.67 | 84.99 | 86.27 | 61.75 | 61.41 | 64.04 | 60.74 | 79.94 | 78.88 |
| SCAP [CVPR'25] | 94.88 | 91.53 | 86.85 | 84.65 | 66.00 | 63.33 | 68.11 | 64.13 | 80.91 | 79.20 |
| L-TTA (Ours) | 95.29 | 92.45 | 91.21 | 89.93 | 70.09 | 65.04 | 73.39 | 66.79 | 88.27 | 87.13 |
| Method | Aircraft | | SUN397 | | DTD | | EuroSAT | | UCF101 | |
| | Acc. | Mac. | Acc. | Mac. | Acc. | Mac. | Acc. | Mac. | Acc. | Mac. |
| CLIP [ICML'21] | 18.63 | 17.89 | 61.41 | 58.69 | 41.65 | 35.23 | 42.22 | 33.63 | 64.78 | 60.82 |
| TPT [NeurIPS'22] | 20.37 | 19.12 | 62.99 | 60.33 | 42.23 | 35.32 | 43.88 | 35.40 | 64.20 | 60.11 |
| C-TPT [ICLR'24] | 20.58 | 19.50 | 60.88 | 56.77 | 42.43 | 35.84 | 40.07 | 33.09 | 64.51 | 58.03 |
| MTA [CVPR'24] | \ | \ | \ | \ | \ | \ | \ | \ | \ | \ |
| TDA [CVPR'24] | 23.36 | 18.39 | 65.70 | 62.02 | 43.21 | 34.84 | 38.30 | 37.56 | 72.00 | 66.05 |
| ZERO [NeurIPS'24] | 19.61 | 18.97 | 61.75 | 59.46 | 41.20 | 36.75 | 33.12 | 27.51 | 66.00 | 62.83 |
| DPE [NeurIPS'24] | 23.73 | 18.94 | 67.68 | 63.54 | 45.38 | 36.50 | 41.50 | 37.75 | 71.62 | 65.96 |
| O-TPT [CVPR'25] | 19.95 | 19.26 | 60.81 | 56.84 | 42.75 | 36.42 | 40.58 | 33.39 | 64.67 | 58.32 |
| SCAP [CVPR'25] | 19.72 | 19.51 | 62.17 | 60.70 | 42.14 | 35.19 | 43.22 | 40.93 | 70.84 | 66.20 |
| L-TTA (Ours) | 30.42 | 28.62 | 68.52 | 67.36 | 48.90 | 37.97 | 46.44 | 43.49 | 74.52 | 68.36 |

Table 39: **Detailed Accuracy on Long-tailed Corruption Benchmark of *elastic transform*.** We conduct 5 runs for each experiment under each imbalance setting. All the methods are built upon **CLIP-VIT-B/16** backbone. The best results are marked in **Bold**.

| Method | Caltech | | Pets | | Cars | | Flowers | | Food101 | |
|---|---|---|---|---|---|---|---|---|---|---|
| | Acc. | Mac. | Acc. | Mac. | Acc. | Mac. | Acc. | Mac. | Acc. | Mac. |
| CLIP [ICML'21] | 92.88 | 90.40 | 85.27 | 85.97 | 61.34 | 61.35 | 63.79 | 60.70 | 80.97 | 80.17 |
| TPT [NeurIPS'22] | 93.52 | 90.85 | 87.09 | 87.70 | 62.54 | 62.65 | 64.42 | 60.92 | 81.15 | 80.40 |
| C-TPT [ICLR'24] | 93.43 | 90.73 | 84.99 | 86.17 | 61.56 | 61.37 | 64.34 | 61.03 | 80.00 | 79.08 |
| MTA [CVPR'24] | \ | \ | \ | \ | \ | \ | \ | \ | \ | \ |
| TDA [CVPR'24] | 94.15 | 91.15 | 88.02 | 86.68 | 67.61 | 62.87 | 70.19 | 62.79 | 84.90 | 82.54 |
| ZERO [NeurIPS'24] | 90.85 | 88.77 | 83.20 | 80.57 | 61.70 | 56.49 | 63.59 | 54.20 | 80.00 | 79.74 |
| DPE [NeurIPS'24] | 94.75 | 91.43 | 90.81 | 88.16 | 68.90 | 63.81 | 72.06 | 63.37 | 86.38 | 85.37 |
| O-TPT [CVPR'25] | 93.22 | 90.67 | 84.99 | 86.27 | 61.75 | 61.41 | 64.04 | 60.74 | 79.94 | 78.88 |
| SCAP [CVPR'25] | 94.88 | 91.53 | 86.85 | 84.65 | 66.00 | 63.33 | 68.11 | 64.13 | 80.91 | 79.20 |
| L-TTA (Ours) | 95.29 | 92.45 | 91.21 | 89.93 | 70.09 | 65.04 | 73.39 | 66.79 | 88.27 | 87.13 |

| Method | Aircraft | | SUN397 | | DTD | | EuroSAT | | UCF101 | |
|---|---|---|---|---|---|---|---|---|---|---|
| | Acc. | Mac. | Acc. | Mac. | Acc. | Mac. | Acc. | Mac. | Acc. | Mac. |
| CLIP [ICML'21] | 18.63 | 17.89 | 61.41 | 58.69 | 41.65 | 35.23 | 42.22 | 33.63 | 64.78 | 60.82 |
| TPT [NeurIPS'22] | 20.37 | 19.12 | 62.99 | 60.33 | 42.23 | 35.32 | 43.88 | 35.40 | 64.20 | 60.11 |
| C-TPT [ICLR'24] | 20.58 | 19.50 | 60.88 | 56.77 | 42.43 | 35.84 | 40.07 | 33.09 | 64.51 | 58.03 |
| MTA [CVPR'24] | \ | \ | \ | \ | \ | \ | \ | \ | \ | \ |
| TDA [CVPR'24] | 23.36 | 18.39 | 65.70 | 62.02 | 43.21 | 34.84 | 38.30 | 37.56 | 72.00 | 66.05 |
| ZERO [NeurIPS'24] | 19.61 | 18.97 | 61.75 | 59.46 | 41.20 | 36.75 | 33.12 | 27.51 | 66.00 | 62.83 |
| DPE [NeurIPS'24] | 23.73 | 18.94 | 67.68 | 63.54 | 45.38 | 36.50 | 41.50 | 37.75 | 71.62 | 65.96 |
| O-TPT [CVPR'25] | 19.95 | 19.26 | 60.81 | 56.84 | 42.75 | 36.42 | 40.58 | 33.39 | 64.67 | 58.32 |
| SCAP [CVPR'25] | 19.72 | 19.51 | 62.17 | 60.70 | 42.14 | 35.19 | 43.22 | 40.93 | 70.84 | 66.20 |
| L-TTA (Ours) | 30.42 | 28.62 | 68.52 | 67.36 | 48.90 | 37.97 | 46.44 | 43.49 | 74.52 | 68.36 |

Table 40: **Detailed Accuracy on 10 datasets of TPT.** The best results are marked in **Bold**.

| Method | Caltech | | Pets | | Cars | | Flowers | | Food101 | |
|---|---|---|---|---|---|---|---|---|---|---|
| VIT/L-14 | 93.02 | 91.56 | 88.19 | 87.89 | 73.99 | 74.32 | 76.45 | 73.93 | 87.69 | 87.32 |
| VIT-H | 93.06 | 91.82 | 88.47 | 87.24 | 74.08 | 74.37 | 76.72 | 72.58 | 88.87 | 87.48 |
| SIGLIP | 93.51 | 92.07 | 90.28 | 88.66 | 76.62 | 70.05 | 76.55 | 72.80 | 87.79 | 87.19 |
| META-Big | \ | \ | \ | \ | \ | \ | \ | \ | \ | \ |
| Method | Aircraft | | SUN397 | | DTD | | EuroSAT | | UCF101 | |
| VIT/L-14 | 21.75 | 23.10 | 66.64 | 63.41 | 51.17 | 44.88 | 39.22 | 45.09 | 73.46 | 69.34 |
| VIT-H | 22.13 | 25.18 | 64.51 | 63.64 | 53.23 | 45.49 | 40.60 | 44.94 | 74.00 | 71.73 |
| SIGLIP | 22.94 | 20.29 | 64.97 | 62.18 | 53.37 | 45.28 | 40.78 | 46.91 | 74.09 | 70.15 |
| META-Big | \ | \ | \ | \ | \ | \ | \ | \ | \ | \ |

Table 41: **Detailed Accuracy on 10 datasets of TDA.** The best results are marked in **Bold**.

| Method | Caltech | | Pets | | Cars | | Flowers | | Food101 | |
|---|---|---|---|---|---|---|---|---|---|---|
| VIT/L-14 | 92.89 | 91.96 | 89.64 | 89.33 | 76.85 | 72.61 | 76.86 | 73.01 | 87.92 | 86.18 |
| VIT-H | 94.23 | 92.72 | 91.17 | 89.49 | 82.82 | 67.84 | 78.56 | 74.64 | 88.66 | 83.92 |
| SIGLIP | 92.72 | 91.78 | 90.69 | 89.85 | 88.70 | 86.71 | 82.68 | 80.51 | 87.31 | 85.91 |
| META-Big | 92.84 | 92.14 | 90.76 | 90.11 | 86.65 | 84.13 | 81.14 | 76.56 | 86.32 | 84.76 |
| Method | Aircraft | | SUN397 | | DTD | | EuroSAT | | UCF101 | |
| VIT/L-14 | 27.53 | 23.94 | 67.41 | 63.17 | 49.42 | 44.43 | 52.93 | 54.69 | 76.09 | 70.22 |
| VIT-H | 33.61 | 27.52 | 66.43 | 62.40 | 54.77 | 50.54 | 52.14 | 53.25 | 78.84 | 73.26 |
| SIGLIP | 42.12 | 39.51 | 67.34 | 64.10 | 59.59 | 55.17 | 35.56 | 37.22 | 77.31 | 73.84 |
| META-Big | 44.53 | 40.08 | 70.59 | 55.88 | 60.05 | 55.88 | 63.67 | 61.14 | 81.02 | 77.13 |

Table 42: **Detailed Accuracy on 10 datasets of DPE.** The best results are marked in **Bold**.

| Method | Caltech | | Pets | | Cars | | Flowers | | Food101 | |
|---|---|---|---|---|---|---|---|---|---|---|
| VIT/L-14 | 92.51 | 91.33 | 89.80 | 89.05 | 77.16 | 73.29 | 77.29 | 74.17 | 85.81 | 84.75 |
| VIT-H | 94.72 | 93.18 | 91.82 | 89.27 | 83.64 | 69.00 | 79.03 | 85.28 | 87.06 | 83.05 |
| SIGLIP | 92.83 | 92.27 | 91.21 | 90.42 | 87.35 | 85.01 | 83.05 | 81.13 | 88.30 | 85.77 |
| META-Big | 92.88 | 92.10 | 90.72 | 90.30 | 87.40 | 85.42 | 83.46 | 81.07 | 88.70 | 87.58 |
| Method | Aircraft | | SUN397 | | DTD | | EuroSAT | | UCF101 | |
| VIT/L-14 | 22.14 | 20.34 | 72.28 | 67.92 | 60.90 | 51.77 | 56.41 | 47.84 | 75.95 | 70.88 |
| VIT-H | 34.05 | 29.44 | 70.29 | 70.38 | 58.94 | 51.62 | 53.36 | 47.17 | 77.06 | 72.31 |
| SIGLIP | 42.23 | 28.48 | 72.36 | 68.35 | 61.62 | 55.92 | 56.69 | 50.78 | 79.34 | 74.15 |
| META-Big | 44.04 | 42.08 | 73.63 | 59.51 | 61.93 | 55.54 | 63.77 | 62.02 | 80.86 | 77.57 |

Table 43: **Detailed Accuracy on 10 datasets of SCAP**. The best results are marked in **Bold**.

| Method | Caltech | | Pets | | Cars | | Flowers | | Food101 | |
|---|---|---|---|---|---|---|---|---|---|---|
| VIT/L-14 | 92.91 | 90.87 | 90.85 | 89.63 | 80.62 | 78.17 | 78.60 | 75.05 | 89.15 | 87.63 |
| VIT-H | 93.02 | 91.86 | 92.10 | 90.22 | 81.88 | 80.32 | 79.21 | 77.15 | 88.13 | 86.72 |
| SIGLIP | 92.74 | 91.21 | 92.58 | 90.10 | 81.95 | 80.47 | 81.70 | 78.52 | 90.46 | 88.35 |
| META-Big | \ | \ | \ | \ | \ | \ | \ | \ | \ | \ |
| Method | Aircraft | | SUN397 | | DTD | | EuroSAT | | UCF101 | |
| VIT/L-14 | 16.76 | 24.31 | 74.92 | 68.62 | 53.91 | 46.59 | 46.63 | 49.66 | 72.05 | 66.83 |
| VIT-H | 26.98 | 26.74 | 76.56 | 70.09 | 53.25 | 45.07 | 45.24 | 48.78 | 72.84 | 68.46 |
| SIGLIP | 43.75 | 36.48 | 63.85 | 60.94 | 50.71 | 45.50 | 48.44 | 40.52 | 70.78 | 66.68 |
| META-Big | \ | \ | \ | \ | \ | \ | \ | \ | \ | \ |

Table 44: **Detailed Accuracy on 10 datasets of L-TTA**. The best results are marked in **Bold**.

| Method | Caltech | | Pets | | Cars | | Flowers | | Food101 | |
|---|---|---|---|---|---|---|---|---|---|---|
| VIT/L-14 | 93.43 | 92.67 | 90.09 | 89.46 | 78.29 | 77.64 | 81.85 | 78.93 | 88.57 | 87.67 |
| VIT-H | 94.62 | 93.46 | 91.30 | 88.20 | 83.74 | 82.09 | 80.77 | 77.66 | 88.94 | 88.41 |
| SIGLIP | 94.51 | 93.62 | 93.18 | 90.93 | 85.26 | 86.00 | 82.70 | 78.81 | 89.39 | 87.62 |
| META-Big | 94.04 | 93.20 | 93.00 | 90.55 | 88.30 | 87.72 | 85.46 | 81.90 | 90.40 | 89.42 |
| Method | Aircraft | | SUN397 | | DTD | | EuroSAT | | UCF101 | |
| VIT/L-14 | 28.51 | 27.66 | 73.82 | 69.86 | 61.60 | 54.26 | 57.93 | 51.40 | 76.61 | 71.94 |
| VIT-H | 35.69 | 32.33 | 73.18 | 71.54 | 61.66 | 54.42 | 57.48 | 52.54 | 79.36 | 73.02 |
| SIGLIP | 46.93 | 42.43 | 74.69 | 61.61 | 62.53 | 60.42 | 58.34 | 52.21 | 79.92 | 74.07 |
| META-Big | 44.49 | 42.80 | 73.74 | 56.43 | 62.49 | 59.46 | 65.81 | 64.40 | 81.37 | 78.73 |

Table 45: **Results on Balanced Datasets.**. The best results are in **Bold**.

| Method | Caltech | | Pets | | Cars | | Flowers | | Food101 | |
|---|---|---|---|---|---|---|---|---|---|---|
| | Acc. | Mac. | Acc. | Mac. | Acc. | Mac. | Acc. | Mac. | Acc. | Mac. |
| CLIP | 94.28 | 92.26 | 88.14 | 86.80 | 65.30 | 63.5 | 67.28 | 62.39 | 83.8 | 83.72 |
| TPT | 94.9 | 92.13 | 88.57 | 87.04 | 66.49 | 64.86 | 69.32 | 64.65 | 84.81 | 84.74 |
| C-TPT | 94.27 | 91.52 | 87.76 | 86.63 | 65.77 | 63.84 | 68.09 | 65.07 | 82.6 | 82.5 |
| MTA | 94.3 | 91.94 | 87.26 | 86.54 | 65.29 | 64.68 | 66.27 | 62.8 | 85.38 | 85.11 |
| TDA | 95 | 92.08 | 89.42 | 89.03 | 68.98 | 67.25 | 72.38 | 65.91 | 86.18 | 86.07 |
| ZERO | 94.02 | 91.33 | 87.24 | 85.91 | 65.75 | 63.68 | 66.58 | 60.53 | 85.26 | 85.12 |
| RLCF | 95.08 | 92.48 | 88.98 | 86.23 | 66.46 | 63.95 | 64.8 | 60.41 | 85.1 | 84.92 |
| DPE | 95.05 | 92.1 | 91.76 | 87.45 | 67.14 | 65.49 | 74.67 | 68.85 | 86.42 | 86.4 |
| WATT | 94.91 | 92.19 | 87.33 | 86.01 | 66.04 | 64.5 | 71.28 | 66.93 | 64.32 | 85.86 |
| CLIPArTT | 92.56 | 89.84 | 87.27 | 85.95 | 65.12 | 63.43 | 66.42 | 61.61 | 82 | 81.78 |
| O-TPT | 94.18 | 91.48 | 87.95 | 86.83 | 64.63 | 63.64 | 69.8 | 64.4 | 82.41 | 82.32 |
| SCAP | 92.72 | 88.67 | 88.73 | 86.35 | 65.79 | 63.53 | 70.06 | 65.04 | 86.05 | 83.29 |
| Hohenstaufen | 95.46 | 93.62 | 92.29 | 90.3 | 70.49 | 67.43 | 75.91 | 70 | 88.34 | 87.33 |
| Method | Aircraft | | SUN397 | | DTD | | EuroSAT | | UCF101 | |
| | Acc. | Mac. | Acc. | Mac. | Acc. | Mac. | Acc. | Mac. | Acc. | Mac. |
| CLIP | 23.88 | 20.73 | 62.58 | 61.62 | 44.56 | 40.69 | 41.33 | 36.03 | 67.97 | 64.9 |
| TPT | 25.45 | 22.6 | 64.97 | 63.98 | 44.72 | 41.2 | 40.58 | 35.1 | 68.37 | 65.48 |
| C-TPT | 22.89 | 19.85 | 64.58 | 62.69 | 44.85 | 41.81 | 37.68 | 35.12 | 66.76 | 62.59 |
| MTA | 24.24 | 22.71 | 63.06 | 61.37 | 44.48 | 41.25 | 44.52 | 40.06 | 68.01 | 64.05 |
| TDA | 25.32 | 22 | 67.61 | 66.14 | 45.86 | 41.31 | 60.2 | 60.11 | 70.68 | 65.18 |
| ZERO | 24.72 | 21.65 | 63.02 | 61.7 | 44.86 | 40.87 | 38.28 | 32.06 | 65.23 | 61.65 |
| RLCF | 23.22 | 21.9 | 69.58 | 65.34 | 45.02 | 41.4 | 42.66 | 40.79 | 64.97 | 60.26 |
| DPE | 29.22 | 26.89 | 69.01 | 68.88 | 48.97 | 44.04 | 56.73 | 50.25 | 70.55 | 64.71 |
| WATT | 24.13 | 22.21 | 66.14 | 64.82 | 46.56 | 42.64 | 44.9 | 38.43 | 62.92 | 59.56 |
| CLIPArTT | 23.13 | 20.21 | 65.74 | 62.58 | 46.4 | 42.18 | 42.29 | 35.03 | 61.59 | 58.27 |
| O-TPT | 23.22 | 19.24 | 64.53 | 60.55 | 46.74 | 43.42 | 38.53 | 35.76 | 66.68 | 62.57 |
| SCAP | 23.46 | 19.49 | 65.71 | 63.31 | 45.61 | 41.79 | 52.57 | 45.74 | 71.88 | 66.62 |
| Hohenstaufen | 27.22 | 28.06 | 72.75 | 70.82 | 50.87 | 45.5 | 58.81 | 48.27 | 71.85 | 68.28 |

Table 46: Introductions of datasets.

| Datasets | Training / Validation / Testing | Types | Classes | Short introduction |
|---|---|---|---|---|
| Caltech101 | 4128 / 1649 / 2465 | Objects | 100 | 101 object categories (one is background) |
| Pets | 2944 / 736 / 3669 | Fg. pets | 37 | 37 cat / dog species |
| Cars | 6509 / 1635 / 8041 | Fg. cars | 195 | Car from different angles |
| Flowers | 4093 / 1633 / 2463 | Fg. flowers | 102 | 102 flower categories |
| Food101 | 50500 / 20200 / 30300 | Fg. food | 101 | 101 food dishes |
| SUN397 | 15880 / 3970 / 19850 | Scenes | 397 | 397 scene types |
| DTD | 2820 / 1128 / 1692 | Texture | 47 | 47 visual textures |
| EuroSAT | 13500 / 5400 / 8100 | Satellie Img. | 10 | Satellite images across 10 land use classes |
| UCF101 | 7639 / 1898 / 3783 | Actions | 101 | 101 human actions |
| Aircraft | 3334 / 3333 / 3333 | Fg. aircraft | 100 | 100 aircraft models |
| ImageNet | 1.28M / - / 50000 | Objects | 1000 | Images across 1,000 classes |
| ImageNet-A | - / - / 7500 | / | 200 | Sub-ImageNet with natural adversarial noise |
| ImageNet-R | - / - / 30000 | multi-domain | 200 | Sub-ImageNet with artistic renditions |
| ImageNet-S | - / - / 50889 | Sketches | 1000 | Sub-ImageNet with shape cues only |
| ImageNet-V | - / - / 10000 | Collocation | 1000 | Sub-ImageNet with viewpoint variations |

