# OpenReview forum: "Long-tailed Test-Time Adaptation for Vision-Language Models"
_ICLR.cc/2026/Conference — ICLR 2026 Poster_

### Official Review · Reviewer_dibS · 2025-10-21

**Soundness:** 3
**Presentation:** 3
**Contribution:** 2
**Rating:** 4
**Confidence:** 4

**Summary:**

The paper targets test-time adaptation (TTA) under long-tailed (LT) test streams for vision–language models (VLMs). It proposes a three-part framework: Synergistic Prototypes (DP+EP) for representation retention, Rebalancing Shortcuts (RSs) with a clustering re-allocation loss for structural balance, and Balanced Entropy Minimization (BEM) to reduce optimization bias toward head classes.

**Strengths:**

- Addresses a practically important setting (LT streams in online TTA).

- Clear, modular design with complementary roles (representation / structure / optimization).

**Weaknesses:**

- (Major) Problem novelty/positioning. The failure modes (head-bias accumulation, tail erosion under EM) are not VLM-specific; they are general LT-TTA issues. While packaging under “VLM LT-TTA” is useful, the problem statement and the proposed mechanisms (prototypes, rebalancing, entropy shaping) appear modality-agnostic, not specific to VLM. The paper should (i) sharpen what is uniquely VLM (e.g., cross-modal drift, text prior effects) and show where L-TTA leverages that, or (ii) reframe as a general LT-TTA method and broaden evidence beyond VLMs.

- Model capacity: Results rely on low-capacity VLMs; validating on more powerful, recent VLMs (both open- and closed-source) would better establish its contribution and clarify whether gains persist or diminish with stronger backbones.

**Questions:**

- What concrete VLM-specific factors make LT-TTA harder than unimodal LT-TTA (e.g., cross-modal prototype drift, text prior effects)? Can you show cases where your gains require the VLM setup? (See Weakness 1)
- Any results with larger/stronger VLMs? (See Weakness 2)

---

> ### Author Response · Authors · 2025-11-21
> **Response to Reviewer dibS (1/3)**
>
> Thanks for your precious opinions. We sincerely appreciate you for the review and are grateful for the time you spent with our submission. We are glad for the acknowledgment that our approach substantially resolves an actual problem, and the proposed methods are complementarily functions. We wish to address your concerns by giving detailed responses to each of  your proposed weaknesses **W1**,**W2** and questions **Q1**,**Q2** as follows:
> ### **W1: The failure modes are not VLM-specific. The paper should (i) sharpen what is uniquely VLM (e.g., cross-modal drift, text prior effects) and show where L-TTA leverages that.**
> We appreciate this comment. Regarding the concerns about the fail-mode specificity of VLMs and the unique elements leveraged by L-TTA, we provide two explanations respectively:
> 1. You kindly mentioned that head-bias accumulation is not VLM-specific. We agree with this and argue that it also emerges in single-modal TTA. However, in VLMs, these issues are exaggerated and exhibit distinct challenges. Specifically, head-bias accumulation in multimodal learning is coupled with modality misalignment challenges, because adapting only one modality will continuously amplify the mismatch between the image and text spaces. Similarly, adapting the two spaces separately fails to fully resolve this gap. Precisely to address this VLM-unique head-bias accumulation, we propose the design of L-TTA, which maintains single-modal adaptation while actively sharing adaptation parameters across different modalities via RSs. This design is exclusive to VLMs rather than being universally applicable.
> 2. You also kindly noted that tail erosion is not VLM-specific. Nevertheless, it possesses distinct characteristics in VLM settings that differentiate it from unimodal TTA. In VLMs, text information itself is biased, as some class-names (e.g., cats, dogs) are naturally more concentrated, dense, and standardized in distribution than others (e.g., owls, yaks). When using Entropy Minimization (EM) with VLMs, image features inherently tend to gravitate toward classes with denser text representations. If these classes happen to be head classes, tail classes become even less likely to learn effective image-text matching, exacerbating tail erosion. To tackle this VLM-unique tail erosion, instead of directly performing EM over logits of text embeddings and individual image features, our L-TTA makes the final prediction dependent on both the Synergistic Prototypes (SyPs) and the current prediction, where SyPs compensates for tail-class semantics through the unique Exclusionary Prototypes (EPs) and effectively mitigates the VLM-specific tail erosion. This design is exclusive to VLMs rather than being universally applicable. Furthermore, the introduced BEM can be regarded as a plain substitution of EM to further address class imbalance and tail erosion.
>
> Thanks for your insightful comment again, and we've added this discussion into the introduction of the revised paper.

---

> ### Author Response · Authors · 2025-11-21
> **Response to Reviewer dibS (2/3)**
>
> ### **W2: Model capacity: Results rely on low-capacity VLMs; validating on more powerful, recent VLMs (both open- and closed-source) would better establish its contribution and clarify whether gains persist or diminish with stronger backbones.**
>
> Thank you for this suggestion. This paper originally follows previous studies and focuses on classic vision-language models like CLIP, but we completely agree that TTA methods (both ours and previous SOTAs) should be tested on broader backbones. Following your suggestion, we selected four larger backbones for experiments: ViT/L-14, ViT/H-14, SigLIP-L/16 (a), and MetaCLIP-BigG (b). Here are the results averaged over 10 fine-grained datasets (detailed results on each dataset are in the **Appendix L of the revised paper**):
>
> Accuracy:
> | Backbone      | TPT   | TDA   | DPE   | SCAP  | L-TTA |
> | ------------- | ----- | ----- | ----- | ----- | ----- |
> | CLIP:ViT/L-14 | 67.16 | 69.75 | 71.03 | 69.64 | 73.07 |
> | CLIP:ViT/H-14 | 67.57 | 72.12 | 73.00 | 70.92 | 74.67 |
> | SigLIP-L/16   | 68.09 | 72.40 | 75.50 | 71.70 | 76.75 |
> | MetaCLIP-BigG | \\    | 75.76 | 76.74 | \\    | 77.91 |
>
> Macro-F1:
> | Backbone      | TPT   | TDA   | DPE   | SCAP  | L-TTA |
> | ------------- | ----- | ----- | ----- | ----- | ----- |
> | CLIP:ViT/L-14 | 66.08 | 66.95 | 67.13 | 67.74 | 70.15 |
> | CLIP:ViT/H-14 | 66.45 | 67.56 | 69.07 | 68.54 | 71.37 |
> | SigLIP-L/16   | 65.56 | 70.46 | 71.23 | 67.88 | 72.77 |
> | MetaCLIP-BigG | \\    | 71.78 | 73.32 | \\    | 74.46 |
>
> (Notes: here \ means the model fails to provide valid outputs.)
>
> The performance improvement of our model is consistent across different backbone architectures, with only minor fluctuations observed. This is because L-TTA effectively leverages inherent class-relations in each sample and actively promotes continuous image-text alignment, which are generally beneficial regardless of the underlying backbone in long-tailed test-time adaptation.
>
> We also thank the reviewer for the valuable suggestion to evaluate our method on closed-source, API-only VLMs like GPT-4V. However, performing test-time adaptation typically requires access to internal features and gradients, which is generally not feasible with such black-box API models. Nevertheless, we fully agree that this is an important and promising research direction, and several recent studies have already begun to explore this area. For instance, IPO (c) instructs an LLM using the image, prompt, and loss function to iteratively refine the prompt. This gradient-free "optimization" offers a viable pathway for scenarios where backpropagation is entirely infeasible and can enhance the interpretability of machine learning.
>
> As future work, we plan to investigate simulating the adaptation process using only API inferences. A promising idea may be maintaining and updating prototypes based on API outputs. We will include a discussion of this promising direction in the final version of the paper.
>
>
> ### **Q1: What concrete VLM-specific factors make LT-TTA harder than unimodal LT-TTA? (similar with Weakness 1)**
> Thank you for this comment. We completely agree that we should clarify how the gains of L-TTA come from the VLM setup. Considering that this question is strongly linked to Weakness 1 (W1) above, could you please refer to our rebuttal to **W1**, where we have conducted a deeper investigation into this point? Thank you for your time and patience in our rebuttal!
>
>
> ### **Q2: Any results with larger/stronger VLMs? (similar with Weakness 2)**
> Thank you for this comment. Considering that this question is strongly linked to Weakness 2 (W2) above, could you please refer to our rebuttal to **W2**, where we have conducted concrete experiments regarding this point? Thank you for your time and patience in our rebuttal!

---

> ### Author Response · Authors · 2025-11-21
> **Response to Reviewer dibS (3/3)**
>
> Refs:
>
> (a): Sigmoid loss for language image pre-training, ICCV 2023
>
> (b): Demystifying CLIP Data, ICLR 2024
>
> (c) Ipo: Interpretable prompt optimization for vision-language models. NeurlPS 2024

---

> ### Author Response · Authors · 2025-11-25
> **Looking forward to your feedback**
>
> Dear Reviewer dibS:
>
> We greatly appreciate your professional feedback, which is invaluable to our research. We would be grateful if you could provide your insights at your convenience to help us further refine this work. We look forward to your valuable comments and are willing to discuss any aspects in more detail. Thank you!
>
> Sincerely,
>
> The Authors

---

> > ### Comment · Reviewer_dibS · 2025-11-26
> > **Response to authors’ response on W1**
> >
> > Thank you for the detailed clarification. However, I remain unconvinced because the response claims the failure modes are “exaggerated” and “distinct” in VLMs without providing **supporting evidence**.
> >
> > In particular, the paper/rebuttal does not present (i) intuitive or real failure evidence of such claim, nor (ii) quantitative diagnostics showing these effects are stronger in VLMs than in unimodal TTA. The current narrative is largely **method-and-result-centric (e.g., Fig. 1)**, but lacks an **upfront,  justification** that this constitutes a genuinely VLM-specific problem setting rather than a restatement of known LT-TTA issues.
> >
> > Absent this motivation, the contribution reads as **“a different solution to a known problem,” not a new VLM-specific problem.** Therefore, my overall assessment and rating remain unchanged.

---

> > > ### Author Response · Authors · 2025-11-29
> > > **Response to Reviewer dibS (1/2)**
> > >
> > > Dear Reviewer dibS:
> > >
> > > Thank you for your feedback. We would like to mention that the evidence for these unique issues is actually substantiated in our experimental comparisons (especially with visual-adaptation-based methods) and in the Appendix. However, they are not directly displayed in Figure 1 due to the tight rebuttal schedule, during which our attention was diverted by various complex experimental requests in your W2 and weaknesses proposed by other reviewers. As a result, we feel sorry that our response to you was confined to textual explanations, and we did not incorporate these evidences into the manuscript. Now we have incorporated these revisions into our new manuscript. We sincerely apologize for this oversight again.
> > >
> > > To move beyond the simplistic **method-and-result** narrative, we have made minor adjustments to Figure 1, where we concisely summarize the evidence for these issues from our subsequent experiments and the Appendix. As for the demonstration of model performance, we have placed it separately in Figure 2. This slight adjustment ensures the overall logic follows a **method-motivation-design-result** flow. We believe this revision perfectly addresses your concerns regarding our narrative structure. Thank you for your valuable feedback!
> > >
> > > Furthermore, we believe there're still potential mis-understandings concerning our VLM-specific failure modes. We introduce them again as the following:
> > >
> > > ## **Our task**
> > > This work focuses on the task of Test-time Adaptation (TTA) for Vision-Language Models (VLMs) under a Long-Tailed (LT) setting, as illustrated in Figure 1(a).
> > >
> > > ## **Unique failure modes of existing methods on our task (Our motivation):**
> > >
> > > Failure Mode 1: Text-induced Tail Erosion; Failure Mode 2: Modality-bias Amplification.
> > >
> > > 1. Text-induced Tail Erosion: This failure mode means that the inherent properties of VLMs can fundamentally exacerbate the class imbalance problem. In panel b.1 of Figure 1, we demonstrate the existence of rich classes (defined as classes that achieve higher accuracy regardless of whether they are head or tail classes). Specifically, we first evaluated the performance of state-of-the-art method SCAP(a) on ImageNet under 50 different random seeds and computed the per-class accuracy. We found that categories like ''goldfish'' consistently achieved nearly 100% accuracy, while categories like ''night snake'' had accuracy close to 0. This indicates that VLMs intrinsically exhibit preference towards certain categories. We name these preferred classes as rich classes and quantify each category's richness based on this experimental result. In another experiment shown in b.1, we varied the alignment between rich classes and head classes (by adjusting the Earth Mover's Distance between richness and class cardinality distribution). We observed the highest model accuracy when rich classes largely overlapped with head classes, with performance declining as this alignment weakened. This demonstrates the impact of rich classes on model performance and underscores the limitation of relying solely on text as the classification benchmark, necessitating auxiliary information.
> > >
> > >
> > > 2. Modality-bias Amplification: This failure mode aims to show that unimodal LT-TTA methods are difficult to transfer directly to the LT-TTA task on VLMs. Specifically, in panel b.2 of Figure 1, we show the per-class accuracy on ImageNet when migrating the classic unimodal LT-TTA method SAR(b) from a ViT model to its corresponding VLM version. In comparison, we observed a significant accuracy drop for the VLM on medium and tail classes, indicating that adaptation confined to a unimodal space is insufficient for VLMs. Furthermore, we plotted the trend of macro-F1 over the progressing test data stream across 50 seeds. We found that the multimodal model not only underperformed the unimodal model overall but also exhibited larger standard deviation and more pronounced performance degradation later in the stream. This highlights the unique instability introduced by the additional modality in TTA, which unimodal TTA fails to adequately address.

---

> > > ### Author Response · Authors · 2025-11-29
> > > **Response to Reviewer dibS (2/2)**
> > >
> > > ## **Our L-TTA's unique design addressing these failure modes:**
> > > 1. To address the first and second issues, we introduce our Synergistic Prototypes (SyPs). Specifically, considering that rich classes in the text embeddings can worsen the long-tailed problem, we go beyond matching each output image feature to text vectors by introducing SyPs to mitigate the influence of pure text. To ensure SyPs possess sufficient balancing capability without being affected by the accumulated long-tailed distribution, we design two synergistic prototypes: Deterministic Prototypes (DPs) and Exclusionary Prototypes (EPs). These are responsible for preserving the core discriminative semantics of the class and the semantics least relevant to the class, respectively. Upon obtaining an input sample's embedding, we add it to the DP of the class with the highest prediction confidence and to the EP of the class with the lowest prediction confidence. Consequently, head classes naturally accumulate richer DPs, while tail classes concurrently accumulate richer EPs. During final prediction, we synergistically consider both prototypes, ensuring the SyPs themselves can balance the long-tailed problem while diluting the bias from the text modality.
> > >
> > > 2. To further address the second issue, we introduce our Rebalancing Shortcuts (RSs). Specifically, considering the poor generalization of traditional unimodal LT-TTA baselines to VLMs, RSs are built upon the SyPs (which cache multimodal information) and aim to progressively establish a new space adapted to the test data distribution in a learnable manner. To ensure this learned space maintains balanced class awareness, RSs dynamically adapt each sample based on the similarity between the current input and a set of hyper-class vectors. Subsequently, we employ a balancing strategy for Mixture-of-Experts (MoE) to balances the contributions of these vectors, ensuring they capture uniform semantic information across different classes. This strategy is implemented by encouraging the activation distribution of each vector to approximate a uniform distribution.
> > >
> > >
> > > 3. Our third design is a simple, lightweight EM variant termed BEM, which directly incorporates a dynamic class prior at the logit level, functioning independently for class rebalancing. Although not directly inspired by our two failure modes, it serves as a plug-and-play module compatible with any long-tailed TTA environment.
> > >
> > >
> > >
> > > ## **Our Results:**
> > >
> > > We present the results of our L-TTA in Figure 2. First, we use t-SNE to visualize the classification results on tail classes for the strongest baseline SCAP and our L-TTA, quantitatively showing that our model achieves significantly better separation on tail classes. Second, we report the macro-F1 score of various models under different imbalance ratios, demonstrating the strong robustness of L-TTA across all settings. Finally, the experimental results in Tables 1, 2, 3, and 4 also confirm the superior performance of our model.
> > >
> > >
> > > We sincerely thank you for your comments and again apologize for not fully addressing your concerns in our previous rebuttal, as we relied on textual explanations and overlooked a complete narrative. We believe this revision thoroughly and conclusively resolves your doubts. We deeply appreciate the crucial role your expertise has played in the continuous improvement and enhancement of our work.
> > >
> > >
> > > Refs:
> > >
> > > (a): SCAP: Transductive Test-Time Adaptation via Supportive Clique-based Attribute Prompting, CVPR 2025
> > >
> > > (b): Towards stable test-time adaptation in dynamic wild world, ICLR 2023

---

### Official Review · Reviewer_QBXH · 2025-11-01

**Soundness:** 3
**Presentation:** 3
**Contribution:** 3
**Rating:** 6
**Confidence:** 4

**Summary:**

The paper introduces Long-Tailed Test-Time Adaptation (L-TTA), a new paradigm that explicitly addresses class imbalance during test-time adaptation of vision-language models. L-TTA focuses on real-world settings where data follow long-tailed distributions, and it consists of three key components: Synergistic Prototypes, Rebalancing Shortcuts, and Balanced Entropy Minimization. Extensive experiments on 15 long-tailed datasets show consistent gains in both accuracy and macro-F1, particularly for tail class.

**Strengths:**

S1. **Realistic problem formulation.**
The paper is the first to define long-tailed test-time adaptation explicitly, bridging TTA and class-imbalance research.

S2. **Theoretical support.**
The authors provide a proposition showing that BEM reduces the optimization gap between head and tail classes, giving analytical depth to their design.

S3. **Comprehensive experiments.**
Evaluations across 15 datasets and diverse settings (OOD, domain shift, noise) demonstrate robustness and clear macro-F1 improvements, validating that the method indeed benefits rare classes.

**Weaknesses:**

W1. **Limited methodological novelty.**
While the paper defines a new and realistic long-tailed test-time adaptation setting, most of its core components --dual prototypes, rebalancing modules, and entropy regularization -- are adapted from existing long-tailed learning techniques. Therefore, the methodological contribution appears somewhat incremental, as it primarily repurposes well-known ideas rather than introducing fundamentally new mechanisms. Nevertheless, the adaptation of these ideas to a label-free, online TTA scenario is well-executed and empirically validated.

W2. **Computational overhead not fully analyzed.**
In a test-time adaptation (TTA) environment, it is crucial to evaluate computational overhead, as most prior works explicitly report their runtime and justify that their methods can operate under realistic TTA constraints. To the best of my knowledge, the proposed method introduces additional parameters compared to existing baselines due to the inclusion of the RS module and dual prototypes. Therefore, a detailed analysis of runtime and memory consumption is required.

W3. **Hyperparameter sensitivity.**
The method relies on several new hyperparameters (i.e., $\beta$ in BEM, $\tau$ for temperature, $\eta$ for CRA weighting). The paper gives fixed settings but limited analysis on sensitivity or tuning difficulty.

W4. **Intuition of synergy of SyPs and RS.**
The authors show the synergy of SyPs and RS with empirical results, but I cannot catch the intuition why those two modules can have positive synergy. Deeper explanation could make reviewers understand easily.

**Questions:**

Q1. In realistic streaming scenarios, the head/tail imbalance may continuously shift over time. For instance, some tail classes may gradually become head classes as data distribution evolves. How would L-TTA behave under such dynamic imbalance transitions? Would its components (e.g., BEM or RSs) still maintain stability and prevent bias accumulation when the head and tail relationship itself changes over time?

Q2. The proposed Synergistic Prototypes (SyPs) employ two prototypes per class to enhance representation learning for tail classes.
Have the authors considered extending this idea to an adaptive or variable number of prototypes per class, which could further capture intra-class diversity and prevent overfitting to limited modes within each class?

---

> ### Author Response · Authors · 2025-11-21
> **Response to Reviewer QBXH (1/3)**
>
> Thanks for your precious opinions. We sincerely appreciate your review and are grateful for the time you have spent on our submission. We are glad that you acknowledge our approach substantially addresses a realistic problem and that the experiments and ablation studies are quite thorough. We wish to address your concerns by giving detailed responses to each of your  proposed weaknesses **W1**,**W2**,**W3**,**W4** and questions **Q1**,**Q2** as follows:
> ### **W1: Limited methodological novelty.**
> We thank the reviewer for this insightful comment. We acknowledge that some designs of our model are adapted from long-tailed learning. However, our work addresses a fundamentally distinct problem setting where the core assumptions and techniques of existing LT methods are inapplicable.
> 1. The proposed Long-Tailed TTA task on VLMs is fundamentally different from standard long-tailed learning. Standard LT methods operate on labeled, static datasets with known class distributions. In contrast, our LT TTA scenario is label-free, online, and cross-modal, which introduces the following unique challenges: (a) unknown and shifting class frequencies, (b) accumulation of pseudo-label errors over time, and (c) novel failure modes stemming from vision-language misalignment (e.g., cross-modal drift, effects of textual prior). These constraints render existing LT methods inoperable.
>
> 2. Our modules are specifically designed for our proposed LT+TTA with VLMs. Synergistic Prototypes (SyPs) jointly update two prototypes under the constraints of no labels and an online setting, which is impossible in conventional LT balancing. Rebalancing Shortcuts (RSs) perform cluster-driven reallocation without class labels, addressing tail erosion in a streaming context. Balanced Entropy Minimization (BEM) is specifically formulated to correct head dominance caused by EM in VLMs. This dilemma is also not present in unimodal LT learning.
>
> In summary, the core novelty lies in the holistic solution crafted for a previously unexplored and challenging problem. While some designs may share a high-level inspiration with LT learning, our substantial redesign and synergistic integration under the stringent constraints of label-free, online, cross-modal adaptation constitute a distinct and significant contribution.
>
> We appreciate the reviewer's acknowledgment that our model is well-executed and empirically sound. We will ensure these critical distinctions are emphasized more clearly in the final version.
>
> ### **W2: Computational overhead not fully analyzed.**
> Thank you for this valuable comment. We completely agree that a more detailed analysis of efficiency is highly necessary. We have expanded our efficiency study to include more baselines: Reinforcement Learning (RL)-based (i.e., RLCF), Training-free (i.e., ZERO), and Batch Normalization (BN)-based (i.e., CLIPArTT, WATT). Also, as suggested by Reviewer 42WD, we have included other corruption types besides Gaussian noise in our experiments. These experiments are renamed to a unified Corruption Benchmark (CB). The results are available in the **Table 5 of the revised paper**. For your convenience, we also include the results here:
>
> | Methods      | TPT    | C-TPT  | MTA   | TDA   | ZERO  | RLCF   | WATT      | DPE   | CLIPArTT | O-TPT  | SCAP  | L-TTA     |
> | ------------ | ------ | ------ | ----- | ----- | ----- | ------ | --------- | ----- | -------- | ------ | ----- | --------- |
> | Time         | 3.80h  | 3.80h  | 1.87h | 0.91h | 0.86h | 8.3h   | 27.7h     | 1.38h | 6.42h    | 3.81h  | 2.96h | 1.45h     |
> | Memory       | 17.94G | 17.94G | 1.29G | 0.89G | 1.68G | 19.84G | 1.54G(xN) | 1.81G | 1.71G    | 17.94G | 1.97G | 1.89G     |
> | HM on LT-CDB | 61.12  | 61.06  | 61.75 | 64.51 | 60.42 | 59.71  | 62.07     | 66.31 | 60.54    | 61.39  | 63.31 | 67.20     |
> | HM on LT-CB  | 40.04  | 38.57  | \\    | 41.04 | 38.04 | -      | \-        | 41.93 | \-       | 38.05  | 40.74 | **46.08** |
>
> Notes:
>
> (a) "\" indicates that the model fails to provide valid outputs;
>
> (b) "-" indicates that  the model did not finish within our rebuttal time budget, and we plan to include the results in the final version.
>
> Our new findings are: 1. While training-free methods (e.g., ZERO) indeed achieve the fastest inference, they exhibit severely limited performance in the LT-TTA setting. Furthermore, they consistently fail in more challenging scenarios, such as the proposed corruption benchmark, since they rely heavily on high-quality data distributions. 2. In terms of efficiency, L-TTA requires less inference time than RL-based or BN-based methods, though it is naturally slower than training-free approaches. Most importantly, L-TTA demonstrates considerable robustness to both long-tailed and corrupted data distributions, substantially outperforming all compared efficiency-oriented baselines. A comprehensive discussion of these efficiency-accuracy trade-offs is provided in the efficiency study of the revised paper.

---

> ### Author Response · Authors · 2025-11-21
> **Response to Reviewer QBXH (2/3)**
>
> ### **W3: Hyperparameter sensitivity.**
> Thank you for raising this concern.
>
> 1. Regarding the issue of limited analysis on sensitivity, we believe part of the issue may stem from how the ablation and sensitivity studies are organized in the paper. In fact, Section 4.2 evaluates the key hyperparameters of L-TTA, including prototype update weights, reallocation strength, and entropy balancing coefficients. Across datasets, we observe high robustness: varying each hyperparameter within a broad range (e.g., 0.5–2× the default value) leads to only minor changes in accuracy, and the overall head/medium/tail improvement trend remains unchanged. To avoid misunderstanding, we have revised the section title to "Ablation Study and Sensitivity Analysis" so that these results are easier to locate.
>
> 2. Regarding the issue of limited analysis of tuning difficulty, we would like to clarify that tuning these hyperparameters does not incur substantial computational concerns, as our model is robust to hyperparameter changes and naturally exhibits a broad and flat global minimum. As can be observed in Figure 3, adjusting hyperparameters $\eta$, $K$, and $\beta$ generally results in moderate changes within a considerable margin; for $\lambda_1$ and $\lambda_2$, which control the contributions of prototype-based prediction and the current prediction, keeping their values larger than (4,4) results in a performance change of less than 0.3%. These results demonstrate that L-TTA does not raise substantive concerns regarding tuning difficulty.
>
> Furthermore, in Table 7 of the revised paper, we have added a new experiment comparing the performance of using a fixed hyperparameter setting versus fine-tuning hyperparameters for each dataset. The results show that using one identical set of hyperparameters does not incur substantial performance fluctuation, and we hope this will address your concern. We appreciate your comment again and will ensure the sensitivity discussion is made more explicit and visible.
>
> ### **W4: Intuition of synergy of SyPs and RS.**
>
> Thanks for this comment!  We completely agree that a clearer explanation of the synergy between SyPs and RS is of necessity, beyond the empirical evidence. The interaction of these two modules is mutually reinforcing. Concretely:
>
> 1. SyPs stabilize the representation space on which RSs rely. SyPs maintain dual prototypes per class to prevent feature drift and tail collapse during online adaptation. This produces cleaner, more reliable class-wise anchors. RSs depend on such stable prototypes to compute cluster-level reallocations: without SyPs, RSs would operate on noisy or drifting class centers and tend to over-correct.
>
> 2. RSs achieve further re-balancing, which keeps SyPs effective. Even if SyPs maintain relatively rebalanced representations through their updating scheme, long-tailed streams still cause structural imbalance. This is because SyPs are training-free and cannot be proactively refined during training. RSs counteract this by reassigning cluster mass in a **trainable** manner, ensuring all classes receive sufficient and effective support.
>
> In conclusion, SyPs and RSs work synergistically. If we employ SyPs alone, the head classes still dominate the structure since SyPs are training-free and cannot be proactively refined by training. If we employ RSs alone, reallocation is possible, but unstable prototypes lead to noisy or incorrect reallocations. When combined, RSs ensure that SyPs can be further refined in a trainable manner, and SyPs provide stable anchors that make RSs precise and low-variance. SyPs and RSs address distinct but tightly coupled long-tailed failure modes in VLMs, and each module strengthens the other. We have incorporated this explanation into the revised paper to make the intuition clearer. Thank you for your comment again.
>
> ### **Q1: How would L-TTA behave under dynamic imbalance transitions?**
> We thank the reviewer for sharing this insight. We agree that in realistic streaming scenarios, class frequencies may evolve over time, and a class that is initially rare may later become frequent (or vice versa). Importantly, our method is designed to operate under continuously changing class-frequency conditions. Thus, if a tail class gradually becomes a head class, SyPs, BEM, and RSs naturally adapt to the continually changing statistics without inducing instability or potential degradation. We can further explain the robustness of our modules to this setting as follows (please refer to the next window):

---

> > ### Author Response · Authors · 2025-11-21
> > **Response to Reviewer QBXH (3/3)**
> >
> > 1. Synergistic Prototypes (SyPs) handle long-tailed drift without relying on class orders. This is because the refined update in SyPs is sample-level, i.e., for each sample, we update its specific features into its own Deterministic Prototype (DP) and the least-relevant features into other classes' Exclusionary Prototype (EP). This means that head classes indeed receive more frequent DP updates, but tail classes conversely benefit from increased EP updates. If head and tail classes alternate during this process, their DP and EP will naturally adjust their absorption and update patterns according to the data stream. In other words, our SyPs establish update parity between head and tail classes.
> >
> > 2. Rebalancing Shortcuts (RSs) are inherently stable under dynamic imbalance because they detect imbalance implicitly via cluster-level reallocation signals (e.g., concentration of assignments), rather than via fixed head/tail annotations. If a class becomes more frequent, RSs naturally reduce its reallocation pressure, and vice versa. This makes the mechanism adaptive to evolving class-frequency patterns.
> >
> > 3. For Balanced Entropy Minimization (BEM), its correction strength scales with the current pseudo-label bias. BEM monitors optimization bias through entropy–confidence dynamics. When a class gains more evidence over time, BEM reduces its penalization; when a previously common class becomes rare, BEM increases its prior effect. All these operations are dynamically adapted along with the data stream itself; therefore, BEM stabilizes adaptation regardless of temporal shifts in head/tail status.
> >
> > As further validation of our analysis, we demonstrate our model's robustness to class order change in Table 8 of the revised paper, where we alter a coefficient $\epsilon$ that controls the sampling probability of head/tail-class samples and report the performance on two datasets (ImageNet and OxfordFlowers, representing large and small datasets). We also show the results here:
> >
> > ImageNet:
> > | $\epsilon$ | 0     | 1/3  | 2/3  | 1     |
> > | --------- | ----- | ----- | ----- | ----- |
> > | Acc.      | 71.30 | 71.34 | 71.47 | 71.52 |
> > | Mac.      | 65.72 | 65.76 | 65.86 | 65.93 |
> >
> > OxfordFlowers:
> > | $\epsilon$ | 0     | 1/3  | 2/3  | 1     |
> > | --------- | ----- | ----- | ----- | ----- |
> > | Acc.      | 74.60 | 74.98 | 74.95 | 75.08 |
> > | Mac.      | 68.62 | 68.85 | 68.98 | 69.01 |
> >
> > The results show that continously changed head/tail classes bring no substantial influence to our model's performance. This is because L-TTA continuously adapts to whichever class is momentarily frequent or rare; therefore, even if the head/tail status shifts over time or appears in different orders, the method simply re-equilibrates online, leading to nearly identical overall performance. Thank you for your comment again.
> >
> >
> >
> > ### **Q2: The proposed Synergistic Prototypes (SyPs) employ two prototypes per class to enhance representation learning for tail classes. Have the authors considered extending this idea to an adaptive or variable number of prototypes per class, which could further capture intra-class diversity and prevent overfitting to limited modes within each class?**
> >
> > Thank you for sharing this insight! We believe that employing an adaptable number of prototypes for each class indeed represents a promising direction. The two prototypes utilized in our Synergistic Prototypes (SyPs) represent the oppositional relationship of "belonging/non-belonging". To employ a dynamic number of prototypes, we could treat each one as an individual expert and design a dynamic expert routing mechanism to aggregate their contributions. Regarding the specifics of expert routing, we can draw significant inspiration from the Mixture-of-Experts (MoE) architecture in LLMs. By strategically adapting these concepts to the unique characteristics of Long-Tailed and VLM-based TTA, we can further enhance the robustness of TTA methods in LT settings. Furthermore, we also note that this insightful idea bears some similarity to previous literature such as AdaPrompt (a) and DynaPrompt (b), which adopt different sets of models (prompts) and dynamically combine their predictions.
> >
> > However, given the time constraints of the rebuttal period and considering this idea may properly belongs to an another independent research, we regret that we cannot provide experiments here. Nevertheless, we sincerely believe your insightful suggestion holds significant research potential within the VLM-TTA+LT domain. Thank you again for your profound and intellectual contribution!
> >
> > refs:
> >
> > (a) Robust test-time adaptation for zero-shot prompt tuning, AAAI 2024
> >
> > (b) Dynaprompt: Dynamic test-time prompt tuning, ICLR 2025

---

> ### Author Response · Authors · 2025-11-25
> **Looking forward to your feedback**
>
> Dear Reviewer QBXH:
>
> We greatly appreciate your professional feedback, which is invaluable to our research. We would be grateful if you could provide your insights at your convenience to help us further refine this work. We look forward to your valuable comments and are willing to discuss any aspects in more detail. Thank you!
>
> Sincerely,
>
> The Authors

---

> > ### Comment · Reviewer_QBXH · 2025-11-26
> > **Response by Reviewer QBXH**
> >
> > I appreciate the authors' efforts in addressing my concerns. Most of my concerns have been resolved by the authors' response. I acknowledge that the paper is well-organized and supports its claims with solid experimental results. However, I still believe that the proposed task (i.e., Long-tail TTA on VLMs) is a quiet incremental setup. Therefore, I have decided to maintain my current score.

---

> > > ### Author Response · Authors · 2025-11-27
> > > **Thank you for your positive feedback!**
> > >
> > > Dear Reviewer QBXH:
> > >
> > > We sincerely thank you for your positive feedback and for recognizing the value of our work. We truly appreciate the time and effort you have dedicated to reviewing our paper. Regarding your comment that the proposed LT+TTA on VLMs is incremental, we would like to respectfully clarify that our work aims to address a critical and relatively under-explored research gap. We believe this direction holds significant practical relevance for real-world applications, and hope this additional context allows for a more comprehensive assessment of our contribution. Once again, we are grateful for your time and insightful comments!
> > >
> > > Sincerely,
> > >
> > > The Authors

---

### Official Review · Reviewer_42WD · 2025-11-01

**Soundness:** 3
**Presentation:** 2
**Contribution:** 3
**Rating:** 4
**Confidence:** 3

**Summary:**

This paper presents the Long-Tailed Test-Time Adaptation (L-TTA) framework, which addresses the performance degradation of vision-language models in test-time adaptation scenarios where the test distribution is long-tailed. Standard TTA methods typically amplify bias toward dominant classes, thereby harming underrepresented ones.
To solve this, they propose a complex, three-part system:

- Synergistic Prototypes (SyPs): A dual-prototype memory system to separately accumulate confident (Deterministic) and improbable (Exclusionary) features, designed to enrich tail class representations.
- Rebalancing Shortcuts (RSs): Learnable cross-attention adapters to dynamically balance the prototypes, optimized with a novel Class Re-Allocation loss.
- Balanced Entropy Minimization (BEM): A new loss function that gates the influence of class priors using prediction confidence, specifically to protect tail classes from the over-optimization of head classes.

Extensive experiments demonstrate that L-TTA achieves strong generalization and outperforms existing methods under a wide range of long-tailed test settings.

**Strengths:**

1. First work to formally study long-tailed test-time adaptation for VLMs, addressing a realistic yet overlooked deployment scenario.
2. Introduces a coherent three-part novel system (SyPs, RSs, BEM) with each component targeting a specific weakness of standard TTA under imbalance.
3. Outperforms 7 baselines across 15 datasets in accuracy and macro-F1, under different imbalance ratios. Gains are consistent on OOD, cross-domain, and noise benchmarks.
4. Achieves competitive adaptation speed without full-model optimization, making the approach practical for real-world use.
5. Provides thorough ablations and sensitivity studies showing the contribution of each component.

**Weaknesses:**

1. The method requires tuning several per-dataset hyperparameters, which contradicts with the core objective of test-time adaptation, where models are expected to generalize without dataset-specific tuning.
2. The Balanced Entropy Minimization (BEM) loss depends on access to class priors ($\pi$), which are not typically available in reality, making this assumption impractical.
3. This paper primarily emphasizes entropy-minimization, prompt-tuning, and prototype-based approaches, but it omits discussion of important emerging TTA families, such as vision-encoder adaptation methods (e.g., CLIP-ArTT[1], WATT[2]), parameter-free approaches (e.g., ZERO[3]), and reinforcement-based TTA frameworks (e.g., RLCF[4]).
4. The notation and exposition can be difficult to follow, and parts of the method are challenging to comprehend. Clarity and narrative coherence could be improved to enhance readability.

**Refs:

[1] Hakim, Gustavo A. Vargas, et al. "Clipartt: Adaptation of clip to new domains at test time." WACV'25.

[2] Osowiechi, David, et al. "WATT: Weight average test time adaptation of CLIP." NeurIPS'24.

[3] Farina, Matteo, et al. "Frustratingly easy test-time adaptation of vision-language models." NeurIPS'24.

[4] Zhao, Shuai, et al. "Test-Time Adaptation with CLIP Reward for Zero-Shot Generalization in Vision-Language Models." ICLR'24

**Questions:**

1. Does the model still outperform baselines when using a consistent set of hyperparameters across datasets? Demonstrating this would help confirm that the performance gains are not dependent on per-dataset tuning. A positive result here would increase my evaluation score.
2. How does the model perform on balanced datasets? It would be helpful to see whether the proposed design adversely affects performance when the class distribution is uniform.
3. Can the authors report performance on additional corruption types (e.g., blur, snow, JPEG compression)? I recognize the rebuttal timeline is limited, but even small-scale experiments would strengthen the evidence of robustness across diverse corruption scenarios.

Finally, I remain open to adjustments or clarifications addressing the weaknesses raised in weaknesses section.

---

> ### Author Response · Authors · 2025-11-21
> **Response to Reviewer 42WD (1/4)**
>
> Thanks for your precious opinions. We sincerely appreciate your review and are grateful for the time you have spent evaluating our submission. We are glad that you acknowledge our approach substantially addresses a realistic problem and that the experiments and ablation studies are quite thorough.  We wish to address your concerns by giving detailed responses to each of your  proposed weaknesses **W1**,**W2**,**W3**,**W4** and questions **Q1**,**Q2**,**Q3** as follows:
> ### **W1: The method requires tuning several per-dataset hyperparameters, which contradicts with the core objective of test-time adaptation, where models are expected to generalize without dataset-specific tuning.**
> Thank you for this insightful comment, which points out a fundamental tension in TTA, i.e., the requirement for complete generalization ability without dataset-specific tuning. We completely agree that TTA should achieve such a level of robustness. However, as also observed in many previous TTA studies, such as NOTE(a), DA-TTA(b), SAR(c), TDA(d), WATT(e), DPE(f), and SCAP(g), achieving stable gains on various challenging datasets often requires careful selection of a few hyperparameters. Similarly, our L-TTA also needs to fine-tune several hyperparameters for each dataset due to their varied distributions, but this is more of an incremental and optimal choice with high efficiency, rather than a required process that significantly reduces the practicality of TTA.
>
> **First**, our ablation studies have shown that our model is robust to hyperparameter changes and naturally exhibits a broad and flat global minimum. As can be observed in Figure 3, adjusting hyperparameters $\eta$, $K$, and $\beta$ generally results in moderate changes within a considerable margin; for $\lambda_1$ and $\lambda_2$, although they largely control the contributions of two sources to the prediction, keeping their values larger than (4,4) results in a performance change of less than 0.3%. These results demonstrate that L-TTA does not raise substantive concerns regarding tuning difficulty.
>
> **Second**, we conducted an additional experiment to further demonstrate hyperparameter robustness by comparing the performance of using a fixed hyperparameter setting ($$) versus settings tuned for each dataset across all 15 datasets (PHFT below stands for Per-dataset Hyperparameter Fine-Tuning; Accuracy and Macro-F1 are reported separately due to the constraints of rendering complex tables):
>
> **Accuracy:**
>
> | Method  | Caltech | Pets  | Cars  | Flowers | Food101 | Aircraft | SUN397 | DTD   | EuroSAT | UCF101 | ImageNet-A | ImageNet-R | ImageNet-S | ImageNet-V2 | ImageNet | Avg.  |
> | ------- | ------- | ----- | ----- | ------- | ------- | -------- | ------ | ----- | ------- | ------ | ---------- | ---------- | ---------- | ----------- | -------- | ----- |
> | w/o PFT | 94.98   | 91.50 | 69.91 | 74.79   | 85.27   | 26.63    | 69.52  | 51.39 | 56.82   | 69.56  | 61.66      | 82.48      | 50.08      | 68.70       | 72.18    | 68.36 |
> | L-TTA   | 95.12   | 91.62 | 70.33 | 74.91   | 85.53   | 27.02    | 69.99  | 51.63 | 57.07   | 70.77  | 61.78      | 82.86      | 50.25      | 68.99       | 71.30    | 68.61 |
>
> **Macro-F1:**
>
> | Method  | Caltech | Pets  | Cars  | Flowers | Food101 | Aircraft | SUN397 | DTD   | EuroSAT | UCF101 | ImageNet-A | ImageNet-R | ImageNet-S | ImageNet-V2 | ImageNet | Avg.  |
> | ------- | ------- | ----- | ----- | ------- | ------- | -------- | ------ | ----- | ------- | ------ | ---------- | ---------- | ---------- | ----------- | -------- | ----- |
> | w/o PFT | 92.23   | 89.85 | 65.12 | 68.78   | 83.16   | 23.31    | 64.40  | 47.02 | 47.82   | 63.58  | 55.76      | 78.29      | 45.75      | 63.87       | 65.67    | 63.64 |
> | L-TTA   | 92.46   | 90.22 | 65.54 | 68.99   | 83.33   | 24.14    | 64.68  | 47.50 | 48.27   | 63.86  | 55.97      | 78.56      | 45.99      | 64.19       | 65.83    | 63.97 |
>
> Our model maintains SOTA performance consistently under fixed hyperparameters, with minimal performance fluctuation impacting its superiority. Furthermore, we observe that PDHF exerts a slightly greater influence on Macro-F1 than on Accuracy, indicating its primary role in achieving more effective class rebalancing across complex data distributions and order variations. We believe this objective may be more challenging but also practically significant compared to the stable accuracy pursued by current works. We also include an extra experiment of this in the Table 8 of the revised paper. Thanks for your comment.
>
> In conclusion, L-TTA is robust to hyperparameter changes and achieves steady performance under a fixed set of hyperparameters. Thank you for your comment again.

---

> ### Author Response · Authors · 2025-11-21
> **Response to Reviewer 42WD (2/4)**
>
> ### **W2: BEM depends on access to class priors ($\pi$), which are not typically available in reality.**
> Thank you for pointing this problem out. We apologize for the ambiguity in our writing. The class priors in BEM are progressively computed through pseudo-label predictions during the TTA process, rather than being directly derived from the dataset. Also, we clarify that the head/tail classes estimated by π will continually change, but this does not substantially influence the performance of L-TTA. This is because our method is designed to operate under continuously changing class-frequency conditions. Thus, if a tail class gradually becomes a head class, SyPs, BEM, and RSs naturally adapt to the continually changing statistics without inducing instability or potential degradation. As further validation, we conducted an additional experiment in Table 7 of the revised paper, where we alter a coefficient controlling the order of head/tail classes in the data stream. The results show that L-TTA is robust to continually changing head/tail class distributions.
>
> ### **W3: This paper primarily emphasizes entropy-minimization, prompt-tuning, and prototype-based approaches, but it omits discussion of important emerging TTA families, such as vision-encoder adaptation methods, and reinforcement-based TTA frameworks.**
> Thank you for pointing out these important TTA approaches. We sincerely apologize for overlooking them in our initial submission. We have added these methods, along with several other non-i.i.d. TTA methods mentioned by the reviewers, to the introduction and related work sections. Furthermore, we have included these methods in our experimental comparisons, as shown in Tables 1, 2, 3, 4 and 5 of the revised paper. We appreciate your insightful comments!
>
> ### **W4: The notation and exposition can be difficult to follow, and parts of the method are challenging to comprehend. Clarity and narrative coherence could be improved to enhance readability.**
> We appreciate your comment regarding clarity. We agree that certain aspects of the notation and narrative can be streamlined and will revise the exposition to improve readability. Since incorporating these modifications directly into the revised paper during the rebuttal phase might reduce the clarity of the changes made, we plan to implement concrete revisions in the camera-ready version. Specifically, we will simplify the notation, add a high-level overview of the three modules before their formal definitions, and clarify how each component addresses its corresponding failure mode. We appreciate the reviewer's suggestion and are committed to substantially improving the paper's readability.
>
> ### **Q1: Does the model still outperform baselines when using a consistent set of hyperparameters across datasets? Demonstrating this would help confirm that the performance gains are not dependent on per-dataset tuning. A positive result here would increase my evaluation score.**
> Thank you for this insightful comment. Yes! L-TTA does not rely on per-dataset hyperparameter tuning. Tuning them is more of an incremental and optimal choice for higher efficiency, rather than an essential process which would reduce the practicality of TTA. We have conducted additional hyperparameter robustness experiments in our response to W1 regarding this factor. Could you please refer to our rebuttal for W1 above? Thank you for your time and patience in reviewing our rebuttal!
>
> ### **Q2: How does the model perform on balanced datasets?**
> Thank you for pointing out this important factor. We completely agree that our model should be tested on balanced datasets. We have added a new experiment without our imbalance settings on 10 fine-grained datasets in **Appendix K of the revised paper**. We also show the averaged results below:
> |      | CLIP   | TPT    | C-TPT  | MTA    | TDA    | ZERO   | RLCF   | DPE    | WATT   | CLIPArTT | O-TPT  | SCAP   | L-TTA |
> | ---- | ------ | ------ | ------ | ------ | ------ | ------ | ------ | ------ | ------ | -------- | ------ | ------ | ------------ |
> | Acc. | 63.912 | 64.818 | 63.525 | 64.281 | 68.163 | 63.496 | 64.587 | 68.952 | 62.853 | 63.252   | 63.867 | 66.258 | 70.399       |
> | Mac. | 61.264 | 62.178 | 61.162 | 62.051 | 65.508 | 60.45  | 61.768 | 65.506 | 62.315 | 60.088   | 61.021 | 62.383 | 66.961       |
>
> The results show that the performance gain of our L-TTA is consistent on balanced datasets as well as in long-tailed settings. Thank you for your time and patience in our rebuttal!

---

> ### Author Response · Authors · 2025-11-21
> **Response to Reviewer 42WD (3/4)**
>
> ### **Q3: Can the authors report performance on additional corruption types (e.g., blur, snow, JPEG compression)? I recognize the rebuttal timeline is limited, but even small-scale experiments would strengthen the evidence of robustness across diverse corruption scenarios.**
> Thank you for this valuable comment! We completely agree that our L-TTA and baseline TTA methods should be tested under LT+ even harsher scenarios, such as corrupted settings. We note that the corruptions you mentioned—{blur, snow, JPEG compression}—are part of the corruption types in ImageNet-C(h). Therefore, we have included **other 16 standard corruption types** (i.e., shot_noise, impulse_noise, speckle_noise, defocus_blur, glass_blur, motion_blur, zoom_blur, snow, frost, fog, brightness, contrast, JPEG_compression, pixelate, saturate, elastic_transform) and conducted additional experiments to the best of our ability. Due to the tight rebuttal timeline, we employed only 10 fine-grained datasets for these experiments; the severity level for all corruption types was set to **5**. We also regret that we were temporarily unable to include several computationally intensive baselines like WATT, CLIPArTT, and RLCF. **We will add their results in the final version and merge the experiments on additional corruption types with those on Gaussian noise.**
>
> The performance for each corruption type is provided in Appendix J of the revised paper. For your convenience, we show here only the results for the **types you mentioned (blur, snow, JPEG compression)** and **the averaged results across all 16 corruption types**. Please refer to Appendix J of the revised paper for complete results:
>
>
>
> **Blur (we select defocus blur)**
> | Metric | CLIP  | TPT   | C-TPT | MTA | TDA   | ZERO  | DPE   | O-TPT | SCAP  | L-TTA |
> | ----- | ----- | ----- | ----- | --- | ----- | ----- | ----- | ----- | ----- | ------------ |
> | Acc.| 14.27 | 14.68 | 12.29 | \\  | 15.12 | 13.79 | 15.59 | 12.45 | 14.82 | 23.43        |
> | Mac.| 11.02 | 11.39 | 10.02 | \\  | 11.30 | 10.31 | 13.46 | 10.14 | 12.89 | 20.34        |
>
> **Snow**
>
> | Metric | CLIP  | TPT   | C-TPT | MTA  | TDA   | ZERO  | DPE   | O-TPT | SCAP  | L-TTA |
> | ----- | ----- | ----- | ----- | ---- | ----- | ----- | ----- | ----- | ----- | ----- |
> | Acc. | 53.92 | 53.59 | 52.98 | \\   | 59.05 | 54.19 | 59.81 | 53.06 | 55.35 | 61.62 |
> | Mac. | 51.53 | 51.14 | 50.30 | \\   | 54.28 | 50.64 | 55.01 | 50.33 | 52.47 | 58.03 |
>
> **JPEG compression**
>
> | Metric | CLIP  | TPT   | C-TPT | MTA  | TDA   | ZERO  | DPE   | O-TPT | SCAP  | L-TTA |
> | ------ | ----- | ----- | ----- | ---- | ----- | ----- | ----- | ----- | ----- | ----- |
> | Acc.  | 54.21 | 57.1   | 54.204 | 51.31 | 59.74 | 54.95 | 61.31 | 54.08 | 57.17 | 63.72 |
> | Mac. | 51.26 | 54.42 | 50.74 | 47.93 | 55.14 | 51.07 | 56.28  | 50.65 | 54.93 | 59.62 |
>
> **Average over 16 types**
>
> | Metric | CLIP  | TPT   | C-TPT | MTA  | TDA   | ZERO  | DPE   | O-TPT | SCAP  | L-TTA |
> | ------ | ----- | ----- | ----- | ---- | ----- | ----- | ----- | ----- | ----- | ----- |
> | Acc.   | 39.86 | 41.50 | 39.95 | \\   | 43.24 | 39.75 | 43.57 | 39.52 | 42.15 | 47.84 |
> | Mac.   | 37.09 | 38.68 | 37.28 | \\   | 39.00 | 36.46 | 40.40 | 36.68 | 39.41 | 44.44 |
>
> Notes: "\\" indicates that the model fails to produce valid output.
>
> As shown in the table of averaged results, the performance gains of L-TTA are maintained and even further increased. Concretely, we have the following findings:
>
> 1. Training-free methods generally fail on this corruption benchmark. For example, MTA fails on 12 out of 16 corruption types (as observed from the detailed results in Appendix J), and the advantages of TDA and ZERO are also largely degraded. We believe this is because they rely heavily on the assumption that the dataset forms reliable and consistent feature clusters; however, this assumption is invalidated by corruptions, which fragment feature distributions and break the alignment between visual and text embeddings.
>
> 2. The performance gains of unimodal & training-required SOTA methods (like SCAP) exhibit degradation, gradually converging toward classic TPT. We attribute this to their lack of substantive cross-modal co-adaptation capability, which causes them to progressively amplify modality misalignments in the corruption benchmark.
>
> 3. The performance improvement achieved by L-TTA is both extensive and consistent. This is because, rather than making assumptions about class distributions, L-TTA fully leverages information from each individual sample (via SyPs and BEM) and employs proactive modality-shared learning (via RSs), which continuously mitigates intra- and inter-modal biases induced by corruption.

---

> > ### Author Response · Authors · 2025-11-21
> > **Response to Reviewer 42WD (4/4)**
> >
> > Refs:
> >
> > (a) Robust continual test-time adaptation against temporal correlation, NeurlPS 2022
> >
> > (b) Distribution alignment for fully test-time adaptation with dynamic online data streams, ECCV 2024
> >
> > (c) Towards stable test-time adaptation in dynamic wild world, ICLR 2023
> >
> > (d) Efficient Test-Time Adaptation of Vision-Language Models, CVPR 2024
> >
> > (e) WATT: Weight Average Test-Time Adaptation of CLIP, NeurlPS 2024
> >
> > (f) Dual prototype evolving for test-time generalization of vision-language models, NeurlPS 2024
> >
> > (g) SCAP: Transductive Test-Time Adaptation via Supportive Clique-based
> > Attribute Prompting, CVPR 2025

---

> ### Author Response · Authors · 2025-11-25
> **Looking forward to your feedback**
>
> Dear Reviewer 42WD:
>
> We greatly appreciate your professional feedback, which is invaluable to our research. We would be grateful if you could provide your insights at your convenience to help us further refine this work. We look forward to your valuable comments and are willing to discuss any aspects in more detail. Thank you!
>
> Sincerely,
>
> The Authors

---

> > ### Comment · Reviewer_42WD · 2025-11-26
> > **Response to authors' comments**
> >
> > I would like to sincerely thank the authors for their time and thorough revisions. I appreciate the effort they have put into addressing my concerns, and I am satisfied that the issues raised have been successfully resolved. I would be pleased to adjust my score accordingly.

---

> ### Author Response · Authors · 2025-11-27
> **Thank you very much for your positive feedback and your willingness to improve the score!**
>
> Dear Reviewer 42WD:
>
> Thank you very much for your positive feedback and your willingness to improve the score!
>
> Sincerely,
>
> The Authors

---

### Official Review · Reviewer_hHnt · 2025-11-02

**Soundness:** 3
**Presentation:** 3
**Contribution:** 2
**Rating:** 6
**Confidence:** 4

**Summary:**

This paper proposes L-TTA (Long-Tailed Test-Time Adaptation), a method for adapting vision-language models (VLMs) to long-tailed, unlabeled test distributions. The framework integrates three components—Synergistic Prototypes (SyPs), Rebalancing Shortcuts (RSs), and Balanced Entropy Minimization (BEM)—to mitigate head-class dominance during TTA. Experiments on 15 datasets show consistent gains over recent TTA baselines such as DPE and SCAP under class-imbalanced setups.

**Strengths:**

1. The paper is nicely organized, with convincing visuals and comprehensive benchmarks.
2. Evaluation across multiple datasets and metrics demonstrates the robustness of L-TTA.
3. The ablation study on components supports the effectiveness of SyPs, RSs, and BEM.

**Weaknesses:**

1. The “long-tailed TTA” setting largely overlaps with non-i.i.d. TTA explored in prior works such as LAME [a], NOTE [b], SAR [c], DA-TTA[d], and DELTA [e]. The challenges analyzed (imbalance over time, EM bias, boundary collapse) are identical to those already studied in non-i.i.d. test-stream TTA. Therefore, I doubt the claim that this is the “first study of TTA under long-tailed scenarios.” The authors may need to demonstrate the conceptual or procedural difference more clearly.
2. Because of the overlap mentioned above, the proposed method also appears similar to existing ones. BEM modifies EM by adding a penalty that down-weights already-confident classes to reduce head-class over-optimization. This seems very close to class reweighting in non-i.i.d. TTA (e.g., DELTA), which injects class-wise weights to counter imbalance during EM. As written, Propositions 1–2 provide intuition but not a decisive distinction from prior class-aware EM approaches, which makes this component appear incremental rather than novel.
3. Section 2 lacks explicit discussion of non-i.i.d. TTA. Adding a subsection contrasting L-TTA with these prior studies is essential for proper positioning.
4. Table 4 omits training-free TTA methods for VLMs (e.g., TDA, ZERO [f]), which are critical for comparisons in speed/accuracy trade-offs.

[a] Parameter-free online test-time adaptation. CVPR 2022.\
[b] Note: Robust continual test-time adaptation against temporal correlation. NeurIPS 2022.\
[c] Towards stable test-time adaptation in dynamic wild world. ICLR 2023.\
[d] Distribution alignment for fully test-time adaptation with dynamic online data streams. ECCV2024.\
[e] Delta: Degradation-free fully test-time adaptation. ICLR2023. \
[f] Frustratingly Easy Test-Time Adaptation of Vision-Language Models. NeurIPS2024.

**Questions:**

1. How is the long-tailed TTA setting formally different from non-i.i.d. TTA? Could the authors provide a side-by-side comparison of assumptions and experimental protocols?
2. How does BEM differ empirically from previous class-weighted EM variants such as in DELTA [e]?

---

> ### Author Response · Authors · 2025-11-21
> **Response to Reviewer hHnt (1/3)**
>
> Thank you for these valuable opinions. We sincerely appreciate your review and are grateful for the time you have spent evaluating our submission. We are glad that you found our paper to be well-organized, with convincing visuals and benchmarks, and that the experiments and ablation studies are substantial. We wish to address your concerns by providing detailed responses to each of your proposed weaknesses **W1**, **W2**, **W3**, **W4** and questions **Q1**, **Q2** as follows:
>
>
> ### **W1: The “long-tailed TTA” setting largely overlaps with non-i.i.d. TTA explored in prior works. The challenges analyzed (imbalance over time, EM bias, boundary collapse) are identical to those already studied in non-i.i.d. test-stream TTA.**
>
> Thank you for this comment. We acknowledge with regret that our presentation did not adequately highlight the fundamental differentiators between LT-TTA and existing Non-i.i.d. TTA paradigms. We clarify that long-tailed TTA and non-i.i.d. TTA are orthogonal concepts rather than variants of each other. The essential conceptual difference can be viewed from four aspects:
>
> 1. From the perspective of assumptions, Non-i.i.d. TTA assumes temporal drift of the test data, but it is not systematically biased toward certain classes; Long-tailed TTA instead focuses on a highly skewed class-level imbalance distribution which induces failure even under stationary covariates.
> 2. From the perspective of failure modes, Non-i.i.d. TTA mainly suffers from feature or domain shift, but Long-tailed TTA suffers from head-class accumulation and tail erosion. In VLMs, the issue is further amplified by modality misalignment and textual priors (we have detailed explanations of these in the rebuttal to Reviewer dibS's Q1), which do not appear in unimodal TTA.
> 3. From the perspective of evaluation criteria, Non-i.i.d. works typically report only the overall accuracy, while for Long-tailed TTA, we report the accuracy, macro-F1, and head/tail accuracy (in the appendix), as we also pursue class balance rather than solely overall accuracy.
> 4. From the perspective of methodology, existing Non-i.i.d. methods assume balanced labels and do not address the head-tail imbalance, while our L-TTA is designed specifically for these long-tailed and VLM-specific failure modes.
>
> Based on the above analysis, Long-tailed TTA is distinct from Non-i.i.d. TTA and poses unique challenges. This analysis also intuitively explains why some Non-i.i.d. state-of-the-art methods like WATT(a) and CLIPArTT(b) (introduced by Reviewer 42WD) yield gradually worse performance when the imbalance ratio increases, or when their leveraged Non-i.i.d. settings are entangled with long-tailed settings. Please see Tables 1, 2, 3, 4 and Appendix J of the revised PDF for these experiments.
>
> From another perspective, in VLMs, long-tailed imbalance creates new failure modes not present in unimodal non-i.i.d. TTA. Concretely, due to the participation of the text modality, the long-tailed problem is entangled with modality misalignment and textual-prior amplification. In our response to Reviewer dibS's Q1, we provided a detailed analysis of the distinctive characteristics of Long-tailed TTA within **VLMs**. We hope you might refer to it if interested.
>
> We also thank you for pointing out the potential impropriety of our claim about being the "first study of TTA under long-tailed scenarios." We want to clarify that our intended meaning is: "first study of TTA under long-tailed scenarios with VLMs," where both the long-tailed setting and the VLM configuration possess distinct significance and are indispensable, based on our analysis above. We have added this analysis to the introduction and related work of the revised paper and will refine our formulations to better explain the significance of our configuration.

---

> ### Author Response · Authors · 2025-11-21
> **Response to Reviewer hHnt (2/3)**
>
> ### **W2: The proposed BEM appears similar to existing ones like DELTA, which makes this component appear incremental rather than novel.**
>
>
> Thank you for pointing out this related work, DELTA(a). We agree that a component of DELTA named Dynamic Online reweighTing (DOT) was proposed earlier to address long-tailed TTA, which might make BEM appear to be an incremental contribution. We will clarify the relationship and distinctions between these two methods:
>
> We acknowledge that the core design of BEM is to penalize head classes through an adaptive logit adjustment term. This functionality is conceptually similar to DOT, which dynamically updates a class-frequency vector via momentum and assigns lower weights to samples from head classes. However:
>
> 1. From the perspective of motivation, BEM only modifies the final logits and is designed as a straightforward substitute for EM. In comparison, loss-reweighting methods like DELTA require delicate per-class weight calculation and re-normalization. This makes them suffer from more hyperparameter tuning and substantial extra computational burden than BEM, reducing their applicability. Furthermore, DELTA is highly sensitive to batch size, smoothing factors, and momentum coefficients, whereas BEM, and our L-TTA overall, are quite robust to hyperparameter changes, enhancing their practical applicability.
>
> 2. From the perspective of application scenarios, we emphasize that our method is constructed for VLM-based TTA. Here, EM not only sharpens confidence for the highest class but also affects the correlation between visual and text modalities. **While most loss re-weighting methods were proposed for vision-only models, directly applying them to VLM-based TTA faces the unique challenge of cross-modal bias amplification (a point also raised by Reviewer dibS)**. Furthermore, given that loss reweighting itself can introduce training instability, its concrete effectiveness on VLMs remains questionable. In comparison, BEM only aims to penalize over-confidence through logit adjustment without explicitly manipulating inter-class losses like DOT. This avoids potential risks with minimal cost.
>
> In conclusion, while DOT is a well-motivated prior work, BEM differs in both motivation (a plain substitute for EM vs. a more computationally complex method) and applicability (logit penalty on high-confidence classes vs. loss reweighting that may induce high instability in VLMs). We will clarify these differences more explicitly in the final version to avoid conflation.
>
> **Additionally**, although we do not expect BEM's minimalist design to surpass DOT or DELTA, we aim to compare them to identify their respective strengths and limitations. However, to the best of our efforts, we did not find an official implementation of DELTA and may need to reproduce it ourselves. Considering the tight rebuttal timeline, we plan to reproduce DELTA and compare it with BEM in the final version. Thank you for your time and patience in reviewing our work.
>
> ### **W3: Section 2 lacks explicit discussion of non-i.i.d. TTA.**
> Thank you for the suggestion. We agree that our related work section requires an explicit comparison with non-i.i.d. TTA. In the revised version we have uploaded, we have added a dedicated section summarizing the assumptions in prior non-i.i.d. works and clearly contrasting them with long-tailed TTA, highlighting why existing methods are not directly applicable to long-tailed and VLM-based TTA. We believe this addition improves the paper's positioning and clarity. Thank you again for your comments!
>
> ### **W4: Table 4 omits training-free TTA methods for VLMs (e.g., TDA, ZERO [f]).**
> Thank you for this valuable comment. We have added these methods to the efficiency comparison (Table 5 of the revised paper). We find that these training-free methods are typically the fastest but show very limited performance in the L-TTA setting; moreover, they consistently fail in harsher scenarios (such as the LT+corruption benchmark (b) proposed by Reviewer 42WD). In comparison, L-TTA may require more inference time but demonstrates substantial robustness and outperforms these methods by up to 8.04%. We provide a more detailed analysis in the new efficiency comparison section of the revised paper. Thank you!
>
> ### **Q1: How is the long-tailed TTA setting formally different from non-i.i.d. TTA? Could the authors provide a side-by-side comparison of assumptions and experimental protocols?**
> Thank you for this comment. We completely agree that a systematic comparison between long-tailed TTA and non-i.i.d. TTA is highly necessary. Considering that this question is strongly linked to Weakness 1 (W1) above, could you please refer to our rebuttal to W1, where we have conducted a deeper investigation into this exact point? Thank you for your time and patience in reviewing our rebuttal!

---

> ### Author Response · Authors · 2025-11-21
> **Response to Reviewer hHnt (3/3)**
>
> ### **Q2: How does BEM differ empirically from previous class-weighted EM variants such as in DELTA?**
> Thank you for this comment. We agree that our BEM may appear somewhat similar to previous class-reweighting methods based on EM. Considering that this question is strongly linked to Weakness 2 (W2) above, could you please refer to our rebuttal to W2, where we have conducted a deeper investigation into this exact point? Thank you for your time and patience in reviewing our rebuttal!
>
>
> refs:
>
> (a) Delta: degradation-free fully test-time adaptation, ICLR 2023
>
> (b) Hendrycks D, Dietterich T. Benchmarking Neural Network Robustness to Common Corruptions and Perturbations, ICLR 2019

---

> ### Author Response · Authors · 2025-11-25
> **Looking forward to your feedback**
>
> Dear Reviewer hHnt:
>
> We greatly appreciate your professional feedback, which is invaluable to our research. We would be grateful if you could provide your insights at your convenience to help us further refine this work. We look forward to your valuable comments and are willing to discuss any aspects in more detail. Thank you!
>
> Sincerely,
>
> The Authors

---

### Author Response · Authors · 2025-11-25
**Thank you for your valuable feedback! We hope our revisions have addressed your questions, and we look forward to any further discussion.**

Dear Reviewers, Area Chairs, Senior Area Chairs and Program Chairs:

We thank all reviewers for their valuable and constructive comments. In response to the reviewers' concerns, we have implemented the following improvements and submitted a revised paper. Revised content is highlighted in blue.

**(Reviewer dibS)**
1. **VLM-specific Motivations:** We thank the reviewer for this comment. We have clarified our motivation in the rebuttal below and expanded the discussion in the introduction, emphasizing the unique challenges of LT-TTA with VLMs. These additions appear in **Section 1** of the revised paper.

2. **Experiments with larger and stronger backbones:** We appreciate this comment. We have conducted additional experiments using four larger backbones, as shown in the rebuttal below and in **Table 3** of the revised paper. Per-dataset results are available in **Appendix L (Tables 40–44)**.


**(Reviewer QBXH)**
1. **Distinction between Long-tailed (LT)-TTA and traditional LT learning:** We thank the reviewer for raising this point. We have included a detailed discussion in the rebuttal below and added it in the revised paper.

2. **Computation complexity analysis:** Thanks for this comment. We have included training-free methods (TDA, ZERO), reinforcement-learning-based methods (RLCF), and visual-adaptation-based methods (WATT, CLIPArTT) in **Table 5** of the revised paper. We also added a “memory cost” metric for a more comprehensive comparison. These results are also shown in the rebuttal below.

3. **Hyper-parameter sensitivity:** Thanks for this comment. We have clarified potential misunderstandings in the rebuttal below and provided additional analysis on hyper-parameter tuning. These results are included in **Table 8** of the revised paper.

4. **Intuition behind the synergy between (Synergistic Prototypes) SyPs and (Rebalancing Shortcuts) RSs:** Thanks for this comment. We have provided a detailed explanation in the rebuttal below and incorporated this discussion into **Section 3** of the revised paper.

5. **Robustness to continually shifting head/tail classes:** Thanks for this comment. Additional experiments and analysis on this issue are provided in the rebuttal below and in **Table 7** of the revised paper.

6. **Extending SyPs to dynamic prototypes:** Thank your for this insightful suggestion!  We have discussed this idea in detail in the rebuttal below.


**(Reviewer 42WD)**
1. **Hyper-parameter finetuning efficiency / comparisons between using fixed hyper-parameters and using per-dataset finetuned hyper-parameters:** In response to this valuable comment, we have added experiments comparing fixed vs. per-dataset tuned hyper-parameters in the rebuttal below., along with detailed analysis. This experiment is also included in the **Table 8** of the revised paper.

2. **$\pi$ in the Balanced Entropy Minimization:** We apologize for any confusion and have clarified the use of $\pi$ in the paper and in the rebuttal below.

3. **Discussion of other TTA methods:** We have now included additional TTA methods in **Section 2** and in all comparative experiments.

4. **Notation issues:** Thanks for this comment. We apologize for this problem and will substantially improving the paper's readability. We have deeper discussion towards this point in the following rebuttal window.

5. **Experiments on balanced datasets:** We have added these results in the rebuttal below and in Appendix K (Table 45) .

6. **Experiments on other corruption types:** Thanks for this comment. We have added the experiments on other 16 corruption types in **Appendix J (Tables 23–39)**. Specific results on your mentioned types (defocus blur, snow, jpeg compression) are also included in the rebuttal below.

**(Reviewer hHnt)**
1. **Distinction between LT-TTA and Non-i.i.d. TTA:** We have provided a detailed clarification in the rebuttal below and added this discussion to Sections 1, 2 of the paper.

2. **The discrepancy between Balanced Entropy Minimization and DELTA:** Thanks for your comment. We have a detailed clarification towards this point in the rebuttal below. Thanks for your time and patience on your paper. This part is also added to the **Section 2** of the revised paper.

3. **Discussion of Non-i.i.d. TTA methods:** We appreciate your comment. We have a detailed clarification towards this point in the rebuttal below. We also add the Non-i.i.d. TTA methods in the **Section 2** of our revised paper.

4. **Computation complexity analysis:** We have incorporated training-free methods (TDA, ZERO) and other relevant approaches into the computational comparison in the rebuttal below and   **Table 5** of the revised paper.

We are deeply grateful to all reviewers for their insightful suggestions, which have significantly enhanced the quality of our work. Should any further questions arise, we are more than willing to provide additional clarification. We look forward to the opportunity for further discussion.

---

### Author Response · Authors · 2025-11-30
**The Authors' Comment to the Area Chair**

Dear Area Chair:

Thank you for your attention to our paper. To facilitate your evaluation, we summarize the content and rebuttal status of our paper in the following:

### **Overview of our paper:**

This paper introduces the first Test-Time Adaptation (TTA) method for Vision-Language Models (VLMs) under Long-Tailed (LT) settings, termed L-TTA. Specifically, existing approaches exhibit two distinctive failure modes in VLM-based LT-TTA: *text-induced tail erosion* and *modality-bias amplification*. To address *text-induced tail erosion*, this work proposes **Synergistic Prototypes (SyPs)**, which mitigate biased text embeddings by constructing two complementary and unbiased prototypes. To counter *modality-bias amplification*, the paper introduces **Rebalancing Shortcuts (RSs)**, which employ learnable vectors in a modality-shared space and balance their contributions to capture more uniformly distributed patterns across classes. Additionally, we present **Balanced Entropy Minimization (BEM)**, a plug-and-play variant of vanilla Entropy Minimization that can be integrated into any long-tailed TTA environment. Extensive and rigorous experiments demonstrate the effectiveness of our approach, while comprehensive ablation studies confirm the synergistic contributions of our proposed components.

### **Overview of the rebuttal status:**

In the initial reviews, the reviewers spoke highly of the following advantages of our paper:

- **All reviewers recognized that our method addressed a highly practical and significant problem!**
- Reviewer hHnt specifically praised the compelling narrative and well-organized presentation;
- Reviewers hHnt, 42WD, and QBXH commended our extensive and thorough experiments;
- Reviewers hHnt and dibS appreciated the clarity and synergistic design of our proposed modules;
- Reviewers hHnt and 42WD highlighted the detailed and comprehensive ablation studies;
- Reviewer QBXH noted the strong theoretical grounding of our method.

At the same time, the reviewers also raised several concerns in their initial reviews. **These issues have been thoroughly addressed point-by-point, as listed in the following comment window.**

Before the time when the reviewer replies were disabled, **Reviewer 42WD (initial score: 4)** has decided to increase the score as our rebuttal successfully addressed all the questions. **Reviewers QBXH (initial score: 6) and hHnt (initial score: 6)** maintained their scores, with Reviewer QBXH responding and expressing explicit satisfaction with our responses. One concern of **Reviewer dibS (initial score: 4)** was not fully resolved at that time due to the stringent rebuttal timeline; Now we have addressed this remaining concern through an additional detailed clarification. We summarize our follow-up rebuttal here:

### **We Have Resolved the Only Remaining Question from Reviewer dibS:**

The remaining concern from Reviewer dibS (mentioned in the response to our rebuttal) is the experiments supporting our narrative to the VLM-specific failure modes. We acknowledge that we did not directly present the experimental evidence in the opening section initially, but our extensive experiments in the paper and the appendix (particularly those on unimodal methods) substantially validate our claims. Furthermore, this concern was not raised by the other three reviewers, with Reviewer hHnt even commending the sufficiency of our claims. Consequently, we think that this issue stems from both room for improvement in our narrative presentation, and the possibility that our reviewer may not have been able to thoroughly consult all sections of the manuscript. Now we have incorporated Reviewer dibS's feedback to make our narrative even more clearer. We also provided the reviewer with a further detailed explanation to facilitate a better understanding of the paper. We believe we have now fully addressed the reviewer's concerns. Thank you for your attention and contribution.


In summary, we had addressed almost all concerns by the time the reviewer response system was closed, receiving three positive evaluations. For the only remaining issue from Reviewer dibS, we have provided detailed explanations. We trust our follow-up rebuttal fully resolves the remaining doubt and appreciate your recognition of our efforts within the tight rebuttal timeline. We sincerely thank you for your attention to our paper and your valuable contributions to the conference.

Sincerely,

The Authors

---

### Meta-Review · Area_Chair_gQ1s · 2025-12-24

**Summary:**

The reviews for this paper are mixed. This paper was reviewed by 4 experts in the field and received the following scores: 2 Marginal Reject (4), and 2 Marginal Accept (6).

This paper introduces Long-Tailed Test-Time Adaptation (L-TTA), a framework designed to adapt vision–language models (VLMs) to unlabeled, long-tailed test distributions, where standard test-time adaptation methods tend to overfit to dominant classes. To address this challenge, L-TTA integrates three main idea: (1) Synergistic Prototypes (SyPs), which maintain dual prototype memories to capture both confident and exclusionary features and enrich tail-class representations; (2) Rebalancing Shortcuts (RSs), learnable cross-attention adapters with a class re-allocation loss that dynamically balance head and tail class influence; and (3) Balanced Entropy Minimization (BEM), a confidence-gated objective that mitigates optimization bias toward head classes during adaptation. Extensive experiments across 15 long-tailed datasets show that L-TTA consistently outperforms recent TTA methods.


The reviewers generally agree that the article is well-written, despite a few missing details. They also acknowledge the importance of the topic but feel that better positioning is needed. The authors carefully reviewed the article and addressed most of the points raised. I believe the main challenge now lies in improving the writing, which is entirely achievable. Therefore, I believe the article should be accepted.

**Reviewer Concerns:**

Initially, the reviewers noted the following weaknesses:

1. interest of Long tail TTA vs non IID TTA : some reviewers did not understand the interest of long-tail TTA, and the proposed BEM algorithm appears similar to EM.

2. Missing Baselines : some reviewers also criticized the absence of some baseline methods, particularly for non-IID TTA, as well as for certain general TTA techniques.

3. Extra Complexity due to the number of hyperparameters in the techniques: some reviewers criticized the number of parameters in the techniques, and I fully agree with them. The article is difficult to read, the writing quality is poor, and Figure 3, intended to facilitate reader understanding, is so complex that it has the opposite effect. I emphasize that the authors need to improve the clarity of the method descriptions. The excessive number of parameters and notations makes the text extremely difficult to read. I also share the reviewers' criticism of this high number of parameters, which is contrary to the objective of a TTA. How can someone adapt L-TTA to a particular new dataset? This question is important!
 4. Too strong assumptions:  the assumption that BEM loss depends on the access to class priors ( $\pi$), which are generally not available in practice, seems a bit strange according to some reviewers.

5. Limited methodological novelty: the proposed technique appears to be somewhat incremental, as it mainly reuses of well-known ideas rather than introducing fundamentally new mechanisms.

6. Hyperparameter sensitivity: the method relies on several new hyperparameters that have not been fully investigated.

7. Computational overhead not fully analyzed: some reviewers criticized the fact that all these additional costs were not properly investigated. 8. Motivations specific to VLM: I am not sure I fully understand this criticism, but I think it is related to what is unique to VLM.

**Reviewer Scores:**

I would like to thank the authors, who also did a remarkable and honest job in summarizing their responses to the various weaknesses. At the beginning of the rebuttal, the paper's score was two 6s (marginally above the acceptance threshold) and two 4s (marginally below the acceptance threshold
). After the rebuttal, the score would have been three 6s (marginally above the acceptance threshold) and one 4 (marginally below the acceptance threshold
). This makes this paper borderline positive. I took the time to carefully read the comments and the entire paper. I checked, and I think, like the authors, that many points have been addressed. I wouldn't say everything has been resolved, but I think the authors have addressed most of the points concerning baselines, sensitivity analysis, experiments, and clarifications on funding. However, one important point is missing regarding the form of the paper: a better presentation of the methodology. I want to mention that I also took the time to read the paper in what is supposed to be the final version, and I think there is still work to be done. I want to emphasize that this point is crucial for the article's acceptance. I am convinced that the authors can make these changes to the final version. That is why I propose accepting the article and would like to congratulate the authors.

---

### Decision · Program_Chairs · 2026-01-26

Accept (Poster)